



# The Earth system model CLIMBER-X v1.0. Part 1: climate model description and validation

Matteo Willeit[1], Andrey Ganopolski[1], Alexander Robinson[2,3,1,4], and Neil R. Edwards[5]

[1]Potsdam Institute for Climate Impact Research, Potsdam, Germany
[2]Complutense University of Madrid, Madrid, Spain
[3]Geosciences Institute CSIC-UCM, Madrid, Spain
[4]Climate and Global Dynamics Laboratory, National Center for Atmospheric Research, Boulder, CO 80305, USA
[5]Environment, Earth and Ecosystems, The Open University, Walton Hall, Milton Keynes, MK7 6AA, UK

**Correspondence:** Matteo Willeit (willeit@pik-potsdam.de)

**Abstract.** The newly developed fast Earth system model CLIMBER-X is presented. The climate component of CLIMBER-X consists of a 2.5D semi-empirical statistical-dynamical atmosphere model, a 3D frictional-geostrophic ocean model, a dynamic-thermodynamic sea ice model and a land surface model. All model components are discretized on a regular lat-lon grid with a horizontal resolution of $5° \times 5°$. The model has a throughput of $\sim 10{,}000$ simulation years per day on a single node with 16 CPUs on a high performance computer and is designed to simulate the evolution of the Earth system on temporal scales ranging from decades to $>100{,}000$ years. A comprehensive evaluation of the model performance for present day and the historical period shows that CLIMBER-X is capable of realistically reproducing many observed climate characteristics, with results that generally lie within the range of state-of-the-art general circulation models. The analysis of model performance is complemented by a thorough assessment of climate feedbacks and model sensitivities to changes in external forcings and boundary conditions. CLIMBER-X also includes a detailed representation of the global carbon cycle and is coupled to an ice sheet model, which will be described in separate papers. CLIMBER-X is available as open–source code and is expected to be a useful tool for studying past climate changes and for the investigation of the long-term future evolution of the climate.

## 1 Introduction

Contemporary Earth system models (ESMs), based on relatively coarse resolution (i.e. order of 200-300 km) atmosphere general circulation models (GCMs) and non-eddy resolving ocean models, currently reach simulation speeds of up to several hundred model years per day. This is sufficient to perform a single simulation or a small ensemble of simulations with the duration of several thousand years, the time typically needed to reach full equilibrium of the climate system and to perform near-future climate projections (IPCC 2013). However, there is also a need for much more computationally efficient climate and ESMs suitable for performing simulations on orbital and even longer time scales, as well for large multi-millennial ensembles of model simulations. This is why a certain ecological niche still remains for models of intermediate complexity, usually referred to as EMICs (Claussen et al., 2002). These computationally efficient models historically have rather coarse spatial resolution and, usually, employ simplified atmospheric components based on energy-balance (i.e. UVic (Weaver et al., 2001), BERN-3D





(Ritz et al., 2011)), statistical-dynamical (CLIMBER-2 (Petoukhov et al., 2000), CLIMBER-3$\alpha$ (Montoya et al., 2005)), quasi-geostrophic (LOVECLIM (e.g. Goosse et al., 2010)) and 3-D (GENIE (Lenton et al., 2007) and GENIE-PLASIM (Holden

et al., 2016)) atmospheric models. Their oceanic components range from 2-D zonally averaged to full 3-D ocean general circulation models. Since these models are positioned as Earth system models, they usually include a number of other than the standard climate models components, such as global carbon cycle, ice sheets, dynamical vegetation, etc.

One such model is CLIMBER-2 (Ganopolski et al., 2001; Petoukhov et al., 2000). Its climate component was originally developed more than 20 years ago and has been widely used, primarily but not exclusively, for the study of past climates (e.g.

Willeit et al., 2019; Ganopolski and Brovkin, 2017; Ganopolski et al., 2016). This model has a very coarse spatial resolution (minimal geographically explicit models) which had been dictated by the need for performing transient simulations on orbital time scales with the computers available at that time. Significant progress in computer performance made such coarse resolution unnecessary. At the same time, the availability of a vast amount of both observational data and results of complex climate and Earth system models open up the possibility to improve model performance and to ensure its consistency with more complex

models.

Since on the one hand there is still demand for such models and, on the other hand, the availability of data and faster computers now enable the development of much better models but with a similarly low computational cost, we decided to develop a new model, CLIMBER-X. CLIMBER-X is based on a similar philosophy as CLIMBER-2 but with higher resolution, more internally consistent components and better treatment of individual processes.

A schematic illustration of all components of CLIMBER-X is presented in Fig. 1. Here we describe only the climate core of CLIMBER-X, which consists of the atmosphere model SESAM, the ocean model GOLDSTEIN, the sea ice model SISIM and the land surface-vegetation model PALADYN. CLIMBER-X also represents the global carbon cycle and therefore allows interactive simulation of the atmospheric $CO_2$ and $CH_4$ concentrations. Besides PALADYN, which simulates carbon cycle processes on land, the model includes HAMOCC (Maier-Reimer and Hasselmann, 1987; Ilyina et al., 2013) as the ocean

biogeochemistry and marine-sediment model. The ice sheet models SICOPOLIS (Greve, 1997) and Yelmo (Robinson et al., 2020) are both included in CLIMBER-X as optional land-ice components and are coupled with the climate component via an updated version of the physically based surface energy and mass balance interface SEMI (Calov et al., 2005) and a basal ice-shelf melt module. The viscoelastic mantle model VILMA (Klemann et al., 2008; Martinec et al., 2018) is used to simulate the bedrock response to changes in loading by solving the sea–level equation. The global carbon cycle model and ice sheet

coupling will be described in detail in forthcoming papers.

## 2  Model description

The climate components of CLIMBER-X include the Semi-Empirical dynamical-Statistical Atmosphere Model (SESAM), the frictional-geostrophic 3D ocean model GOLDSTEIN, the SImple Sea Ice Model (SISIM) and the land model PALADYN (Fig. 1). They are coupled through fluxes of energy and water, the properties which are conserved in the entire system. Momen-

tum is not conserved in the system but surface wind stress is computed in the atmosphere and passed to the ocean component.




The atmosphere, ocean, sea–ice and land models are all discretized on the same regular longitude-latitude grid with a horizontal resolution of $5° \times 5°$, which facilitates exchanges among them. The climate model components also share the same base timestep of one day, which is also the coupling frequency between modules. Shorter timesteps are used internally in the atmosphere and sea ice models for reasons of numerical stability. CLIMBER-X is designed to simulate the mean climatological

state. It does not explicitly resolve the diurnal cycle and synoptic variability, although the effect of synoptic weather systems on the transport of energy and water is accounted for. The model does not represent interannual internal climate variability such as the El-Nino Southern Oscillation (ENSO).

In CLIMBER-X the land-ocean distribution, topography and land cover can evolve with time. A high-resolution ($\approx 10 \, \mathrm{arc-minutes}$) topography input file is used to automatically generate the land-sea mask, the topography, the orographic roughness and the

ocean bathymetry. For present-day the RTopo-2 data of Schaffer et al. (2016) are used. Runoff routing is also derived automatically from the high-resolution topography by following the steepest slope and using the algorithm of Planchon and Darboux (2002) to fill depressions. No manual intervention is needed in the process, so that CLIMBER-X can in principle also be applied for simulations of times in the past when the distribution of continents was very different from present day. CLIMBER-X also allows for dynamic transient changes of the land-sea mask, which is important for simulations on long time scales, when

sea-level changes or ice-sheet expansion and retreat can cause substantial changes to the land and ocean areas, with important effects on the fluxes of energy and water between the surface and the atmosphere. Furthermore, because the model resolution is relatively coarse and the removal or addition of a whole cell could cause a large perturbation in the model, we have implemented land/sea fractions in each grid cell to ensure more continuous transitions. This fractional approach, common to many models, also increases the effective horizontal model resolution.

CLIMBER-X is written entirely in Fortran 90+, making use of derived types to structure the single model components in a way which makes information transfer and exchange between different modules more transparent. The code is parallelized using the shared memory OpenMP standard and integrates around 10,000 model years in one day on a single node with 16 CPUs on a high performance computer (Lenovo NeXtScale nx360M5, Xeon E5-2667v3 8C 3.2GHz, Infiniband FDR14 Lenovo/IBM). The model is therefore $\approx$100–1000 times faster than state-of-the-art climate models that simulate weather,

when using comparable computational resources. Model input/output relies completely on the NetCDF data standard and uses the NCIO package (Robinson and Perrette, 2015) to read from and write to files. It therefore depends on the NetCDF library being available on the system where the model is run. Experimental settings and model parameters are defined in namelists, with one for each model component. A Python script is used to easily generate ensembles of simulations with different settings and parameters.

A conceptual overview of the individual model components is given in the following sections, while more details and equations are given in the Appendix.

## 2.1 Atmosphere model: SESAM

The atmospheric component of CLIMBER-X is designed as the Semi-Empirical dynamical-Statistical Atmosphere Model (SESAM), based on a similar approach to the atmosphere of CLIMBER-2 (Petoukhov et al., 2000). Originally, the atmo-



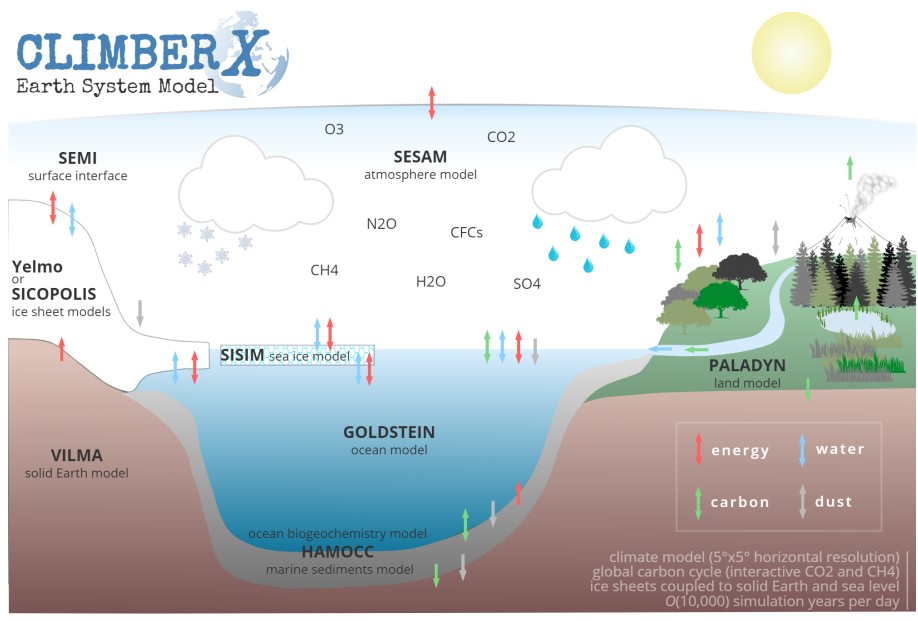

**Figure 1.** Schematic illustration of the CLIMBER-X model, including exchanges and coupling between the different modules.

spheric component of CLIMBER-2 was positioned as a model based on equations derived (with some assumptions) from first
principles. However, it has been found that the governing equations and some key parameterizations are hard to derive from
first principles and a number of modifications have been necessary to match results with observational data or results of more
physically based models. This approach is explicitly and extensively used during the development of SESAM. Compared to
CLIMBER-2, the new atmospheric component of CLIMBER-X has a much higher spatial resolution ($5° \times 5°$) and most key
parameterizations have been modified to improve performance against present-day climate and GCM sensitivity experiments.
We extensively made use of available high-quality climatological data and results of complex climate model simulations which
were not available when the CLIMBER-2 model was developed. This approach is reflected by the term semi-empirical in the
model acronym.

The spatial resolution of SESAM, in terms of its representation of atmospheric fields, can be characterized as 2.5 dimen-
sional. This is related to the fact that, although prognostic variables (temperature, specific humidity, dust and eddy kinetic
energy), as well as many diagnostic characteristics (cloudiness, condensation, short-wave radiation, etc.) are 2-dimensional,
the calculation of horizontal energy and water transport and the vertical fluxes of longwave radiation make use of information
about the 3-D distribution of temperature, humidity and wind velocity. At the surface, each model grid cell is divided into
different macro surface types, namely ocean, sea ice, land and ice sheets. The macro surface types share the same atmospheric
column but differ in surface elevation, albedo and surface temperature and humidity. Sensible heat flux, evaporation, shortwave
and longwave radiation fluxes are computed separately for each macro surface type.





The 3D atmospheric structure is derived using assumptions about the universal vertical structure of temperature and relative humidity in the atmosphere. The temperature profile through the troposphere is a quadratic function of height, except in a 1500 m layer close to the surface where the lapse rate is a function of the near-surface atmospheric stability. Inversion strength is determined by the surface radiative balance and the difference between near-surface air temperature and skin temperature. Temperature is vertically uniform in the stratosphere. The tropopause height is derived assuming that the stratosphere is in radiative equilibrium. Relative humidity is taken to be constant within the planetary boundary layer, to decay exponentially in the low troposphere and to remain constant throughout the rest of the troposphere. Stratospheric relative humidity is prescribed at a uniform constant value. This approach differs from that used in CLIMBER-2, where specific humidity profiles were prescribed instead of relative humidity. This can lead to large biases in relative humidity in the mid-upper troposphere, where humidity is important for its effect on longwave radiation. Therefore, in SESAM the vertical profile of relative humidity is parameterized instead. See Appendix A1 for more details on the parameterisation of the vertical structure.

Similarly to CLIMBER-2, in SESAM the horizontal wind velocity is divided into geostrophic and ageostrophic components. The geostrophic components of velocity at sea level are computed from sea-level pressure (SLP) and at any height through the troposphere by using the thermal wind approximation. The components of the surface wind are computed using the Taylor model (see e.g. Hansen et al. (1983)). The ageostrophic wind components in the planetary boundary layer are computed from the sea-level pressure gradient and the cross-isobar angle. Sea-level pressure is the sum of zonally averaged and azonal components. The zonal component of SLP is derived based on the assumption that the strength and position of the major cells of the atmospheric meridional overturning circulation are controlled by average meridional temperature gradients and zonally averaged surface drag, similarly to Petoukhov et al. (2000). An additional dependence on zonal mean surface elevation has been added in SESAM to account for the effect of topography. The zonally averaged SLP is directly related to the zonal mean meridional (ageostrophic) wind in the planetary boundary layer and therefore the mean meridional circulation. The vertical structure of the zonally averaged meridional circulation is parameterized, following the approach of Petoukhov et al. (2000). The azonal component of mean SLP resulting from stationary planetary waves is separated into a thermal and an orographic component. The thermal component is assumed to be proportional to the azonal surface temperature anomaly reduced to sea level using the interrelation between long-term large-scale azonal temperature and pressure in quasi-stationary planetary-scale waves, similarly to Petoukhov et al. (2000). The effect of topographic stationary waves is particularly important for the NH mid-latitude winter. The simple 1D barotropic model for forced topographic Rossby waves of Charney and Eliassen (1949), applied to each latitudinal belt separately, is used to account for this effect in SESAM. Note that in CLIMBER-2 topographic waves were not represented because of the low spatial model resolution, whereas this limitation does not exist for CLIMBER-X. The main limitation in atmospheric dynamics in SESAM lies in the employed geostrophic approximation, which is not valid close to the equator, resulting in deficiencies in the simulated dynamics in the tropics. See Appendix A2 for more details on the model dynamics.

SESAM includes a prognostic equation for vertically integrated energy and another for total-column water content, from which surface air temperature and humidity are derived. Sources and sinks of energy in the column include net longwave and shortwave radiation, surface sensible heat flux and latent heat release from condensation. Sources and sinks of water in the





column are given by surface evaporation and precipitation. The 3D wind field is used to advect the 3D potential temperature field and 3D specific humidity fields. Horizontal transport due to synoptic-scale processes is described as macroturbulent diffusion with an isotropic coefficient of horizontal diffusion expressed through the eddy kinetic energy (EKE). Precipitation

is generated in the model whenever near-surface air relative humidity exceeds a critical threshold of 95%. The water excess is then instantaneously added to precipitation. Over land, precipitation is additionally generated from column water content with a specified turnover time that is inversely proportional to atmospheric relative humidity. See Appendix A3 and Appendix A4 for more details on the energy and water balance equations.

The (vertically integrated) EKE is a prognostic variable in the model. EKE production is proportional to the baroclinicity of

the atmosphere as defined by the Eady baroclinicity measure (e.g. Hoskins and Valdes, 1990). EKE dissipation depends on the surface aerodynamic drag coefficient and is proportional to $\mathrm{EKE}^{3/2}$. EKE is itself transported by macroturbulent diffusion and advected by the wind at $700\,\mathrm{hPa}$. The synoptic component of surface wind speed is then taken to be proportional to $\sqrt{\mathrm{EKE}}$. See Appendix A5 for more details on the model representation of synoptic processes.

The shortwave radiation scheme accounts for water vapor, clouds, sulfate aerosols, dust and ozone. The total shortwave

radiation range is divided into two subintervals: ultraviolet+visible and near-infrared. The cloud albedo is a function of solar zenith angle and the optical thickness of the clouds (Feigelson et al., 1975). The optical properties of dust are treated following the Yamamoto and Tanaka (1972) scheme. The integral transmission function of ozone is taken from Lacis and Hansen (1974). In the near-infrared band the absorption due to water vapor is described according to Feigelson et al. (1975). The effect of sulfate aerosols is included following the approach of Bauer et al. (2008). The scheme accounts for both the direct radiative

forcing and the indirect effect through changes in cloud optical thickness. For simplicity, the radiative effect of black carbon and organic carbon aerosols is not represented in the model as it is implicitly assumed that their combined net effect is negligible to a first approximation. See Appendix A7 for more details on shortwave radiation.

To compute the longwave radiation fluxes, the atmosphere column is subdivided into 15 unevenly spaced levels. The long-wave radiative scheme accounts explicitly for water vapor, carbon-dioxide and ozone through their integral transmission func-

tions. The longwave radiative effect of the well mixed greenhouse gases $\mathrm{CH}_4$, $\mathrm{N}_2\mathrm{O}$ and $\mathrm{CFCs}$ is accounted for through the use of an equivalent $\mathrm{CO}_2$ concentration in the radiative scheme, using the respective radiative forcing following Etminan et al. (2016). Since CLIMBER-X does not include an atmospheric chemistry module, the spatial distribution of sulfate aerosol load and the 3D field of ozone concentration must be prescribed as input to the model. Both can be time-dependent. See Appendix A8 for more details on longwave radiation.

SESAM includes a single effective cloud layer. In this respect it is therefore even simpler than CLIMBER-2, which separately treats stratiform and convective clouds. Cloud fraction is a function of atmospheric relative humidity, effective vertical velocity and near-surface temperature inversion strength. Cloud top height is related to the height of the troposphere, modified by a factor depending on the vertical velocity at the cloud base. Cloud optical thickness is a function of surface air temperature, cloud fraction, column water content and sulfate aerosol load. See Appendix A6 for more details on cloud parameterisations.

A simple representation of the global dust cycle is included in SESAM based on a single representative dust particles diameter (Bauer and Ganopolski, 2010). Dust emissions are computed in the land model and are an input to the atmosphere,





where the dust concentration is assumed to decrease exponentially with height with a fixed height scale of $2\,\mathrm{km}$. Dust is transported by the 3D wind field and by macro-turbulent diffusion. Dust sinks include both dry and wet deposition.

## 2.2 Ocean model: GOLDSTEIN

The ocean component of CLIMBER-X is based on GOLDSTEIN (Edwards et al., 1998; Edwards and Shepherd, 2002; Edwards and Marsh, 2005). GOLDSTEIN is a 3D frictional-geostrophic balance model, and is also employed in several other existing EMICs, e.g. GENIE (Marsh et al., 2011) and Bern3D (Müller et al., 2006). In the original GOLDSTEIN implementation, the equations were coded in terms of non-dimensional quantities. For the inclusion into CLIMBER-X the code has been dimensionalized for better readability and more efficient debugging.

The horizontal velocity in the ocean is diagnosed from a frictional-geostrophic balance and the vertical velocity is then derived from the continuity equation. The model assumes hydrostatic balance. The velocity relaxation to the velocities of the previous time step employed in GENIE and Bern3D has been removed in CLIMBER-X, without drawbacks for model stability. Islands for barotropic flow are automatically derived from the topography, but the actual islands around which barotropic flow is permitted can be controlled by namelist settings. By default, non-zero barotropic flow is allowed through the Drake Passage

and through the Indonesian archipelago. The Bering and Davis Straits are closed for barotropic flow, but baroclinic tracer exchange is permitted. The friction for baroclinic velocities is set to be $4\,\mathrm{days}^{-1}$ and is globally uniform, while the friction for barotropic velocities is increased by a factor three near continental boundaries and shallow topographic features. Close to the equator the minimum absolute value of the Coriolis parameter is set to $5 \times 10^{-6}\,\mathrm{s}^{-1}$. In contrast to other models using GOLDSTEIN (e.g. Holden et al., 2016; Marsh et al., 2011; Müller et al., 2006) in CLIMBER-X the wind stress as simulated

by the atmospheric model is applied at the ocean surface, without any enhancement factor. Seawater density is computed using the UNESCO equation of state of Millero and Poisson (1981).

The principal limitations to the ocean dynamics follow from reduced resolution and the neglect of nonlinear momentum advection. The result is that there are no ocean eddies, and hence no mechanism for internally generated variability, and that boundary currents are unrealistically broad. One notable consequence is strong damping of the Antarctic Circumpolar Current

and the lack of a Gulf Stream extension propagating to high latitudes.

The tracer transport is described by an advection-diffusion equation, written in flux form. To reduce numerical diffusion associated with the original advection scheme we have implemented a flux-corrected transport scheme following Zalesak (1979). The model uses an explicit iso- and diapycnal diffusion scheme in its small-angle approximation (Redi, 1982) combined with a diffusive parameterization of the eddy-induced transport following the Gent–McWilliams skew flux (Gent and Mcwilliams,

1990; Griffies, 1998). The coefficients for both eddy-induced transport and isopycnal mixing are assumed to be equal by default ($1500\,\mathrm{m}^2\mathrm{s}^{-1}$). Slopes of the isoneutral surfaces above $1 \times 10^{-3}$ are tapered following Gerdes et al. (1991).The diapycnal diffusivity profile follows Bryan and Lewis (1979) with a minimum diffusivity of $0.1\,\mathrm{cm}^2\mathrm{s}^{-1}$ at the surface and a maximum of $1.5\,\mathrm{cm}^2\mathrm{s}^{-1}$ at the ocean floor. If the stratification of a water column is statically unstable, convective adjustment is applied using the scheme of Rahmstorf (1993). A simple mixed layer scheme, based on Kraus and Turner (1967) is used in the model.

Heat and freshwater fluxes at the ocean surface provide the top boundary conditions for the temperature and salinity prognostic





equations. Since GOLDSTEIN is based on the rigid-lid approximation, the freshwater flux is implemented in terms of a virtual salinity flux, computed using local salinity. A global correction is employed to make sure that the global annual net freshwater flux into the ocean is zero, in the absence of changes in land-ice volume. This is not sufficient to ensure conservation of salinity in the ocean, because of the anti-correlation between surface freshwater flux and surface salinity (e.g. Yin et al., 2010). Hence,

the virtual salinity flux is additionally corrected to ensure that the annual mean global net surface flux is zero. The geothermal heat flux of Lucazeau (2019) is applied as bottom ocean boundary condition.

The model is discretized on a regular lat-lon grid with $5° × 5°$ resolution, matching the atmosphere model resolution, and 23 unequally spaced vertical layers, with a $10\,\mathrm{m}$ top layer and layer thickness increasing with depth and reaching $500\,\mathrm{m}$ at the ocean bottom. Ocean bathymetry is smoothed by convolving bathymetry with a 4 directly neighbouring grid points kernel, to

remove strong gradients that would lead to numerical instabilities. To improve model stability, an additional limitation on the topographic slope entering in the Coriolis term (planetary vorticity advection) of the barotropic velocity equation is applied.

Since the land-sea mask is fully flexible in CLIMBER-X, GOLDSTEIN has been adapted to deal with newly forming and disappearing oceanic grid cells. It is always ensured that all grid cells of the ocean domain are connected. If 'ocean lakes' are formed at any point in time, the isolated ocean cells are removed from the ocean model domain and are treated as lake

in the land model instead. A minimum ocean fraction of 0.1 in a $5° × 5°$ grid cell is required for a cell to be considered part of the ocean domain. Newly formed cells are initialized using information from neighbouring grid cells. Changes in sea level, and therefore ocean volume, are additionally accounted for by scaling the thicknesses of the ocean layers below a depth of $1000\,\mathrm{m}$ to match the actual ocean volume derived from the high-resolution topograhy and provided as input to the ocean model. Total tracer inventories in the ocean are conserved in this process. Islands for barotropic flow are automatically updated

with the mask. Velocities are diagnostic quantities in the model, and therefore no intervention on velocity fields is required when updating the land-sea mask.

GOLDSTEIN is a rigid lid model and thus does not provide information on the elevation of the free surface. However, because sea surface height is needed in the sea ice momentum equation, as a first approximation it is diagnosed from the density above a reference depth of $1500\,\mathrm{m}$.

Additional tracers, including CFCs, dye and age tracers have been added to the model as useful diagnostics. The air-sea fluxes of CFCs are computed following the OMIP protocol (Orr et al., 2017).

Additional details of the ocean model are provided in Appendix B.

## 2.3 Sea ice model: SISIM

SISIM (SImple Sea Ice Model) is a dynamic-thermodynamic sea ice model, with a single ice layer and a snow layer on top.

Sea ice thermodynamics is based on the zero-layer model of Semtner (1976). It involves the determination of accumulation and melting of snow/sea ice from above and accretion and melting of sea ice from below.

The surface energy balance equation is solved separately for the ice-free and the ice-covered areas of each oceanic grid cell. Over sea ice, it is solved implicitly for the skin temperature, which is the temperature that balances all energy fluxes at the surface. The fluxes to be balanced include the net shortwave and longwave radiation at the surface, the sensible and latent





heat fluxes to the atmosphere and the conductive heat flux into the snow/ice. Whenever the skin temperature over the snow/sea ice layer is above the melting point, the skin temperature is reset to $0\,°C$ and the excess energy is used to melt snow/sea ice. Snowfall increases snow thickness and if the snow depth exceeds $1\,m$ the snow excess is converted to sea ice. The net energy at the base of the ice layer is determined by the conductive heat flux through the snow/ice layer and the turbulent heat flux between the ice and the seawater below. Whether accretion or ablation occurs depends on the sign of the net energy at the

ice/water interface. Over ocean water, the heat flux into the ocean is computed as the residual of the radiation and surface energy fluxes, whereby the surface energy fluxes are computed using sea surface temperature. Sea ice forms whenever the top layer ocean temperature drops below the freezing point of seawater.

The surface sensible and latent heat fluxes are proportional to the temperature and moisture gradients between the surface and near-surface air with the exchange coefficients depending on atmospheric stability through a bulk Richardson number.

Sea ice albedo is temperature dependent, with a decrease in albedo as the sea–ice skin temperature approaches the melting point. This represents the decrease in albedo following the formation of meltwater ponds on the ice surface. The snow albedo scheme is the same as used in the land model and includes a dependence on snow grain size and dust and soot concentration following Dang et al. (2015). The conductive heat flux within the sea ice/snow layer is assumed to be directly proportional to the temperature difference between the surface skin and the salinity-dependent freezing temperature at the base of the ice layer

(Semtner, 1976). The sea ice heat conductivity is scaled to represent sub-grid thickness distribution following Fichefet and Maqueda (1997). The turbulent heat flux at the ice base is determined from the difference between salinity dependent freezing point temperature and top layer ocean temperature using a constant exchange coefficient following McPhee (1992) and Weaver et al. (2001).

Sea ice drift velocities are computed from the momentum balance equation (e.g. Hibler, 1979) with the elastic-viscous-plastic

rheology of Hunke and Dukowicz (1997) and Bouillon et al. (2009). Numerically the solution of the momentum equation follows the implementation in the GFDL model SIS2 (Adcroft et al., 2019; Delworth et al., 2006), discretized on the Arakawa C grid to match the ocean model grid. The derived velocities are then used to advect the grid-cell mean ice and snow thicknesses and the sea ice concentration using a flux-corrected transport scheme (Zalesak, 1979). No explicit diffusion is applied.

Sea ice is allowed to cover the ocean fraction of the grid cell that is not occupyied by shelf ice. That is, floating shelf ice

restricts the domain available to sea ice.

Additional details of the sea ice model are provided in Appendix C.

## 2.4 Land model: PALADYN

CLIMBER-X includes the land model PALADYN (Willeit and Ganopolski, 2016), which serves as land surface scheme for the exchange of energy and water between the surface and the atmosphere but also represents vegetation dynamics and, more

generally, the land carbon cycle processes. PALADYN explicity resolves thermodynamic and hydrologic processes over several soil layers. Modifications to the model relative to the description in Willeit and Ganopolski (2016) are described next.





PALADYN has been complemented with a parameterisation of dust emissions following the CLIMBER-2 scheme as described in Bauer and Ganopolski (2010). Dust is emitted to the atmosphere from deserts and grasslands, whenever the soil is dry enough and the wind speed exceeds a critical threshold.

The snow albedo scheme has been refined with the inclusion of the effect of dust and soot on snow albedo following Dang et al. (2015) and the snow grain size parameterisation has been retuned using output of the regional climate model MARv3.6 simulations for Greenland (Fettweis et al., 2017).

CLIMBER-X, and therefore also PALADYN, does not resolve the diurnal cycle. However, diurnal variations can be important, particularly for highly non-linear processes, such as snowmelt. For instance, even if the daily mean surface air temperature

is below freezing, the daily maximum temperature could still be substantially above the freezing point, thus leading to snow melt during the day. In the presence of a relatively thin snow layer it is likely that the snow melted during the day would not completely refreeze during the night, but would infiltrate into the soil or contribute to surface runoff. Following this line of thought, a parameterisation of the diurnal cycle for snow melt, partly following Krapp et al. (2017), has been implemented to accelerate the spring-time retreat of snow-covered area, which was too slow in the original PALADYN formulation (Willeit and

Ganopolski, 2016). The diurnal cycle of skin temperature is assumed to have a sinusoidal shape with the amplitude depending on the difference between daily mean and daily minimum net shortwave radiation at the surface. The latter is computed from the diurnal cycle of insolation at the top of the atmosphere using the daily-mean surface albedo. If the resulting daily maximum skin temperature is above the freezing point, the energy flux available for snow melt is computed after Krapp et al. (2017).

The possibility to prescribe land use changes has also been introduced into PALADYN, following Burton et al. (2019). A

fraction of each grid cell is prescribed as being used for agriculture, with no distiction between cropland and pasture being made. Land use is then represented as a limitation to the space available for a plant functional type (PFT) to expand into. For instance, the three woody PFTs (broadleaf trees, needle-leaf trees and shrubs) are prevented from growing in the agricultural fraction, while the two grass PFTs (C3 grass and C4 grass) are allowed to grow anywhere in the grid cell and are simply interpreted as agricultural grasses if they grow into the pasture or cropland grid cell fraction. The representation of the effect of

land use change on land carbon fluxes will be discussed in more detail in the CLIMBER-X companion paper about the global carbon cycle.

Since the original PALADYN publication, lakes have been introduced as an additional surface type in the model. Lake fractions are prescribed. Lake thermodynamics follows the implementation in CLM4.5 (Oleson et al., 2013; Subin et al., 2012) and includes freezing and melting of lake water and a single snow layer on top of lake ice. Temperatures and ice fractions are

simulated for seven vertical layers, with a top layer thickness of $1\,\mathrm{m}$ and the bottom layer thickness depending on the depth of the lake, which is determined from lake bathymetry. For unfrozen lakes, convective mixing occurs in the lake if the density stratification becomes unstable. Additional vertical mixing occurs when partly frozen layers exist below not fully frozen layers in order to maintain ice contiguous at the top of the lake. It is additionally always ensured that the lake is mixed to a minimum depth of $20\,\mathrm{m}$. The surface energy fluxes over lakes are computed similarly to the other surface types as described in Willeit

and Ganopolski (2016).





## 3 Model evaluation for historical period and present day

The historical period, with its extensive availability of climate data from direct observations, forms the basis for the evaluation of any climate model. Here we evaluate the performance of CLIMBER-X for the climatological period from 1981 to 2010 and for the period of time from 1850 to 2015, corresponding to the historical period covered by CMIP (Coupled Model Intercom-

parison Project) simulations. For that we compare the results of CLIMBER-X simulations with different observation-based datasets as well as atmosphere and ocean reanalysis data. To give an overview of how CLIMBER-X compares to state-of-the-art general circulation models, we also include results from model simulations from the recent coupled model intercomparison projects. CMIP5 (Taylor et al., 2012) and CMIP6 (Eyring et al., 2016) model data are used interchangeably according to data availability.

The forcings for the historical CLIMBER-X simulations include variations in solar radiation (Matthes et al., 2017), radiative forcing of volcanic eruptions (Prather et al., 2013), globally uniform $CO_2$, $CH_4$ and $N_2O$ concentrations from Köhler et al. (2017), globally uniform CFC11 and CFC12 concentrations from Meinshausen et al. (2016), 3D $O_3$ concentrations and 2D $SO_4$ load from the ensemble mean of CMIP6 models and land use change (pasture and cropland fractions) from Ma et al. (2020). The model is initialized from a 5000–year equilibrium simulation with pre-industrial boundary conditions.

### 325 3.1 1981–2010 climatology

In the following, different simulated climatological characteristics are compared to observations to asses the model performance for present day. Unless stated otherwise the comparison with observations is for the time interval from 1981 to 2010.

The global mean near-surface air temperature averaged over the time period 1981–2010 in CLIMBER-X is 14.1 °C, which compares well to the 14.35 °C in ERA-Interim reanalysis and is in the middle of the wide range of roughly 13–15 °C spanned

by CMIP5/6 models (e.g. Bock et al., 2020). Global temperature is largely determined by the global radiation and energy budget, whose components are in good agreement with observations and CMIP5 models, as shown in detail in Tab. 1. The strength of the hydrological cycle is a bit overestimated in the model, resulting in higher evaporation and precipitation than observed, particulary over the ocean (Tab. 2).

Atmosphere and ocean dynamics play an important role in the climate system by transferring energy from low to high

latitudes, thereby reducing the equator-to-pole temperature gradient arising from differential solar heating. The total meridional energy transport simulated by CLIMBER-X is compared to observation-based estimates in Fig. 2a. Overall the agreement is good, with a slight underestimate of the peak poleward energy transport. The meridional energy transport by the atmosphere, which constitutes the dominant contribution to total energy transport, matches well with observations (Fig. 2b). A separation of the different atmospheric processes contributing to latitudinal energy transport is also shown in Fig. 2b, including the partition

between dry static energy and latent heat fluxes and the contribution by eddies. Eddies dominate the meridional energy transport at mid- to high latitudes, while the Hadley cells are the most important contribution in the tropics, in agreement with a similar decomposition performed with other models (e.g. Yang et al., 2015).





**Table 1.** Earth's radiation and energy budget in CLIMBER-X, compared to Observations and CMIP5 models from Wild et al. (2013). Units are $\text{Wm}^{-2}$.

|  | CLIMBER-X | Observations mean (min,max) | CMIP5 median (min,max) |
|---|---|---|---|
| *TOA components* |  |  |  |
| Solar down | 340.2 | 340 (340,341) | 341.6 (338.9,341.6) |
| Solar up | 103.3 | 100 (96,100) | 102.8 (96.3,107.8) |
| Solar net | 236.9 | 240 | 239.5 (233.8,244.7) |
| Thermal up | 236.5 | 239 (236,242) | 238.5 (232.4,243.4) |
| *Atmospheric components* |  |  |  |
| Solar net | 72.2 | 79 (74,91) | 74.0 (69.7,79.1) |
| Thermal net | -175.8 | -184 | -179.4 (-171.9,-194.0) |
| *Surface components* |  |  |  |
| Solar down | 190.6 | 185 (179,189) | 189.1 (181.9,197.4) |
| Solar up | 25.9 | 24 (22,26) | 24.2 (20.9,31.5) |
| Solar net | 164.7 | 161 (154,166) | 164.8 (159.6,170.1) |
| Thermal down | 337.4 | 342 (338,348) | 338.2 (327.7,347.5) |
| Thermal up | 398.1 | 397 (394,400) | 397.3 (392.6,403.7) |
| Thermal net | -60.7 | -55 | -58.4 (-65.2,-49.4) |
| Net radiation | 104.0 | 106 | 105.4 (100.3,116.6) |
| Latent heat | 81.9 | 85 (80,90) | 85.8 (78.8,92.9) |
| Sensible heat | 21.0 | 20 (15,25) | 18.7 (14.5,27.7) |

**Table 2.** Global hydrological cycle in CLIMBER-X, compared to Observations (Trenberth et al., 2007). Units are $10^{15}\text{kgyr}^{-1}$.

|  | CLIMBER-X | Observations |
|---|---|---|
| Precipitation | 527 | 486 |
| Precipitation land | 120 | 113 |
| Precipitation ocean | 407 | 373 |
| Evaporation | 527 | 486 |
| Evaporation land | 77 | 73 |
| Evaporation ocean | 450 | 413 |
| Runoff | 40 | 40 |



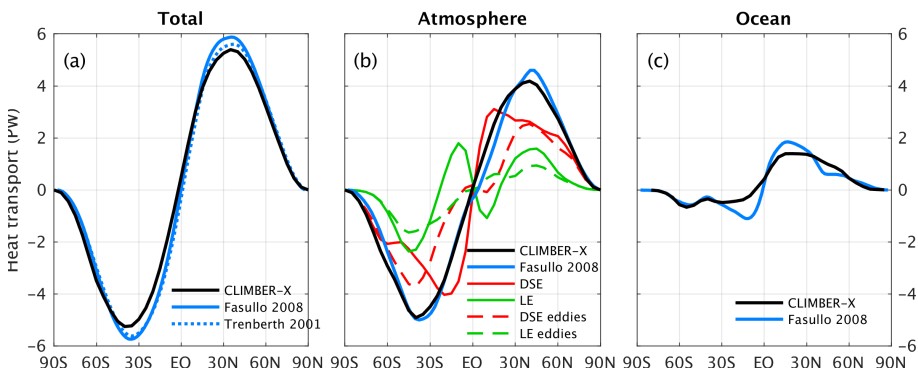

**Figure 2.** Meridional energy transport. (a) Total simulated meridional energy transport by ocean and atmosphere compared to observations (Trenberth and Caron, 2001; Fasullo and Trenberth, 2008). (b) Meridional energy transport by the atmosphere compared to estimates from Fasullo and Trenberth (2008) and modelled decomposition into dry static energy (DSE) and latent heat (LE) fluxes and contribution by eddies. (c) Global ocean meridional energy transport compared to observations (Fasullo and Trenberth, 2008).

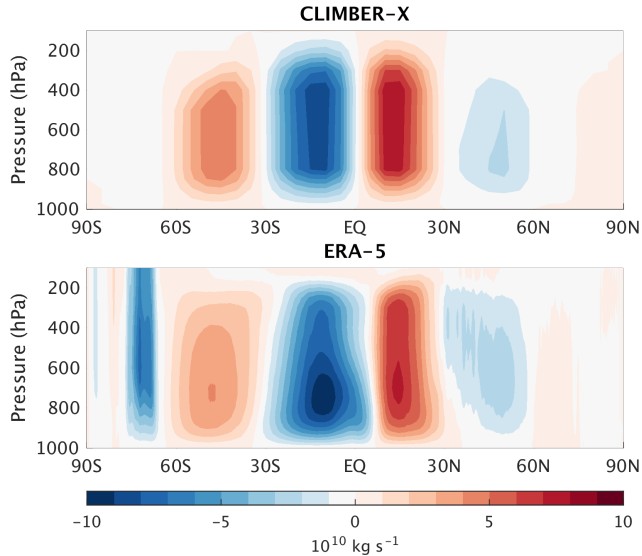

**Figure 3.** Annual mean simulated meridional atmospheric streamfunction compared to ERA-5 reanalysis (Hersbach et al., 2020).

The mean meridional circulation in the atmosphere is characterized by the presence of 6 cells and is a defining feature of atmospheric dynamics. Figure 3 shows that the parameterization of the meridional circulation employed in CLIMBER-X does a reasonable job at describing the Hadley cells in the Tropics and the Ferrel cells at mid-latitudes.

The zonal mean sea level pressure, which is directly related to the mean meridional circulation, shows good agreement with reanalysis data, both for December–February (DJF) and for June–August (JJA) (Fig. 4c,f). The model also reproduces the main



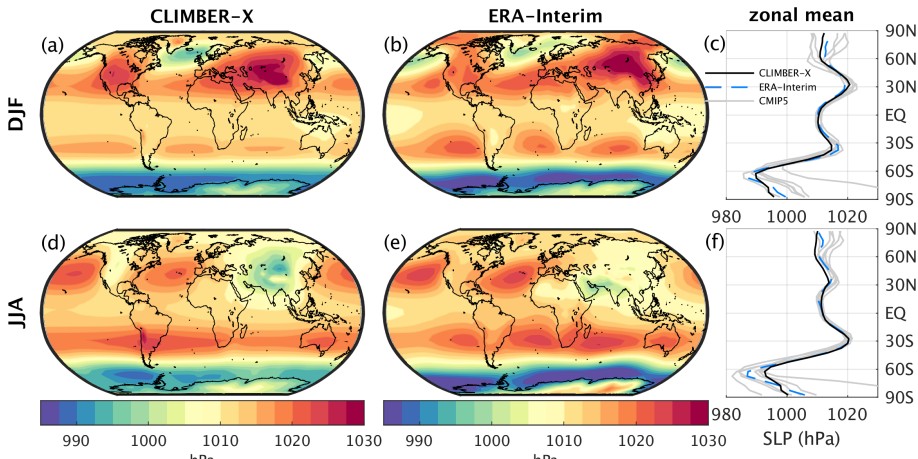

**Figure 4.** Sea level pressure in CLIMBER-X for (a) DJF and (d) JJA compared to ERA-Interim reanalysis (Dee et al., 2011) (b,e). The zonal mean is additionally compared to CMIP5 models in (c) and (f).

features of observed azonal sea level pressure, such as the highs over land and lows over ocean in Northern Hemisphere winter (Fig. 4a,b), and vice versa during summer (Fig. 4d,e).

The simulated near-surface air temperature is compared to reanalysis in Fig. 5. In the zonal mean, the simulated temperatures fall mostly within the range given by different CMIP6 models (Fig. 5c,g). The main model biases include too warm temperatures over eastern continents in NH winter, too cold tropics, particularly over land, and a warm bias over East Antarctica (Fig. 5b,f). The absolute model bias is generally comparable to the bias seen in CMIP6 models, except for the tropics (Fig. 5d,h). The cold bias in the Tropics is not only a surface feature, but is persistent throughout the Troposphere (Fig. 6). The

rest of the 3D temperature structure is well simulated by the model, with zonal mean temperature biases below a few degrees over large parts of the domain. During the winter months and at high latitudes, CLIMBER-X also captures the near-surface temperature inversions. The simulated tropopause height shows a realistic latitudinal profile, but is generally a few kilometers too low in the Tropics (Fig. 6). CLIMBER-X also reproduces the higher seasonal variations in temperature over the continents compared to the ocean, in particular at high latitudes (Fig. 7).

The simulated atmospheric relative humidity is generally high in the planetary boundary layer and shows pronounced minima in the subtropics, broadly in agreement with observations (Fig. 8). The model does a reasonably good job at reproducing the observed precipitation distribution (Fig. 9). In terms of zonal mean precipitation, the peak associated with the intertropical convergence zone (ITCZ), the minima in the subtropics and the maxima at mid-latitudes are well captured by the model (Fig. 9c,f). The main deficiencies are found in the subtropics, with too much precipitation simulated over the ocean in the

subsidence areas (Fig. 9a,b,d,e). This is partly related to the too weak subtropical high pressure systems (Fig. 4).

Cloud cover in CLIMBER-X shows the characteristic latitudinal profile, with minima in the subtropical subsidence areas and maxima in the tropics and at mid- to high latitudes (Fig. 10a,f). A realistic simulation of cloud cover is a pre-requisite for a good representation of radiation fluxes. Both shortwave and longwave radiation fluxes at the top of the atmosphere (TOA)

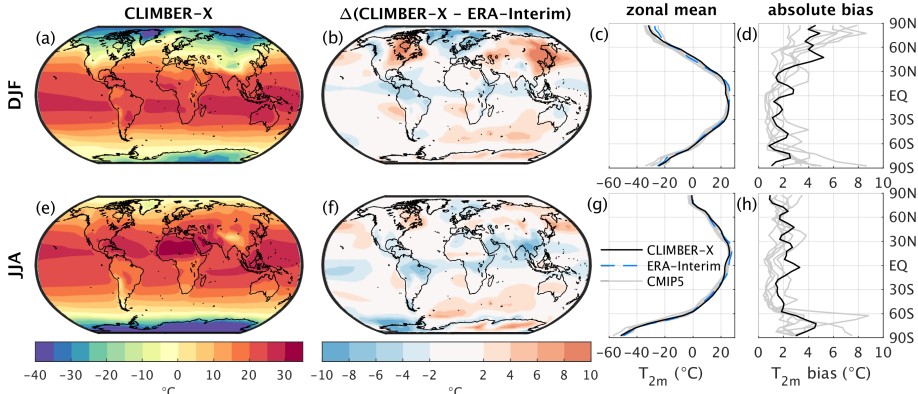

**Figure 5.** Near-surface air temperature in CLIMBER-X for (a) DJF and (e) JJA and model bias relative to ERA-Interim reanalysis (Dee et al., 2011) (b,f). The zonal mean is additionally compared to CMIP5 models in (c) and (g). The zonal mean absolute model bias in CLIMBER-X and a selection of CMIP5 models relative to ERA-Interim is shown in (d) and (h), for DJF and JJA, respectively.

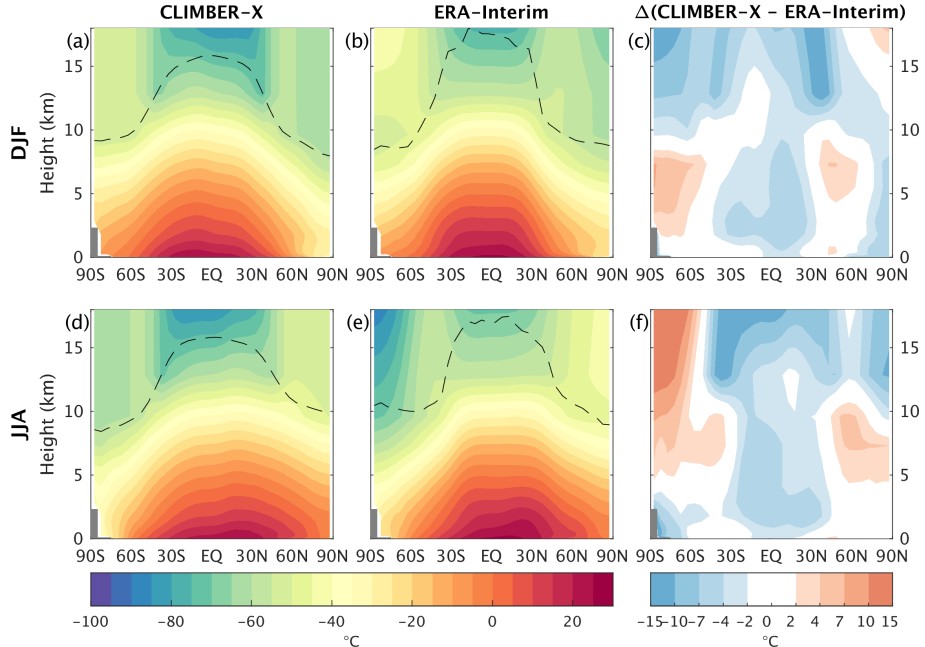

**Figure 6.** Zonal mean temperature simulated by CLIMBER-X for (a) DJF and (d) JJA compared to ERA-Interim reanalysis (Dee et al., 2011) (b,e). The temperature bias relative to ERA-Interim is shown in (c,f). The dashed black line indicates the height of the tropopause.

are in good agreement with satellite observations and reanalysis (Fig. 10b,d,g,i). The zonal mean radiative fluxes are generally
within the range of CMIP5 models (Fig. 10c,e,h,j). Net shortwave radiation at TOA is slightly overestimated at high latitudes
in NH summer (Fig. 10h), while net longwave TOA radiation exhibits some systematic biases in the tropics (Fig. 10e,j).

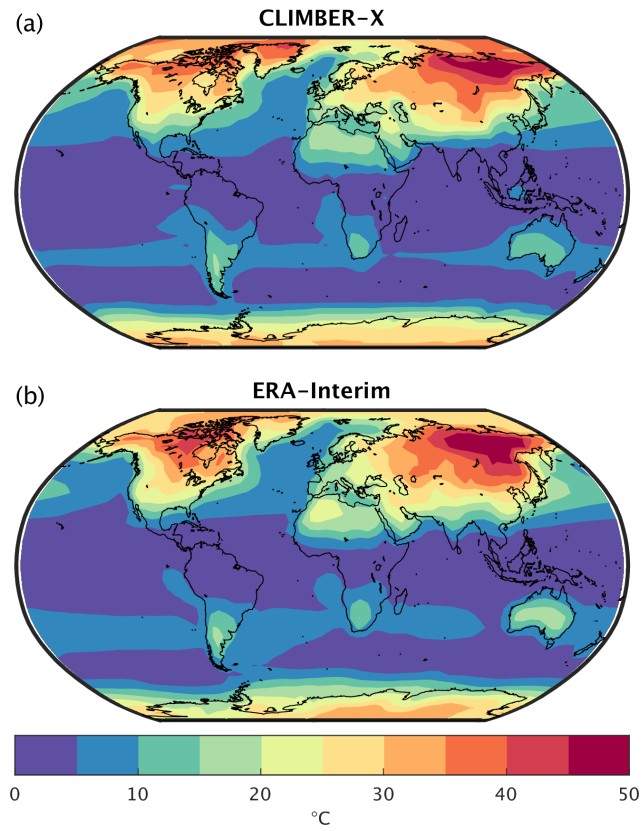

(a) CLIMBER–X

(b) ERA–Interim

**Figure 7.** Near-surface air temperature seasonality in (a) CLIMBER-X compared to (b) ERA-Interim reanalysis (Dee et al., 2011) computed as the difference between maximum and minimum monthly temperature.

The ocean overturning circulation in CLIMBER-X is characterized by the presence of an Atlantic overturning cell and by Antarctic bottom water formation (Fig. 11). The maximum of the Atlantic overturning streamfunction at 26°N is 18.5 Sv, a bit higher than indicated by observations (Frajka-Williams et al., 2021) (Fig. 12). The simulated Atlantic meridional circulation

(AMOC) penetration depth of ∼3800 m, as measured by the zero crossing in the streamfunction, is about 500 m too shallow compared with that directly observed, a problem common also to many CMIP6 models (Fig. 12). The maximum meridional heat transport by the Atlantic ocean is ∼1.1 PW. Deep water forms in the model at several locations in the northern North Atlantic, i.e. in the Labrador Sea south of Greenland and in the Nordic Seas, in agreement with observational estimates (Fig. 13). Deep water is also formed at several locations around Antarctica, in particular in the Ross and Weddell Seas, as shown by the annual

maximum mixed layer depth in Fig. 13. No deep water is formed in the North Pacific.

Ocean temperature and salinity fields compare well with observations in the deep ocean (Fig. 14, 15), with model biases mostly concentrated in the upper 1000 m. Biases in simulated temperature include a too warm Arctic Ocean, too cold intermediate waters in the sub-tropics in the Atlantic and the Indian ocean and too warm surface water in the North Pacific (Fig. 14c,f,i). Salinity biases of up to 1 psu are present in all ocean basins in the upper ∼1000 m (Fig. 15c,f,i).

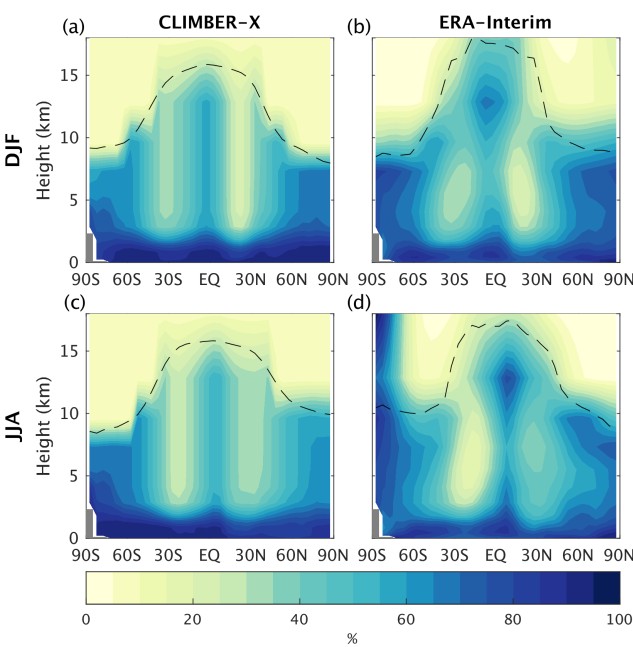

**Figure 8.** Zonal mean relative humidity simulated by CLIMBER-X for (a) DJF and (c) JJA compared to ERA-Interim reanalysis (Dee et al., 2011) (b,d). The dashed black line indicates the height of the tropopause.

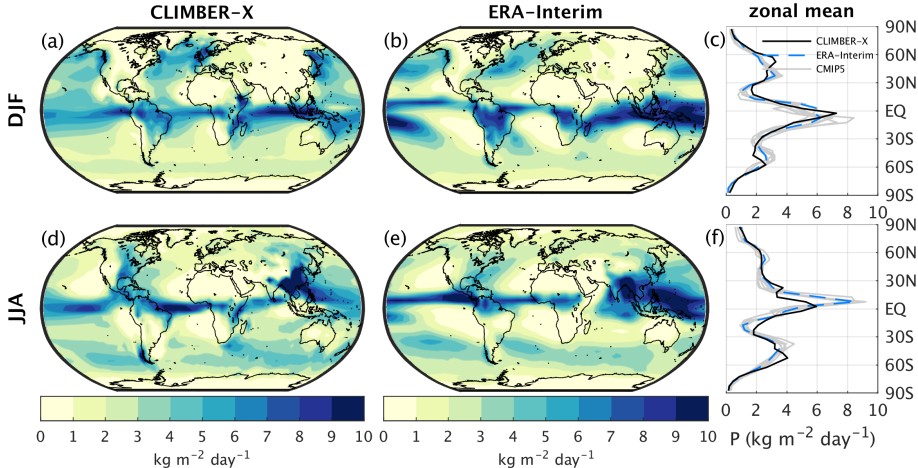

**Figure 9.** Precipitation modelled by CLIMBER-X for (a) DJF and (d) JJA compared to ERA-Interim reanalysis (Dee et al., 2011) (b,e). The zonal mean is additionally compared to CMIP5 models in (c) and (f).

The seasonality in sea–ice area in both the NH and SH is well reproduced by CLIMBER-X (Fig. 16), and is mostly within the range of CMIP6 models. The spatial extent of minimum and maximum the sea ice cover is also in generally good agreement with observations (Fig. 17). Arctic winter sea ice cover is overestimated in the Fram Strait and in the Barents sea, while it is





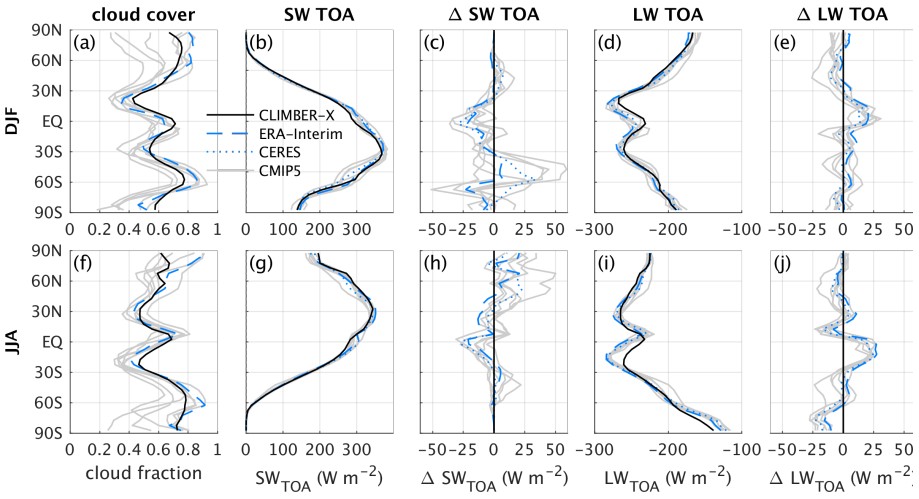

**Figure 10.** Simulated zonal mean cloud fraction and top-of-the-atmosphere net radiation fluxes for (top) DJF and (bottom) JJA, compared to satellite observations (Loeb et al., 2018), reanalysis (Dee et al., 2011) and CMIP5 models.

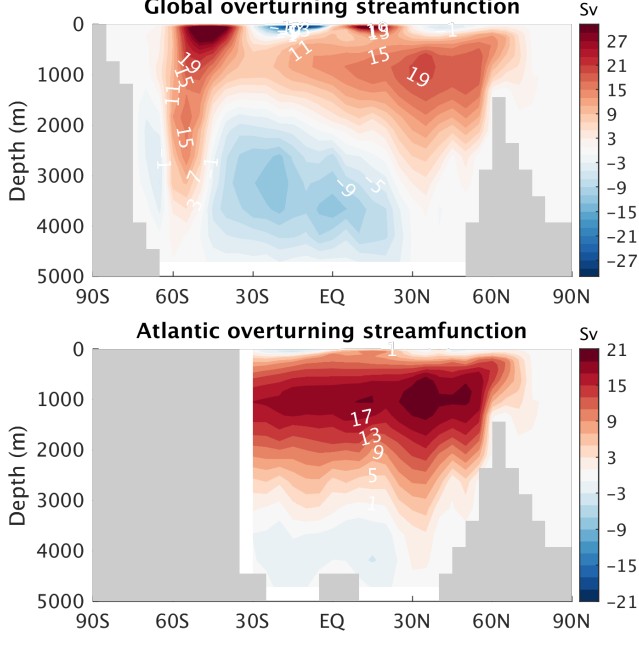

**Figure 11.** Simulated ocean overturning circulation: (top) global and (bottom) Atlantic.

underestimated in the Sea of Okhotsk (Fig. 17a,e). Minimum and maximum sea ice extent in the Southern Ocean are well represented in the model (Fig. 17b,d,f,h).

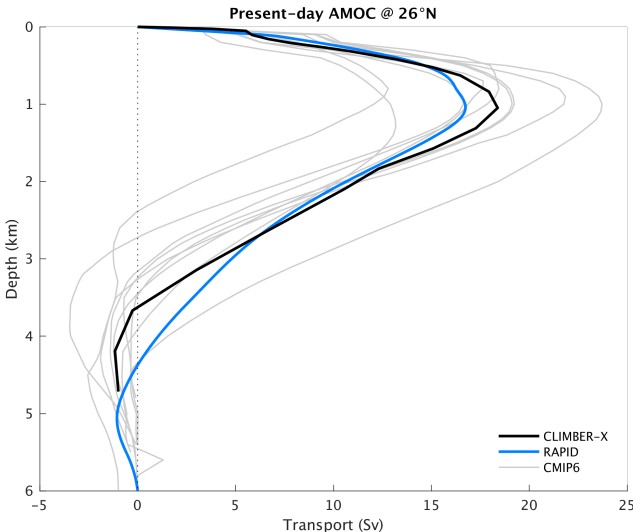

**Figure 12.** Vertical profile of the simulated Atlantic meridional overturning streamfunction at 26N (black) compared to observations from the RAPID array (Frajka-Williams et al., 2021) (blue) and a selection of CMIP6 models (grey). The CLIMBER-X and CMIP6 streamfunction is computed from historical simulations as the average over the time period from 2000 to 2014, while the RAPID values represent an average from 2004 to 2020.

The simulated total permafrost area is $18.5 \times 10^6 \mathrm{km}^2$, close the the observed value of $18.8 \times 10^6 \mathrm{km}^2$ (e.g. Tarnocai et al., 2009). In terms of spatial extent, permafrost area is in good agreement with observations over Eurasia, while it is underestimated in Eastern Canada (Fig. 18).

### 3.2 Simulations for the historical period

The CLIMBER-X simulated historical evolution of global mean temperature is compared to observations (Morice et al., 2012) and CMIP6 models in Fig. 19a. The model reproduces rather well the observed historical temperature trends and the response to volcanic eruptions. CLIMBER-X does not represent internal climate variability and its results therefore cannot be compared one-to-one with observations. However, the simulated temperature shows a very good match to the ensemble mean of CMIP6 models, where internal variability has effectively been removed. The contribution of the different forcings to the historical temperature evolution also show good agreement with the corresponding CMIP6 ensemble means, except for the last two decades when CMIP6 models tend to overestimate the observed temperature change (Fig. 19b-d).

The rate of heat uptake by the ocean is consistent with observations (Levitus et al., 2012) until around the year 2000, but is overestimated after that (Fig. 20). However, the historical ocean heat uptake in CLIMBER-X is in better agreement with observations compared to most EMICs, including CLIMBER-2 (Eby et al., 2013).



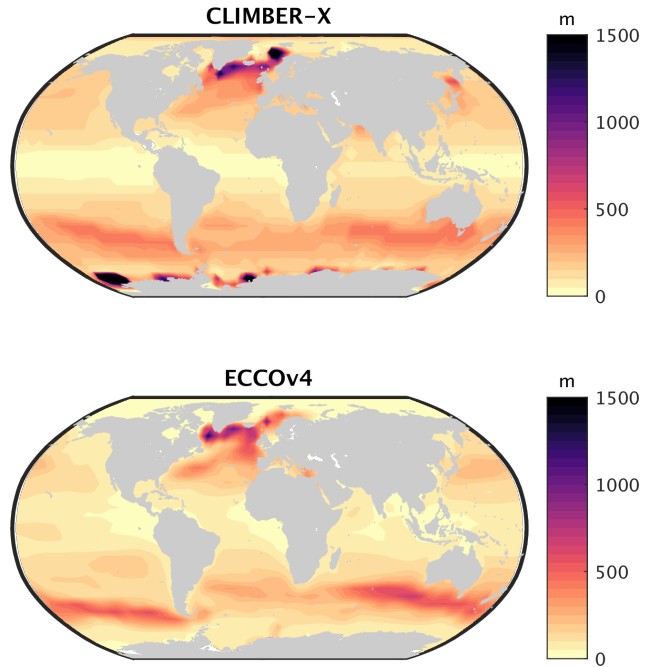

**Figure 13.** Maximum monthly mixed layer depth simulated by CLIMBER-X (a) compared with ECCOv4 reanalysis (ECCO et al., 2021) (b).



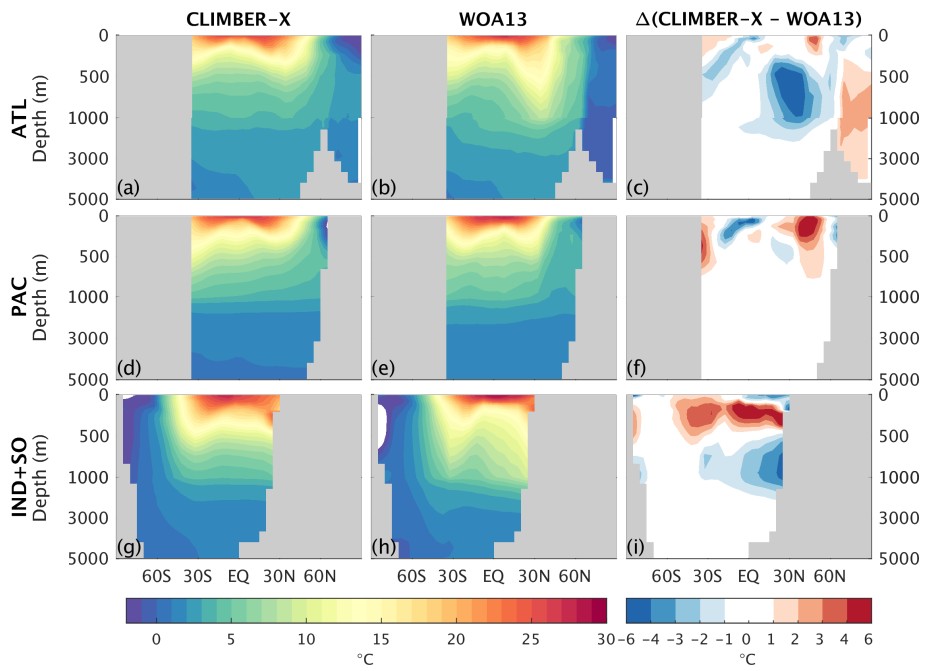

**Figure 14.** Zonal mean ocean temperature simulated by CLIMBER-X (left) compared with WOA13 data (Levitus et al., 2015) (middle) for the (top) Atlantic, (middle) Pacific and (bottom) Indian and Southern Ocean. The right panels show the model bias.



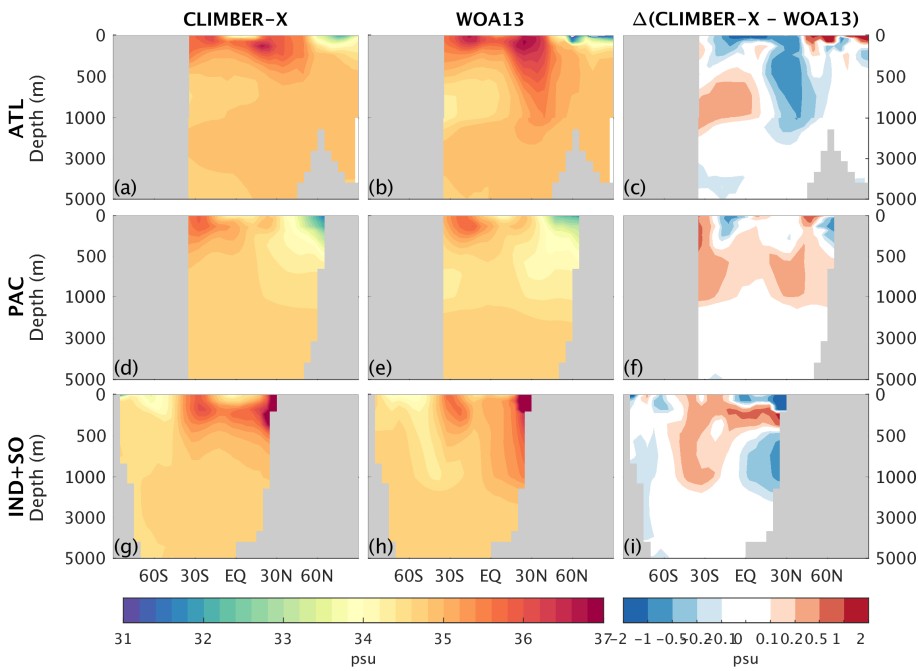

**Figure 15.** Zonal mean ocean salinity simulated by CLIMBER-X (left) compared with WOA13 data (Levitus et al., 2015) (middle) for the (top) Atlantic, (middle) Pacific and (bottom) Indian and Southern Ocean. The right panels show the model bias.

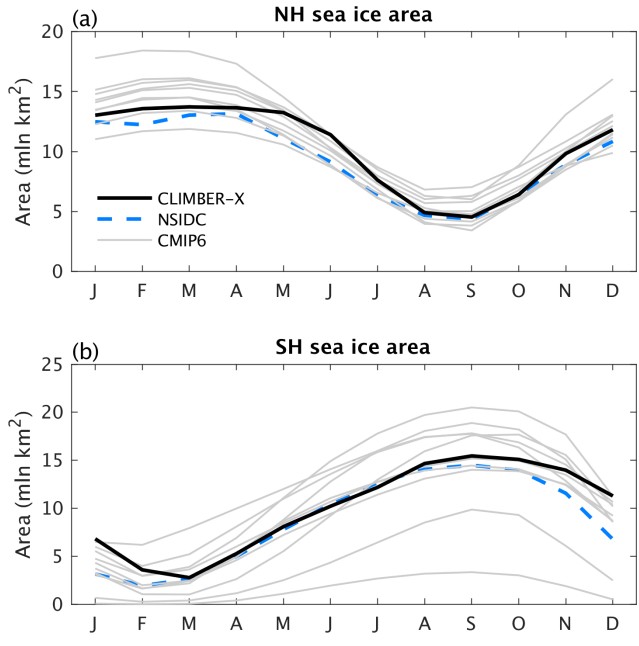

**Figure 16.** Seasonal variation of total sea ice area for (a) the NH and (b) the SH as simulated by CLIMBER-X (black) compared to observations (Meier et al., 2021) (blue) and CMIP5 models (grey).



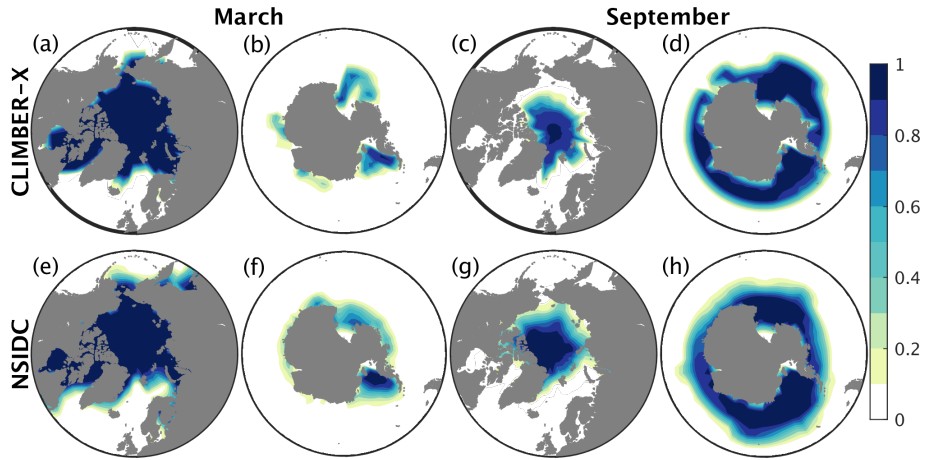

**Figure 17.** Sea ice concentration in the NH and SH in CLIMBER-X (top) compared to observations (Meier et al., 2021) (bottom) for (left) March and (right) September.

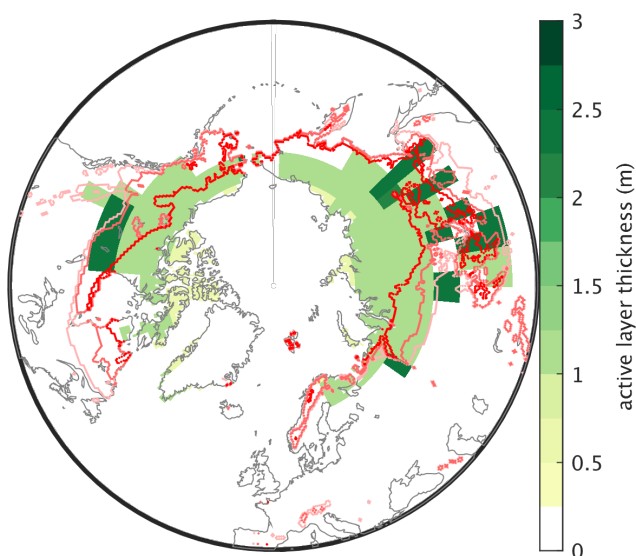

**Figure 18.** Modelled permafrost extent and active layer thickness compared to the observed extent of continuous, discontinuous and isolated permafrost (red lines, from dark red to light red) from Brown et al. (1998). The active layer thickness is calculated as the mean over the period 1981–2010 in grid cells that are permafrost during the whole time period.



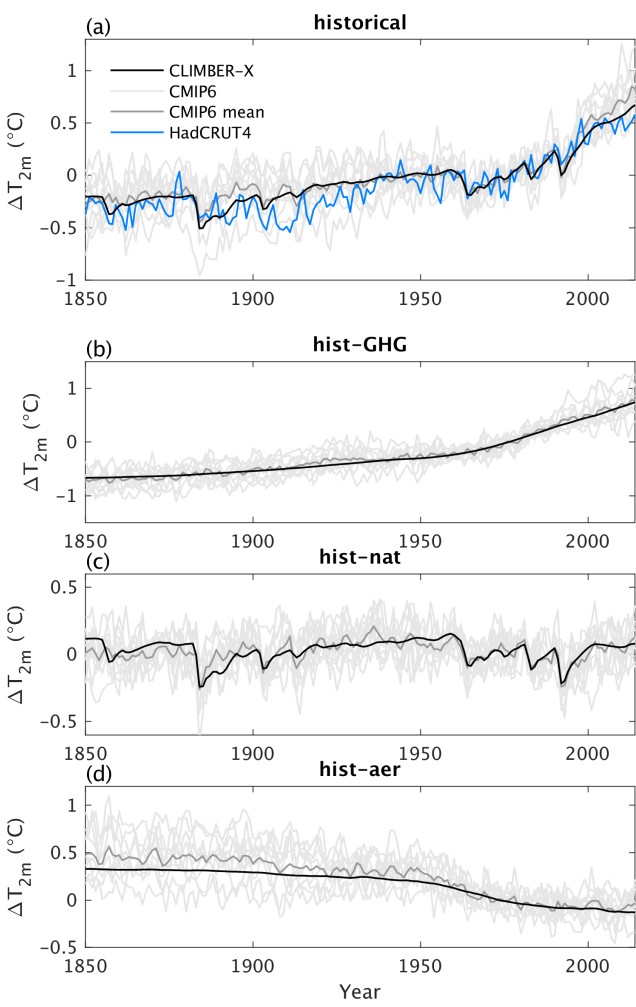

**Figure 19.** (a) Historical global mean near-surface air temperature simulated by CLIMBER-X (black) compared to observations (blue) (Morice et al., 2012) and CMIP6 models (grey). (b-d) Global mean near-surface air temperature in CLIMBER-X and CMIP6 models for idealized historical simulations with greenhouse gas concentration forcing only, natural (solar and volcanic) forcing only and aerosol forcing only, respectively.



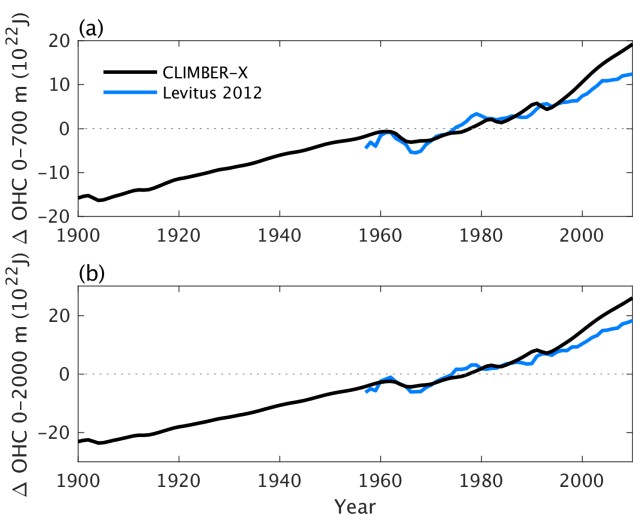

**Figure 20.** Historical ocean heat content anomalies in (a) top 700 m and (b) top 2000 m simulated by CLIMBER-X (black) and derived from observations (blue) (Levitus et al., 2012).





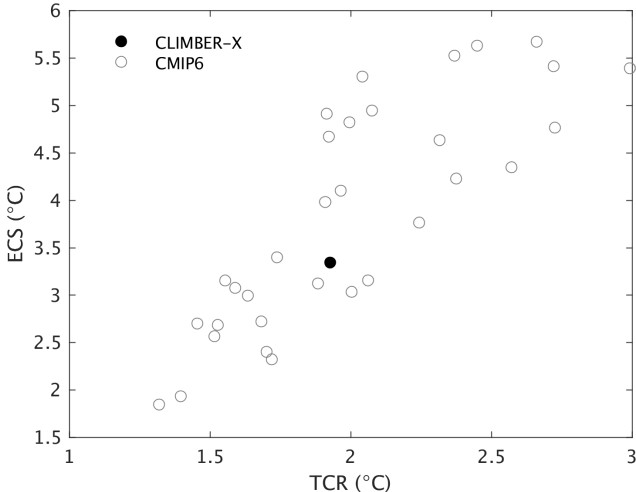

**Figure 21.** Equilibrium climate sensitivity versus transient climate response for CLIMBER-X and CMIP6 models. CMIP6 model data are from Nijsse et al. (2020).

## 4 Model sensitivities

Present-day observations provide a relatively poor constraint on model sensitivities to different climate forcings. Comparison with state-of-the-art general circulation models for experiments with different forcings and boundary conditions is therefore crucial for a model like CLIMBER-X. To this end we performed a comprehensive analysis of climate feedbacks, compared the response to changes in $CO_2$ for standard CMIP abrupt4xCO2 and 1%/year $CO_2$ increase experiments, evaluated the vegetation feedback and tested the response to last glacial maxiumum boundary conditions, which provides insights into the

model response to different orbital configuration, topography and land sea mask. We also performed standard freshwater hosing experiments to investigate the stability properties of the Atlantic meridional ocean circulation. An overview of the results of these experiments is presented next.

### 4.1 Transient climate response, climate sensitivity and Charney feedbacks

The equilibrium climate sensitivity of the standard version of CLIMBER-X is 3.3 K as computed from the temperature change

at equilibrium (5000 years-long experiment) for a doubling of atmospheric $CO_2$. This is in the middle of the range of 1.5–4.5 initially derived by Charney et al. (1979), which is also the estimated range given by the latest IPCC report (IPCC 2021, 2021). The transient climate response of CLIMBER-X, defined as the global temperature change at the time of $CO_2$ doubling in 1%/year $CO_2$ increase experiments, is 1.8 K. A plot of equilibrium climate sensitivity versus transient climate response shows that CLIMBER-X is also well within the range of CMIP6 models (Fig. 21).

CLIMBER-X includes code to diagnose the strength of the different climate feedbacks, which allows for a more detailed analysis of the processes controlling climate sensitivity in the model. The feedbacks are evaluated using the partial radiation



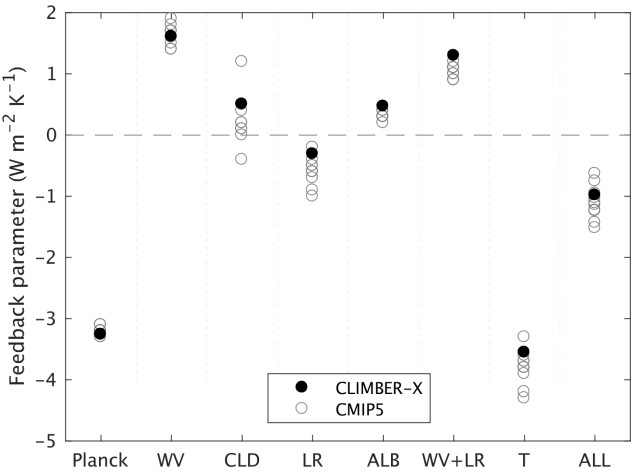

**Figure 22.** Global feedback parameters for CLIMBER-X and CMIP5 models. The feedbacks are (from left to right): Planck, water vapor, cloud, lapse rate, albedo, water vapor+lapse rate, temperature (Planck+lapse rate) and the sum of all feedbacks. CMIP5 data are from IPCC AR5.

perturbation method (Bony et al., 2006; Wetherald and Manabe, 1988). In this method, partial derivatives of model top of the atmosphere radiation with respect to changes in modelled fields (such as water vapour, lapse rate and clouds) are determined by diagnostically rerunning the model radiation code.

The global feedback parameters for CLIMBER-X computed for $CO_2$ doubling relative to 280 ppm are shown in Fig. 22 and generally compare well with feedback parameters computed for different models (e.g. Bony et al., 2006). The water vapor feedback is the largest postive feedback in the model, followed by cloud and albedo feedbacks. The lapse rate feedback is globally negative, in agreement with CMIP5 models (Fig. 22).

   A look at the zonal mean feedback parameters gives further insight into the spatial distribution of the feedbacks (Fig. 23).
The albedo feedback is large in high latitudes and is related to the sea-ice retreat and reduced snow cover in a warmer climate (Fig. 23e). The water vapor feedback is associated with an increase in water vapor content in a warmer atmosphere and is larger in the tropics than at the poles (Fig. 23f). The lapse rate feedback is negative in the tropics because of the larger warming of the mid-upper troposhere relative to the surface, while is it large and positive in high latitudes as a result of a pronounced erosion of surface temperature inversions mainly due to retreating sea ice. (Fig. 23g). These results are all in agreement with feedbacks
diagnosed in different general circulation models (e.g. Colman et al., 2001; Crook et al., 2011). Feedbacks related to clouds are the most uncertain and account for a large portion of the spread in climate sensitivity within current general circulation models (e.g. Zelinka et al., 2020). Clouds affect both longwave and shortwave radiation at the top of the atmosphere through different processes. Cloud feedbacks in CLIMBER-X are shown in Fig. 23b-d, including a separatation into contributions from changes in cloud fraction, cloud optical thickness and cloud height. The net cloud feedback is positive at all latitudes with the notable
exception of the Southern Ocean, where a pronounced increase in optical thickness with warming causes a large negative shortwave cloud feedback. Shortwave and longwave radiation cloud feedbacks generally act to (at least partly) compensate



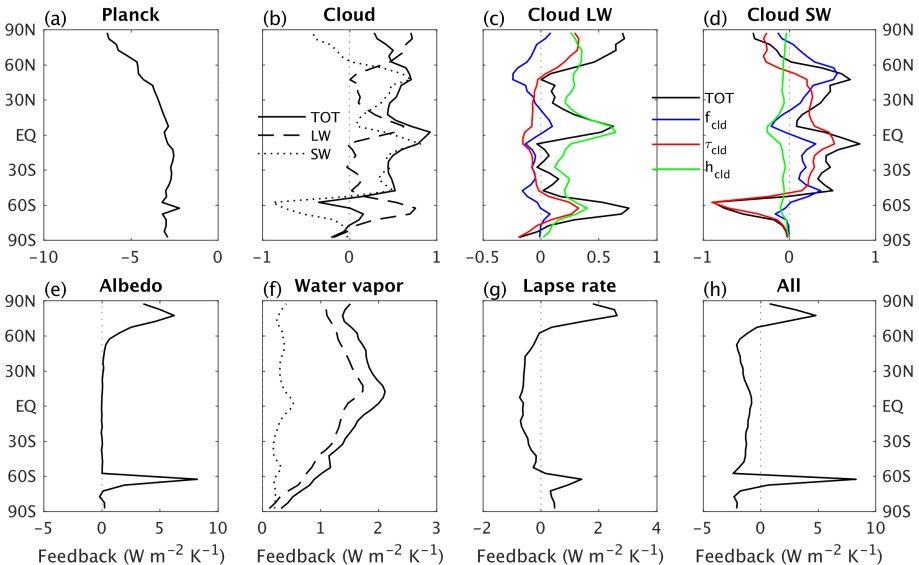

**Figure 23.** Zonal mean feedback parameters for CLIMBER-X. The total feedbacks (black solid lines) are further separated into contributions from longwave (LW, dashed black lines) and shortwave (SW, dotted black lines) radiation. Cloud feedbacks are additionally explicitly decomposed into LW and SW in (c) and (d), with the different colors representing feedbacks from changes in cloud fraction (blue), cloud optical thickness (red) and cloud top height (green).

in CLIMBER-X (Fig. 23c-d), in agreement with ESMs (e.g. Zelinka et al., 2012). The effect of cloud optical depth changes are generally larger in the shortwave, while changes in cloud height have a larger effect on longwave radiation. Combined, all feedbacks are mostly negative, except at high latitudes where sea ice melting leads to large positive albedo and lapse rate
feedbacks (Fig. 23h).

Hydrological sensitivity, which quantifies the relative global precipitation change per unit change in global temperature, is an important measure of the response of the hydrological cycle to climate change. In agreement with CMIP models, CLIMBER-X shows an increase in global precipitation by $\sim 2\,\%$ per degree global temperature increase in the $1\,\%$ per year $CO_2$ increase experiment (Fig. 24b), while the atmospheric water content increase approximately follows the Clausius-Clapeyron increase
of $7\,\%\,\mathrm{K}^{-1}$ (Fig. 24a).

## 4.2 Vegetation feedback

Changes in vegetation structure and its spatial distribution can affect the climate through the effect on surface energy and water fluxes (e.g Levis et al., 1999; Bala et al., 2006; Falloon et al., 2012). In CLIMBER-X, the vegetation feedback amplifies temperature change by more than $1\,^{\circ}\mathrm{C}$ in mid- to high northern latitudes for a reduction of $CO_2$ to $180\,\mathrm{ppm}$ (Fig. 25a), while it
is small for $CO_2$ doubling (Fig. 25b). A similar assymetry in the vegetation feedback between low and high $CO_2$ has also been found in CLIMBER-2 (Willeit et al., 2014). The reason for the strong positive vegetation feedback for low $CO_2$ originates from a pronounced southward retreat of boreal forest (Fig. 26b), which causes a substantial increase in surface albedo through





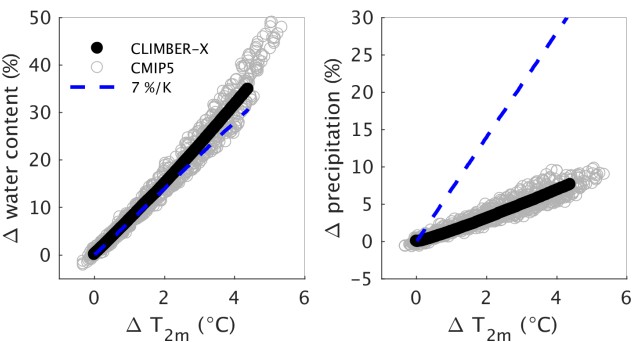

**Figure 24.** Relative global water content (a) and global precipitation (b) change versus global temperature change for CLIMBER-X and CMIP5 models from the 1%/year $CO_2$ increase experiment. Each circle represents one year of the simulations.

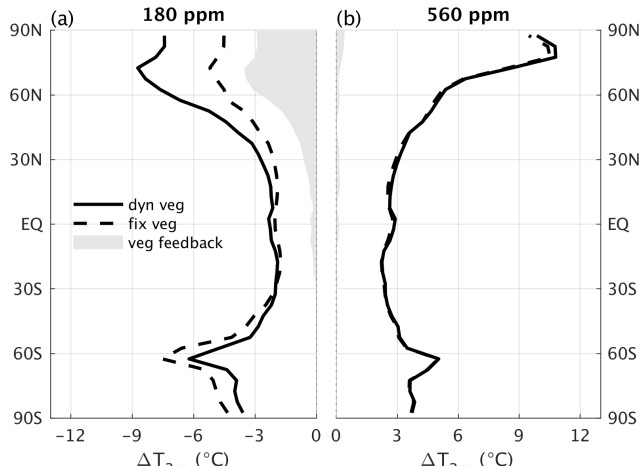

**Figure 25.** Zonal mean temperature response for atmospheric $CO_2$ concentrations of (a) 180 ppm and (b) 560 ppm relative to 280 ppm with (solid) and without (dashed) vegetation feedback.

the missing snow masking effect of trees (e.g. Bonan, 2008). In CLIMBER-X, a $CO_2$ increase leads in general to an expansion of forests and a reduction in desert area, while the opposite happens for lower $CO_2$ levels (Fig. 26).



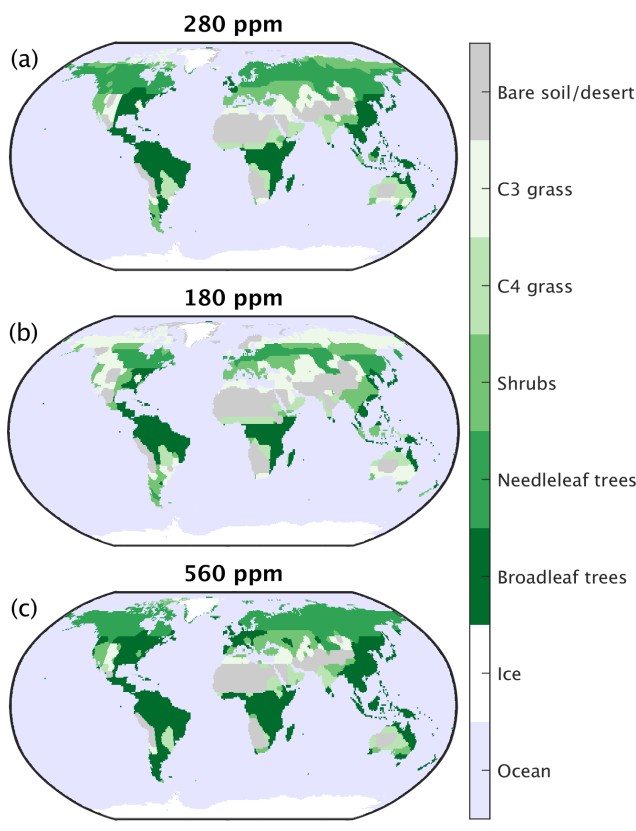

**Figure 26.** Simulated dominant plant functional types at equilibrium for (a) the preindustrial control, (b) 180 ppm and (c) 560 ppm of atmospheric $CO_2$.



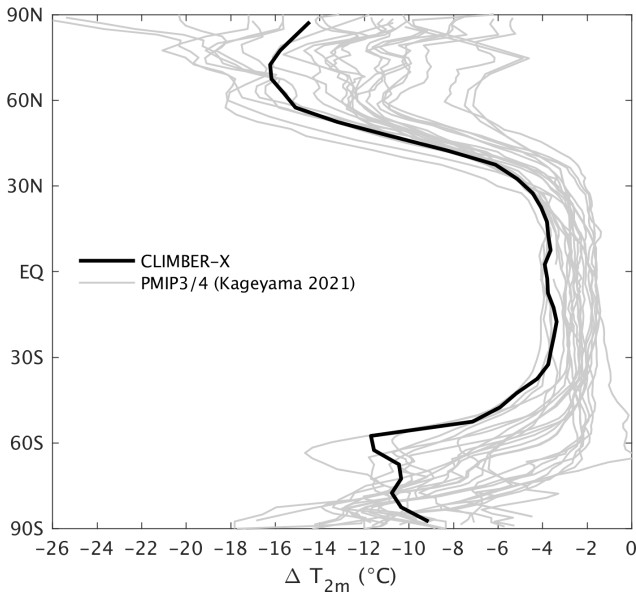

**Figure 27.** Last glacial maximum annual mean zonal near-surface air temperature differences relative to preindustrial compared to PMIP3/4 models (Kageyama et al., 2021).

### 4.3 Last Glacial Maximum

For the last glacial maximum (LGM), we prescribe boundary conditions following the PMIP4 protocol (Kageyama et al., 2017), with the GLAC-1D ice sheet, bathymetry and land-sea mask reconstruction (Tarasov et al., 2012). The simulation is started from the present–day equilibrium followed by a switch to LGM boundary conditions during which the topography, bathymetry and ocean volume are adjusted. Total ocean salinity is conserved in this process. The model is then run for 5000 years to ensure equilibrium is reached.

The simulated global cooling relative to pre-industrial is 6.4 K, close to the value of 6.2 K obtained with CLIMBER-2 (Ganopolski et al., 1998), which is also the most recent reconstruction-based estimate (Tierney et al., 2020). It is however on the cold side of the range produced by PMIP4 models (3.3–7.2 K) (Kageyama et al., 2021). The zonal mean annual temperature change is compared to PMIP3/4 models in Fig. 27. CLIMBER-X shows a cooling in the tropics that is more pronounced than in most other models, while at high latitudes the temperature difference falls well inside the PMIP3/4 range. In terms of sea surface temperatures, the model results agree well with the proxy-based reconstruction by Tierney et al. (2020) (Fig. 28), which also shows a pronounced cooling in the Tropics, as opposed to e.g. Paul et al. (2021) (Fig. 28c).

The global ocean cools by 2.5 °C, in excellent agreement with 2.57±0.24 °C in Bereiter et al. (2018), with the deep ocean temperature being close to the freezing point of seawater ($< -1$ °C) in all ocean basins, in accordance with Adkins et al. (2002). The AMOC is weaker and shallower at the LGM relative to the preindustrial, with a maximum overturning strength of $\sim$14 Sv and extending down to a depth of $\sim$2500 m (Fig. 29a). This is contrary to most PMIP3/4 models, which tend to produce



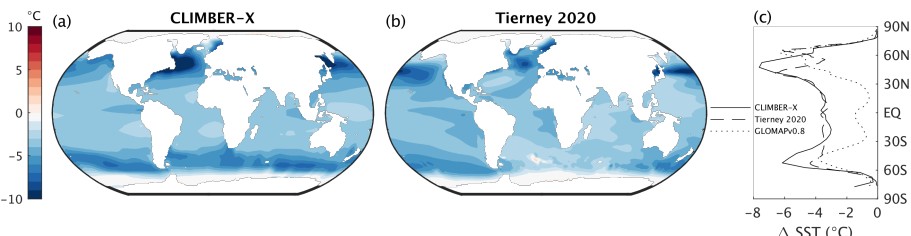

**Figure 28.** Last glacial maximum annual mean sea surface temperature differences relative to preindustrial compared to reconstructions (Paul et al., 2021; Tierney et al., 2020).

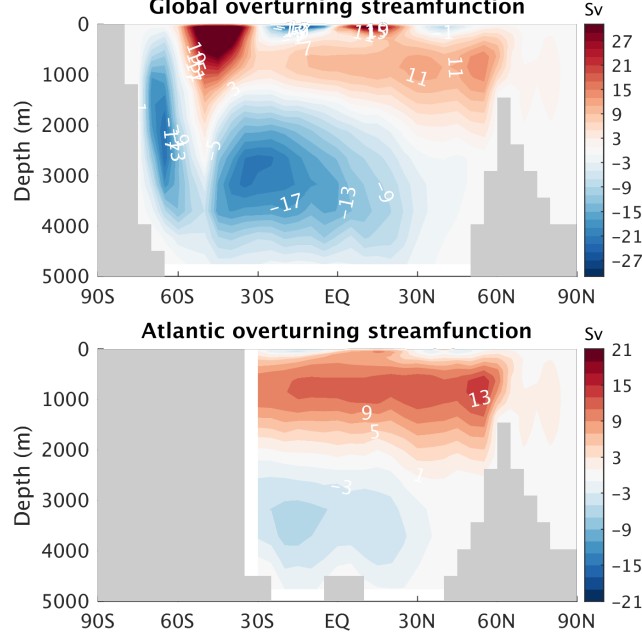

**Figure 29.** Ocean overturning circulation at the last glacial maximum: (top) global and (bottom) Atlantic.

a more vigorous and deeper Atlantic overturning at LGM (Weber et al., 2007; Kageyama et al., 2021), but is possibly in better agreement with proxy reconstructions (e.g. McManus et al., 2004; Bohm et al., 2015). The AMOC weakening is accompanied by a strengthening of Antarctic bottom water formation (Fig. 29b), which is related to an increase in brine rejection associated

with a pronounced expansion of sea ice in the Southern Ocean (Fig. 30b), similarly to what has been found in other models (e.g. Nadeau et al., 2019; Shin et al., 2003; Stouffer and Manabe, 2003).





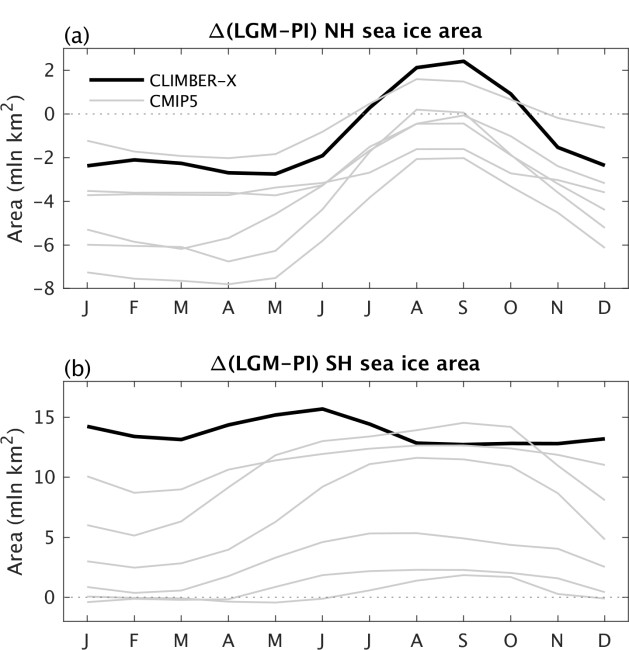

**Figure 30.** Difference in seasonal sea ice area in (a) NH and (b) SH between the last glacial maximum and the pre-industrial. CLIMBER-X results (black) are compared with CMIP5 models (grey).



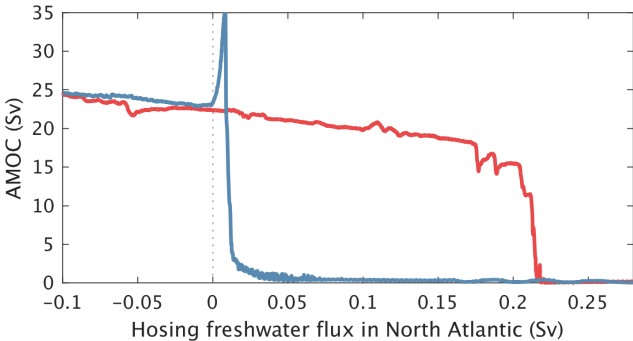

**Figure 31.** AMOC hysteresis for freshwater hosing applied to the latitudinal belt between 50 °N and 70 °N in the Atlantic. Freshwater hosing is changed at a rate of 0.025 Sv per 1000 years.

## 4.4 AMOC stability

The meridional overturning circulation in the Atlantic ocean plays an important role in the global climate system. Since the pioneering work of Stommel (1961), who used a simple box model to show that the AMOC could have multiple stable states,
the stability of the AMOC has received considerable attention, with several studies using ocean general circulation models confirming the bi-stable nature of the system (e.g. Manabe and Stouffer, 1988; Rahmstorf, 1995).

Consistent with findings by other EMICs (Rahmstorf et al., 2005), CLIMBER-X also shows a hysteresis behavior of the AMOC when the freshwater balance of the North Atlantic is perturbed (Fig. 31). The width of the hysteresis is ~0.2 Sv. Under pre-industrial conditions, the AMOC is in a mono-stable regime in the model, although relatively close to the bistable regime.
The present-day stability of the AMOC is still debated (e.g Weijer et al., 2019).

The different climatic conditions associated with the two equilibrim states of the AMOC are illustrated by mean annual temperature and precipitation differences in Fig. 32. The temperature difference shows the classic sea-saw pattern, with cooling in the NH and warming in the SH as a response to AMOC shutdown (Fig. 32a). The cooling reaches up to 10 K in the North Atlantic and is compensated by a warming of up to 10 K in the Southern Ocean. The cooling produced in the North Atlantic is
consistent with GCM simulations (e.g. Vellinga and Wood, 2002; Jackson et al., 2015), while the large warming in the Southern Ocean is not seen in these GCMs, possibly because they are not run into equilibrium. Precipitation changes in the AMOC off state compared to the on state are clearly seen in the Tropics as a result of a pronounced southward shift in the intertropical convergence zone (Fig. 32b). Precipitation is also reduced in the North Atlantic, simply as a consequence of the colder climatic conditions.

## 500   5   Applicability and limitations of CLIMBER-X

CLIMBER-X does not resolve synoptic variability and does not exhibit interannual internal variability and is therefore not suited to investigate weather extremes or internal climate oscillations like ENSO. The atmospheric component of CLIMBER-



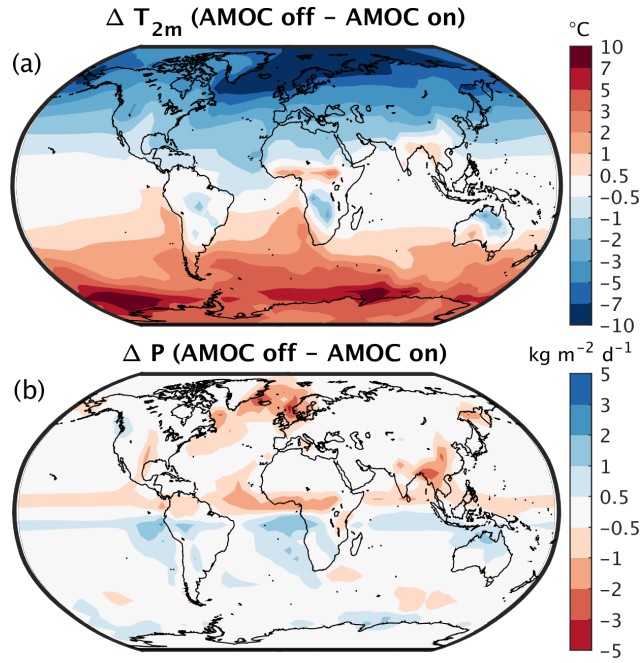

**Figure 32.** Annual mean (a) near-surface air temperature and (b) precipitation differences between AMOC off and on states.

X is based on a statistical-dynamical approach, which employs a number of significant simplifications and assumptions. On the one hand side, such an approach allows developing a model that is several orders of magnitude faster than GCMs with similar resolution. On the other, these simplifications and a set of parameterizations explicitly derived from present–day climate limit the models' applicability to climate states fundamentally different from the present one, for which these assumptions don't necessarily hold anymore. Some examples include Snowball Earth, for which the parameterisation of the mean meridional circulation is not suited and simulations of the nuclear winter or the climate response to an asteriod impact, because the lapse rate parameterisation is not valid under these circumstances.

## 6   Conclusions

We have described the major features of the climate component of the newly developed CLIMBER-X Earth System model. CLIMBER-X relies on the geostrophic approximation for the description of both atmospheric and oceanic circulation. It does not explicitly resolve atmospheric and oceanic eddies, but the effect of both is parameterized as a diffusive process. The simplified dynamics, together with the use of a daily time step for most processes and the lower spatial resolution of $5° \times 5°$, provides a substantial advantage in terms of computational costs relative to general circulation models, which are based on the primitive equations. Nevertheless, in terms of the number of physical processes which are represented in the model, CLIMBER-X is comparable to state-of-the-art climate models. This is highlighted for instance by the realistic representation of climate feedbacks in the model. Moreover, CLIMBER-X also includes a model for the global carbon cycle and is coupled to an ice



sheet model (both of which will be described in detail in forthcoming papers), and thus allows simulations of the evolution

of Earth system as a whole and the investigation of complex interactions and feedbacks between different components of the Earth system on time scales ranging from decades to hundreds of thousands of years. CLIMBER-X is therefore an ideal tool to explore the long-term past and future evolution of the Earth system.

*Code and data availability.* The source code of CLIMBER-X v1.0 is archived on Zenodo (https://doi.org/10.5281/zenodo.6301052), to-gether with the input data and the instructions to install and run the model. The FFTW library, which is already included in CLIMBER-X,

is available from https://www.fftw.org/. CMIP6 model data are licensed under a Creative Commons Attribution-ShareAlike 4.0 International License (https://creativecommons.org/licenses) and can be accessed through the ESGF nodes (for instance esgf-data.dkrz.de/search/cmip6-dkrz/). ERA-Interim data are available from https://apps.ecmwf.int/datasets/data/interim-full-moda/levtype=sfc/, while ERA5 reanalysis data can be downloaded from the Copernicus Climate Data Store at https://cds.climate.copernicus.eu/.

## Appendix A: SESAM detailed description

## A1    Vertical structure

SESAM is based on a number of simplifications compared to current state-of-the-art atmospheric models. One such simplifi-cation is that, in all equations apart from that describing atmospheric dynamics, the atmospheric pressure at each model level is not computed using the hydrostatic approximation but is assumed to be an exponential function of the height above sea level only:

$$p(z) = p_0 e^{-\frac{z}{H_a}}, \tag{A1}$$

where $H_a = R_d T_0 / g$ is the atmospheric height scale. The mean sea level pressure $p_0$ is determined from the atmospheric mass conservation condition:

$$\int_\Omega p(z_s)\, d\omega = \frac{M_a g}{A_e}, \tag{A2}$$

where $\Omega$ indicates the surface of the Earth, $z_s$ is surface elevation above sea level, $M_a$ is the total atmospheric mass and $A_e$ is

the area of the Earth surface. Hence, the mean sea level pressure will change with changing topography.

For atmospheric dynamics the dependence of pressure on atmospheric temperature is explicitly accounted for through the parameterisation of sea level pressure and through the thermal wind equation (see Appendix A2 below).

Air density is described as a function of elevation:

$$\rho(z) = \rho_0 e^{-\frac{z}{H_a}}, \tag{A3}$$

with the reference density at sea level computed from the ideal gas law:

$$\rho_0 = \frac{p_0}{R_d T_0}. \tag{A4}$$





The vertical profile of temperature is computed as follows:

$$T(z) = T_{\mathrm{a}} - \int_{z_{\mathrm{s}}}^{z} \Gamma(z')dz', \tag{A5}$$

where $T_{\mathrm{a}}$ is the prognostic atmospheric temperature at the surface and $\Gamma$ is the lapse rate. Although the atmospheric temperature

lapse rate is remarkably close to a constant value of $\sim 6.5\,\mathrm{K\,km^{-1}}$, the deviations from this value are important in order to reproduce both a realistic present-day climate and climate feedbacks. In SESAM, $\Gamma$ is parameterized as:

$$\Gamma(z) = \begin{cases} \Gamma_{\mathrm{s}}, & z \leq z_{\mathrm{s}} + H_{\Gamma,\mathrm{s}}, \\ \Gamma_{\mathrm{b}} + (\Gamma_{\mathrm{t}} - \Gamma_{\mathrm{b}}) \frac{z}{H_{\Gamma,\mathrm{t}}}, & z > z_{\mathrm{s}} + H_{\Gamma,\mathrm{s}} \quad \text{and} \quad z \leq H_{\mathrm{T}}, \\ 0, & z > H_{\mathrm{T}}. \end{cases} \tag{A6}$$

In general the lapse rate therefore linearly increases with height, with $\Gamma_{\mathrm{b}}$ and $\Gamma_{\mathrm{t}}$ depending on atmospheric humidity $q_{\mathrm{a}}$ only:

$$\Gamma_{\mathrm{b}} = c_1^{\Gamma} - c_2^{\Gamma} q_{\mathrm{a}}, \tag{A7}$$

$$\Gamma_{\mathrm{t}} = \Gamma_{\mathrm{b}} - c_2^{\Gamma} q_{\mathrm{a}} + c_3^{\Gamma}. \tag{A8}$$

In a layer close to the surface the lapse rate depends on near-surface stability:

$$\Gamma_{\mathrm{s}} = \begin{cases} c_4^{\Gamma} \sqrt{\max(0, T_{\mathrm{a}} - T_{\star})}, & \text{ocean} \\ c_5^{\Gamma} (T_{\mathrm{a}} - T_{\star}), & \text{land and} \quad T_{\mathrm{a}} - T_{\star} > 0 \\ c_6^{\Gamma} (T_{\mathrm{a}} - T_{\star}), & \text{land and} \quad T_{\mathrm{a}} - T_{\star} < 0 \\ c_5^{\Gamma} (T_{\mathrm{a}} - T_{\star}), & \text{ice} \end{cases} \tag{A9}$$

where $T_{\star}$ is the skin temperature and $\Gamma_{\mathrm{s}}$ is limited to be lower than $7.5 \times 10^{-3}\,\mathrm{Km^{-1}}$ over ocean and $10 \times 10^{-3}\,\mathrm{Km^{-1}}$ over land and ice. In particular, equation A9 allows SESAM to reproduce near-surface inversions which are important for surface

climate.

The tropopause height $H_{\mathrm{T}}$ is derived assuming that the stratosphere is in radiative equilibrium:

$$\frac{\partial H_{\mathrm{T}}}{\partial t} = -c_1^{\mathrm{tp}} (R_{\mathrm{str,net}} + S). \tag{A10}$$

The net radiation in the stratosphere, $R_{\mathrm{str,net}}$, includes the balance of longwave radiation and the shortwave radiation absorbed by ozone. The effect of atmospheric dynamics on tropopause height is explicitly included in the prescribed latitudinal profile

of $S$, which only depends on the position ($\phi_{\mathrm{ITCZ}}$) and width ($\Delta\phi_{\mathrm{Had}}$) of the Hadley cells:

$$S = c_2^{\mathrm{tp}} \left[ 1 - c_3^{\mathrm{tp}} \left( 1 - \sin^8 \frac{0.85 (\phi - \phi_{\mathrm{ITCZ}})}{0.5 \Delta\phi_{\mathrm{Had}}} \right) \right]. \tag{A11}$$

Potential temperature, which is a conserved quantity for adiabatic motions, is computed as:

$$\theta(z) = T(z) + \Gamma_{\mathrm{d}} z, \tag{A12}$$





where $\Gamma_d = g/c_p$ is the dry adiabatic lapse rate and $c_p$ is the specific heat capacity of air at constant pressure.

In CLIMBER-2 specific humidity was specified as decaying exponentially with height, but this can imply unrealistic relative humidities in the upper troposphere, where humidity is important for longwave radiation. This problem is also highlighted by the fact that there is no tight coupling between water vapor and lapse rate feedbacks in CLIMBER-2, contrary to what is observed in most climate models. To overcome this limitation, in SESAM the vertical profile of specific humidity is expressed through temperature and a parameterisation of the relative humidity variation with height:

$$r(z) = \begin{cases} r_a & z \leq z_{pbl} \\ r_a e^{-\frac{z - z_{pbl}}{H_r}} & z > z_{pbl} \quad \text{and} \quad z \leq z_s + c_4^r \\ r_a e^{-\frac{z_s + c_4^r - z_{pbl}}{H_r}} & z > z_s + c_4^r \quad \text{and} \quad z \leq H_T \\ r_{st} & z > H_T, \end{cases} \tag{A13}$$

where $z_{pbl} = z_s + c_5^r$ is the elevation of the planetary boundary layer. The relative humidity height scale $H_r$ is constant and uniform in the extratropics and depends on vertical velocity at $700\,\mathrm{hPa}$ in the tropics:

$$H_r = f_{trop} \cdot c_1^r \cdot e^{c_2^r \cdot w_{700}} + (1 - f_{trop}) \cdot c_1^r \cdot c_3^r, \tag{A14}$$

with $f_{trop} = 1 - \sin^8 \varphi$, where $\varphi$ is defined below in Appendix A2. Specific humidity is then computed as:

$$q(z) = r(z) \cdot q_{sat}(T(z), p(z)), \tag{A15}$$

where the specific humidity at saturation $q_{sat}$ is computed assuming saturation over ice at temperatures below -15 °C, saturation over water at temperatures above $T_0$ and a weighted mean of saturation over water and ice in the intermediate temperature range.

## A2    Dynamics

The dynamics of the atmosphere in SESAM is similar to that in CLIMBER-2 but with several notable improvements. Horizontal velocity in the atmosphere is computed as the sum of geostrophic and ageostrophic components:

$$\boldsymbol{u} = \boldsymbol{u}_g + \boldsymbol{u}_a. \tag{A16}$$

The geostrophic components of velocity at any height within the troposphere are obtained using the thermal wind approximation:

$$u_g(z) = -\frac{1}{\rho_0 f R_e} \frac{\partial p_{sl}}{\partial \phi} - \int_0^z \frac{g}{T_0 f R_e} \frac{\partial T}{\partial \phi} dz, \tag{A17}$$

$$v_g(z) = \frac{1}{\rho_0 f R_e \cos \phi} \frac{\partial p_{sl}}{\partial \lambda} + \int_0^z \frac{g}{T_0 f R_e \cos \phi} \frac{\partial T}{\partial \lambda} dz, \tag{A18}$$





**Table A1.** Parameters for vertical structure.

| Parameter | value |
| --- | --- |
| *Lapse rate* | |
| $c_1^\Gamma$ | $3.8 \times 10^{-3}\,\mathrm{K m^{-1}}$ |
| $c_2^\Gamma$ | $0.02\,\mathrm{K m^{-1}}$ |
| $c_3^\Gamma$ | $6 \times 10^{-3}\,\mathrm{K m^{-1}}$ |
| $c_4^\Gamma$ | $5 \times 10^{-3}\,\mathrm{K^{1/2} m^{-1}}$ |
| $c_5^\Gamma$ | $2 \times 10^{-3}\,\mathrm{m^{-1}}$ |
| $c_6^\Gamma$ | $10 \times 10^{-3}\,\mathrm{m^{-1}}$ |
| $H_{\Gamma,\mathrm{s}}$ | $1500\,\mathrm{m}$ |
| $H_{\Gamma,\mathrm{t}}$ | $15000\,\mathrm{m}$ |
| *Relative humidity profile* | |
| $c_1^\mathrm{r}$ | $2500\,\mathrm{m}$ |
| $c_2^\mathrm{r}$ | $200\,\mathrm{s m^{-1}}$ |
| $c_3^\mathrm{r}$ | $2.4$ |
| $c_4^\mathrm{r}$ | $3000\,\mathrm{m}$ |
| $c_5^\mathrm{r}$ | $1000\,\mathrm{m}$ |
| $r_{\mathrm{st}}$ | $0.05$ |
| *Tropopause height* | |
| $c_1^\mathrm{tp}$ | $100\,\mathrm{m^3 W^{-1}}$ |
| $c_2^\mathrm{tp}$ | $18\,\mathrm{W m^{-2}}$ |
| $c_3^\mathrm{tp}$ | $1$ |

where $p_{\mathrm{sl}}$ is the sea level pressure, $f$ is the Coriolis parameter, $R_{\mathrm{e}}$ is the radius of the Earth and $\lambda$ and $\phi$ are longitude and latitude, respectively. The ageostrophic wind components in the planetary boundary layer are computed from sea level pressure and cross-isobar angle, $\alpha$, as:

$$u_{\mathrm{a}} = -\frac{\sin\alpha\cos\alpha}{\rho_0 |f| R_{\mathrm{e}}\cos\phi}\frac{\partial p_{\mathrm{sl}}}{\partial\lambda}, \tag{A19}$$

$$v_{\mathrm{a}} = -\frac{\sin\alpha\cos\alpha}{\rho_0 |f| R_{\mathrm{e}}}\frac{\partial p_{\mathrm{sl}}}{\partial\phi}. \tag{A20}$$

As in Petoukhov et al. (2000) the ageostrophic wind in the planetary boundary layer is compensated in the upper troposphere in order to conserve mass in the atmospheric column. Since the geostrophic approximation is not valid close to the equator, the Coriolis parameter is limited to be $|f| > 3 \times 10^{-5}\,\mathrm{s^{-1}}$ in the geostrophic and $|f| > 1 \times 10^{-5}\,\mathrm{s^{-1}}$ in the ageostrophic wind equations.





The cross-isobar angle is determined from the condition that the shear stress is continous between the Ekman layer and the surface layer (Petoukhov et al., 2000):

$$C_\mathrm{D} U_\mathrm{s}^2 = \frac{U_\mathrm{s}}{\epsilon} \sin\alpha \sqrt{2|f|K}, \tag{A21}$$

where $\epsilon = \sqrt{1 - \sin 2\alpha}$, $U_\mathrm{s}$ is the module of the surface wind, $K$ is the kinematic vertical viscosity coefficient in the planetary
boundary layer (PBL) and the drag coefficient $C_\mathrm{D}$ is

$$C_\mathrm{D} = \left( \frac{\kappa}{\ln \frac{z_\mathrm{ref}}{z_0 + z_\mathrm{oro}}} \right)^2, \tag{A22}$$

where $\kappa$ is the von-Karman constant, $z_0 = 100\,\mathrm{m}$ a reference height, $z_0$ the surface roughness length and $z_\mathrm{oro}$ the orographic roughness, computed from the subgrid scale standard deviation of orography as:

$$z_\mathrm{oro} = 0.004\sigma_\mathrm{oro}. \tag{A23}$$

Eq. A21 is solved for $\alpha$ with the approximation $U_\mathrm{s} \approx \sqrt{2K}$.

The near-surface wind components are computed using the Taylor model (see e.g. Hansen et al. (1983)) with the addition of a simple representation of katabatic winds:

$$u_\mathrm{s} = \epsilon\left(u_\mathrm{g}(0)\cos\alpha - v_\mathrm{g}(0)\sin\alpha\right) + u_\mathrm{k}, \tag{A24}$$

$$v_\mathrm{s} = \epsilon\left(v_\mathrm{g}(0)\cos\alpha + u_\mathrm{g}(0)\sin\alpha\right) + v_\mathrm{k}. \tag{A25}$$

Katabatic winds $(u_\mathrm{k}, v_\mathrm{k})$ are important in surface inversion conditions over slopes, as is at present mainly the case over the large Greenland and Antarctic ice sheets. They are included in the model based on a simple balance of bouyancy force and friction, ignoring Coriolis and background pressure gradient, following the Prandtl 1942 model (e.g. Fedorovich and Shapiro, 2009):

$$u_\mathrm{k} = \sqrt{\frac{gh}{C_\mathrm{D}} \frac{T_\mathrm{2m} - T_\star}{T_\mathrm{2m}} \frac{1}{R_\mathrm{e}\cos\phi} \left|\frac{\partial z_\mathrm{s}}{\partial\lambda}\right|} \cdot \mathrm{sign}\left(-\frac{\partial z_\mathrm{s}}{\partial\lambda}\right), \tag{A26}$$

$$v_\mathrm{k} = \sqrt{\frac{gh}{C_\mathrm{D}} \frac{T_\mathrm{2m} - T_\star}{T_\mathrm{2m}} \frac{1}{R_\mathrm{e}} \left|\frac{\partial z_\mathrm{s}}{\partial\phi}\right|} \cdot \mathrm{sign}\left(-\frac{\partial z_\mathrm{s}}{\partial\phi}\right), \tag{A27}$$

with $h = 100\,\mathrm{m}$ and $T_\mathrm{2m}$ the near-surface air temperature.

Sea level pressure is computed as the sum of zonally averaged and azonal components:

$$p_\mathrm{sl} = \overline{p_\mathrm{sl}} + p_\mathrm{sl}^*. \tag{A28}$$

The zonal component of sea level pressure is computed using a parameterization similar to that described in Petoukhov et al.
(2000). This parameterization is based on the assumption that the strength and position of the major cells of atmospheric meridional overturning circulation are controlled by average meridional temperature gradients, the zonally averaged surface





drag and surface elevation. The zonally averaged sea level pressure is defined through the zonal mean meridional (ageostrophic) wind component in the PBL:

$$\frac{\partial \overline{p_{\mathrm{sl}}}}{\partial \phi} = -\overline{v_{\mathrm{a}}}(\phi) \frac{\rho_0 f R_{\mathrm{e}}}{\sin \alpha \cos \alpha}, \tag{A29}$$

where the zonally averaged meridional (sea level) ageostrophic velocity is parameterized as:

$$\overline{v_{\mathrm{a}}}(\phi) = -C_i \Delta T_i^j F_{\mathrm{z}}(\phi) \sin \varphi, \quad (i-1)\pi < \varphi < i\pi, \tag{A30}$$

with:

$$\varphi = 6 \cdot D_{\mathrm{had}} \left( \phi - \frac{\phi_{\mathrm{ITCZ}}}{c_1^{\mathrm{mmc}} \left(\phi - \phi_{\mathrm{ITCZ}}\right)^2 + 1} \right). \tag{A31}$$

$C_i$ are empirical parameters, $i = 1$ corresponds to the Hadley cell, $i = 2$ - Ferrel cell and $i = 3$ - polar cell, $j = 1$ corresponds

to the Northern and $j = 2$ to the Southern Hemisphere. Contrary to CLIMBER-2, the same $C_i$ values are used for the Northern and Southern Hemisphere. The position of the inter-tropical convergence zone, $\phi_{\mathrm{ITCZ}}$, depends on the temperature difference between the two hemispheres:

$$\phi_{\mathrm{ITCZ}} = c_2^{\mathrm{mmc}} \left(T_{\mathrm{NH}} - T_{\mathrm{SH}}\right). \tag{A32}$$

$D_{\mathrm{had}}$ controls the width of the Hadley cells and is a function of tropical temperature:

$$D_{\mathrm{had}} = \frac{c_3^{\mathrm{mmc}}}{T_{\mathrm{trp}} - c_4^{\mathrm{mmc}}}. \tag{A33}$$

This ensures that the Hadley cells are expanding with warming, consistent with empirical evidence (e.g. Hu et al., 2018) and models (Frierson et al., 2007). $T_i^j$ is the mean temperature of each cell. The temperature gradients are proportional to meridional differences in zonal mean sea level temperature:

$$\Delta T_i^j = \begin{cases} \overline{T}(\phi_i^j) - \max(\overline{T}(\phi)), & i = 1 \\ \overline{T}(\phi_i^j) - \overline{T}(\phi_{i-1}^j), & i = 2, 3. \end{cases} \tag{A34}$$

The latitudes used to compute the gradients are fixed at $|\phi_1^j| = \pi/6$, $|\phi_2^j| = \pi/3$ and $|\phi_3^j| = \pi/2$. The topography factor is defined as:

$$F_{\mathrm{z}}(\phi) = \overline{1 - z_{\mathrm{s}}/c_5^{\mathrm{mmc}}}. \tag{A35}$$

The azonal sea level pressure is the sum of a thermal and an orographic component:

$$p_{\mathrm{sl}}^* = p_{\mathrm{sl,T}}^* + p_{\mathrm{sl,O}}^*. \tag{A36}$$

The azonal sea level pressure component arising from the thermally induced stationary planetary waves is computed following Petoukhov et al. (2000) and Petoukhov et al. (2003):

$$p_{\mathrm{sl,T}}^* = -\frac{g p_0 H_0}{2 R_{\mathrm{d}} T_0^2} T_{\mathrm{sl}}^*, \tag{A37}$$





**Table A2.** Parameters for atmospheric dynamics.

| Parameter | value |
|---|---|
| *Mean meridional circulation* | |
| $C_1$ | 0.3 |
| $C_2$ | 0.05 |
| $C_3$ | 0.005 |
| $c_1^{\mathrm{mmc}}$ | 5 |
| $c_2^{\mathrm{mmc}}$ | $0.017\,\mathrm{K}^{-1}$ |
| $c_3^{\mathrm{mmc}}$ | $90\,\mathrm{K}$ |
| $c_4^{\mathrm{mmc}}$ | $200\,\mathrm{K}$ |
| $c_5^{\mathrm{mmc}}$ | $750\,\mathrm{m}$ |
| *Azonal sea level pressure* | |
| $H_0$ | $10000\,\mathrm{m}$ |
| $\tau_{\mathrm{e}}$ | $5\,\mathrm{days}$ |

where $T_{\mathrm{sl}}$ is the skin temperature reduced to sea level using a constant lapse rate of $6.5\,\mathrm{K km}^{-1}$. The effect of topographic stationary waves is accounted for using the simple 1D barotropic model for forced topographic Rossby waves of Charney and Eliassen (1949) as described in Held (1983) and Holton (2004), with the linearized vorticity equation written as:

$$\overline{u}\frac{\partial \zeta}{\partial \lambda} + \beta v + \frac{\zeta}{\tau_{\mathrm{e}}} = -\frac{f}{H_{\mathrm{T}}}\overline{u}\frac{\partial z_{\mathrm{s}}}{\partial \lambda}, \tag{A38}$$

where $\zeta$ is the relative vorticity, $\beta$ is the meridional derivative of the Coriolis parameter $f$, $\tau_{\mathrm{e}}$ is the damping time scale due to Ekman pumping and $\overline{u}$ is taken to be the zonal mean zonal wind at $500\,\mathrm{hPa}$. A meridional wave number corresponding to a latitudinal half-wavelength of $35\,^{\circ}$ is assumed. The equation is solved independently for each latitudinal belt by Fourier expansion of the topography $z_{\mathrm{s}}(\lambda)$, by writing the geostrophic wind vector and the relative vorticity in terms of a streamfunction $\Psi$. The FFTW3 library (Frigo and Johnson, 2005) is used for the Fourier transform. The sea level pressure perturbation due to topographic stationary waves is then derived as:

$$p_{\mathrm{sl,O}}^{*} = \Psi f \rho(500\,\mathrm{hPa}). \tag{A39}$$

### A3 Thermodynamics

The energy balance equation, vertically integrated from the surface to the top of the atmosphere, is written as:

$$\frac{\partial Q_{\mathrm{T}}}{\partial t} = -\frac{1}{R_{\mathrm{e}}\cos\phi}\left[\frac{\partial}{\partial \lambda}\int_{z_{\mathrm{s}}}^{H_{\mathrm{T}}}\rho(u\theta + \widehat{u'\theta'})dz + \frac{\partial}{\partial \phi}\int_{z_{\mathrm{s}}}^{H_{\mathrm{T}}}\cos\phi\,\rho(v\theta + \widehat{v'\theta'})dz\right] + c_{\mathrm{v}}^{-1}\left(\mathrm{SW_a} + \mathrm{LW_a} + L_{\mathrm{e}}P_{\mathrm{w}} + L_{\mathrm{s}}P_{\mathrm{s}} + \mathrm{SH}\right),$$


$$\tag{A40}$$

where $Q_{\mathrm{T}} = \int_{z_{\mathrm{s}}}^{H_{\mathrm{TOA}}} \rho T dz$ is the heat content of the atmospheric column, $\mathrm{SW_a}$ is the shortwave radiation absorbed by the atmosphere, $\mathrm{LW_a}$ is the net atmosphere longwave radiation balance, $P_{\mathrm{w}}$ is rainfall and $P_{\mathrm{s}}$ is snowfall, $L_{\mathrm{e}}$ is the latent heat of evaporation and $L_{\mathrm{s}}$ the latent heat of sublimation, SH is the surface sensible heat flux, and $c_{\mathrm{v}}$ is the heat capacity of air at constant volume. The horizontal heat transport due to synoptic processes, $\widehat{u'\theta'}$ and $\widehat{v'\theta'}$, is represented as macroturbulent diffusion as described in Appendix A5. In CLIMBER-2 only the non-thermal wind is used in the energy balance equation, with the beta effect accounted for separately. This allowed a large timestep of up to $1\,\mathrm{day}$. Because the CLIMBER-X horizontal resolution is much higher than in CLIMBER-2, a relatively short time step of $\sim 2\,\mathrm{h}$ is anyway needed and with such small time step the energy equation is stable even using the full wind vector. The energy balance equation is solved for $T_{\mathrm{a}}$ and the near-surface air temperature is diagnosed as:

$$T_{2\mathrm{m}} = \frac{T_{\mathrm{a}} + T_{\star}}{2}. \tag{A41}$$

$T_{2\mathrm{m}}$ is also the temperature that is used to compute the surface turbulent sensible heat flux.

## A4 Hydrology

The water balance equation, vertically integrated from the surface to the top of the atmosphere, is written as:

$$\frac{\partial Q_{\mathrm{q}}}{\partial t} = -\frac{1}{R_{\mathrm{e}}\cos\phi}\left[\frac{\partial}{\partial\lambda}\int_{z_{\mathrm{s}}}^{H_{\mathrm{T}}}\rho(uq + \widehat{u'q'})dz + \frac{\partial}{\partial\phi}\int_{z_{\mathrm{s}}}^{H_{\mathrm{T}}}\cos\phi\rho(vq + \widehat{v'q'})dz\right] + E - P \tag{A42}$$

where $Q_{\mathrm{q}} = \int_{z_{\mathrm{s}}}^{H_{\mathrm{TOA}}} \rho q dz$ is the water vapor content of the atmospheric column, $E$ is surface evaporation and $P$ is total precipitation. The horizontal moisture transport due to synoptic processes, $\widehat{u'q'}$ and $\widehat{v'q'}$, is represented as macroturbulent diffusion as described in Appendix A5.

The water balance equation is solved for $q_{\mathrm{a}}$ and the near-surface air specific humidity, which is used to compute the surface latent heat flux, is diagnosed as:

$$q_{2\mathrm{m}} = r_{2\mathrm{m}} \cdot q_{\mathrm{sat}}(T_{2\mathrm{m}}), \tag{A43}$$

with $r_{2\mathrm{m}} = (r_{\mathrm{a}} + r_{\star})/2$, and $r_{\star} = q_{\mathrm{a}}/q_{\mathrm{sat}}(T_{\star})$. Precipitation in the model is generated as follows:

$$P = \max\left(0, C + C_{\mathrm{slope}} + E\right)\frac{r_{\mathrm{a}}}{r_{\mathrm{a}}^{\mathrm{max}}} + \frac{Q_{\mathrm{q}}r_{\mathrm{a}}}{\tau_{\mathrm{p}}}, \tag{A44}$$

where $C$ is moisture convergence into the atmospheric column by advection and diffusion and $C_{\mathrm{slope}}$ explicitly represents an additional moisture convergence due to synoptic activity on slopes. The first term on the right hand side of eq. A44 represents a gradual removal of converging moisture by precipitation as atmospheric relative humidity increases, with all water entering an atmospheric column being added to precipitation when the atmospheric relative humidity reaches a maximum value $r_{\mathrm{a}}^{\mathrm{max}}$. The second term on the right of eq. A44 generates additional precipitation by removing atmospheric moisture with a given time



**Table A3.** Parameters for hydrological cycle.

| Parameter | value |
|---|---|
| $\tau_{\mathrm{p}}$ | 50 days |
| $r_{\mathrm{a}}^{\mathrm{max}}$ | 0.95 |
| $c_{\mathrm{slope}}^{\mathrm{p}}$ | 0.005 |

scale $\tau_{\mathrm{p}}$, and is applied only over land. The moisture convergence due to synoptic activity on slopes is computed assuming that moisture convergence is proportional to the vertical velocity induced by synoptic winds impacting on the slope:

$$C_{\mathrm{slope}} = c_{\mathrm{slope}}^{\mathrm{p}} \sqrt{K} |\nabla z_{\mathrm{s}}| \rho_0 q_{\mathrm{a}}, \tag{A45}$$

where the synoptic wind is expressed through the eddy kinetic energy, $K$.

## A5 Synoptic processes

Horizontal fluxes of energy and water originating from unresolved synoptic variability are represented as a macroturbulent diffusion process (Petoukhov et al., 2000):

$$\widehat{u'\theta'} = \widehat{u'T'} = -A_{\mathrm{T}} \frac{1}{R_{\mathrm{e}}\cos\phi} \frac{\partial T}{\partial \lambda}, \tag{A46}$$

$$\widehat{v'\theta'} = \widehat{v'T'} = -A_{\mathrm{T}} \frac{1}{R_{\mathrm{e}}} \frac{\partial T}{\partial \phi}, \tag{A47}$$

$$\widehat{u'q'} = -A_{\mathrm{q}} \frac{1}{R_{\mathrm{e}}\cos\phi} \frac{\partial q}{\partial \lambda}, \tag{A48}$$

$$\widehat{v'q'} = -A_{\mathrm{q}} \frac{1}{R_{\mathrm{e}}} \frac{\partial q}{\partial \phi}. \tag{A49}$$

The diffusivities $A_{\mathrm{T}}$ and $A_{\mathrm{q}}$ are isotropic, vertically uniform and depend on eddy kinetic energy $K$:

$$A_{\mathrm{T}} = c_5^{\mathrm{syn}} \sqrt{K}, \tag{A50}$$

$$A_{\mathrm{q}} = c_6^{\mathrm{syn}} \cdot K. \tag{A51}$$

The different dependence of the diffusivities on eddy kinetic energy follows from Caballero and Hanley (2012).

The kinetic energy of synoptic eddies is described by an evolution equation:

$$\frac{\partial K}{\partial t} = -\nabla \cdot (\boldsymbol{u}K) + \nabla(A_{\mathrm{T}}\nabla K) + P_{\mathrm{K}} - D_{\mathrm{K}} \tag{A52}$$

Kinetic energy production is proportional to the maximum Eady model baroclinic growth rate (e.g. Hoskins and Valdes, 1990):

$$P_{\mathrm{K}} = c_1^{\mathrm{syn}} + c_2^{\mathrm{syn}} \frac{f}{N} \left| \frac{\partial \boldsymbol{u}}{\partial z} \right|, \tag{A53}$$





**Table A4.** Parameters for synoptic processes.

| Parameter | value |
|---|---|
| $c_1^{\mathrm{syn}}$ | $1\times10^{-4}\,\mathrm{m^2 s^{-3}}$ |
| $c_2^{\mathrm{syn}}$ | $1.6\times10^{4}\,\mathrm{m^2 s^{-2}}$ |
| $c_3^{\mathrm{syn}}$ | $8\times10^{-7}\,\mathrm{m^{-1}}$ |
| $c_4^{\mathrm{syn}}$ | $1\times10^{-4}\,\mathrm{m^{-1}}$ |
| $c_5^{\mathrm{syn}}$ | $2\times10^{5}\,\mathrm{m}$ |
| $c_6^{\mathrm{syn}}$ | $2\times10^{4}\,\mathrm{s}$ |
| $c_7^{\mathrm{syn}}$ | $0.7$ |
| $c_8^{\mathrm{syn}}$ | $1\times10^{-3}$ |

where the Brunt-Vaisala frequency $N$ is defined as:

$$N = \sqrt{\frac{g}{\theta}\frac{\partial\theta}{\partial z}}, \tag{A54}$$

and all vertical gradients are computed between 850 and 500 hPa. The dissipation of eddy kinetic energy is given by:

$$D_{\mathrm{K}} = \left(c_3^{\mathrm{syn}} + c_4^{\mathrm{syn}} C_{\mathrm{D}}\right) K^{3/2}. \tag{A55}$$

The synoptic component of near-surface wind is computed from eddy kinetic energy as follows:

$$U_{\mathrm{syn}} = c_7^{\mathrm{syn}} \epsilon \cos\alpha \sqrt{K}. \tag{A56}$$

The synoptic vertical velocity at 700 hPa is calculated as:

$$w_{\mathrm{syn}} = c_8^{\mathrm{syn}} \sqrt{K}. \tag{A57}$$

The module of surface wind, which is used for calculation of the surface fluxes, is defined as:

$$U_{\mathrm{s}} = \sqrt{u_{\mathrm{s}}^2 + v_{\mathrm{s}}^2 + U_{\mathrm{syn}}^2}, \tag{A58}$$

and the wind stress over the ocean is computed as:

$$\tau_\lambda = C_{\mathrm{D}}\rho_0 u_{\mathrm{s}} U_{\mathrm{s}}, \tag{A59}$$

$$\tau_\phi = C_{\mathrm{D}}\rho_0 v_{\mathrm{s}} U_{\mathrm{s}}. \tag{A60}$$





## A6 Clouds

Total cloud cover fraction in SESAM is a combination of clouds related to atmospheric relative humidity and vertical velocity ($f_{\mathrm{cld}}^{\mathrm{r}}$) and clouds related to surface temperature inversion conditions ($f_{\mathrm{cld}}^{\mathrm{low}}$):

$$f_{\mathrm{cld}} = 1 - (1 - f_{\mathrm{cld}}^{\mathrm{r}}) \cdot \left(1 - f_{\mathrm{cld}}^{\mathrm{low}}\right). \tag{A61}$$

The relative humidity and vertical velocity mediated cloud fraction, which provides the main contribution to cloud cover in the model, is given by:

$$f_{\mathrm{cld}}^{\mathrm{r}} = \left(c_1^{\mathrm{cld}} + c_2^{\mathrm{cld}} \tanh\left(c_3^{\mathrm{cld}} w_{\mathrm{eff}}\right)\right) \cdot r_{\mathrm{a}}^{c_4^{\mathrm{cld}}}. \tag{A62}$$

It is therefore proportional to $r_{\mathrm{a}}^{c_4^{\mathrm{cld}}}$, with the proportionality factor depending on the effective vertical velocity at cloud level ($w_{\mathrm{eff}}$). In addition to the mean vertical velocity, $w_{\mathrm{eff}}$ also includes contributions from vertical velocities resulting from synoptic disturbances and sub-grid scale orography:

$$w_{\mathrm{eff}} = w(700\mathrm{hPa}) + c_{\mathrm{weff}} \cdot (w_{\mathrm{syn}} + w_{\mathrm{oro}}). \tag{A63}$$

The synoptic vertical velocity is given by eq. A57 while the orographic component is a function of surface wind speed and grid-cell standard deviation of surface elevation ($\sigma_{\mathrm{oro}}$):

$$w_{\mathrm{oro}} = c_{\mathrm{woro}} U_{\mathrm{s}} \sigma_{\mathrm{oro}}. \tag{A64}$$

The fraction of low clouds related to the presence of surface temperature inversion, when $r_\star > r_{\mathrm{a}}$, is defined as:

$$f_{\mathrm{cld}}^{\mathrm{low}} = c_5^{\mathrm{cld}} f_{\mathrm{freezedry}} \frac{(r_\star - r_{\mathrm{a}}) + c_6^{\mathrm{cld}}}{2c_6^{\mathrm{cld}}} \cdot r_{\mathrm{a}}^{c_4^{\mathrm{cld}}}. \tag{A65}$$

The factor $f_{\mathrm{freezedry}}$ represents the freezedry mechanism following Vavrus and Waliser (2008) and decreases the low cloud amount in very cold and dry conditions:

$$f_{\mathrm{freezedry}} = 0.1 + 0.9 \frac{q_{\mathrm{a}}}{c_7^{\mathrm{cld}}}, \tag{A66}$$

and is limited to the range $[0, 1]$.

The cloud base height is assumed to coincide with the top of the planetary boundary layer, $H_{\mathrm{pbl}}$. Cloud top height follows the height of the tropopause, modified by a factor depending on the mean vertical velocity at 700 hPa:

$$H_{\mathrm{cld}} = c_1^{\mathrm{hcld}} + c_2^{\mathrm{hcld}} H_{\mathrm{T}} \cdot \left(1 + c_3^{\mathrm{hcld}} w(700\mathrm{hPa})\right). \tag{A67}$$

Cloud optical thickness is parameterized as a function of surface air temperature, cloud fraction and column water content:

$$\tau_{\mathrm{cld}} = c_3^\tau \left[1 + \tanh\left(-\frac{T_{2\mathrm{m}} - T_0 - c_1^\tau}{c_2^\tau}\right)\right] (f_{\mathrm{cld}} Q_{\mathrm{q}})^{c_4^\tau}, \tag{A68}$$

and is further modified to account for the indirect effect of sulfate aerosols following Bauer et al. (2008).





**Table A5.** Cloud parameters.

| Parameter | value |
|---|---|
| *Cloud fraction* | |
| $c_1^{\text{cld}}$ | 0.47 |
| $c_2^{\text{cld}}$ | 0.5 |
| $c_3^{\text{cld}}$ | $200\,\text{s}\,\text{m}^{-1}$ |
| $c_4^{\text{cld}}$ | 1.5 |
| $c_5^{\text{cld}}$ | 0.5 |
| $c_6^{\text{cld}}$ | 0.1 |
| $c_7^{\text{cld}}$ | $0.003\,\text{kg}\,\text{kg}^{-1}$ |
| $c_{\text{weff}}$ | 0.25 |
| $c_{\text{woro}}$ | $1\times10^{-5}\,\text{m}^{-1}$ |
| *Cloud top height* | |
| $H_{\text{pbl}}$ | $1500\,\text{m}$ |
| $c_1^{\text{hcld}}$ | $2000\,\text{m}$ |
| $c_2^{\text{hcld}}$ | 0.27 |
| $c_3^{\text{hcld}}$ | $200\,\text{s}\,\text{m}^{-1}$ |
| *Cloud optical thickness* | |
| $c_1^{\tau}$ | $5\,\text{K}$ |
| $c_2^{\tau}$ | $30\,\text{K}$ |
| $c_3^{\tau}$ | 2 |
| $c_4^{\tau}$ | 0.5 |

**A7   Shortwave radiation**

The computation of shortwave radiation fluxes is based on a two-stream delta-Eddington approximation of the transport equa-
tion in a gas-aerosol atmosphere (see Petoukhov et al. (2003) for a more detailed description). The net downward shortwave
radiation fluxes are computed at the top of the atmosphere and at the surface as a weighted mean of clear sky (cs) and cloudy
(cld) fluxes:

$$\text{SW}_{\text{top}} = f_{\text{cld}}\text{SW}_{\text{top}}^{\text{cld}} + (1 - f_{\text{cld}})\text{SW}_{\text{top}}^{\text{cs}}, \tag{A69}$$

$$\text{SW}_{\text{sur}} = f_{\text{cld}}\text{SW}_{\text{sur}}^{\text{cld}} + (1 - f_{\text{cld}})\text{SW}_{\text{sur}}^{\text{cs}}. \tag{A70}$$




The clear sky and cloudy fluxes are computed separately for two spectral bands - visible (vu) and near infrared (ir), and for each macro surface type. The individual components of the shortwave radiation flux are computed as:

$$\mathrm{SW}_{\mathrm{top}}^{\mathrm{cs}} = \mathrm{SW}_{\mathrm{top}}^{\downarrow} - \mathrm{SW}_{\mathrm{top}}^{\downarrow}(f_{\mathrm{vu}}\alpha_{\mathrm{atm,vu}}^{\mathrm{cs}} + (1-f_{\mathrm{vu}})\alpha_{\mathrm{atm,ir}}^{\mathrm{cs}}), \tag{A71}$$

$$\mathrm{SW}_{\mathrm{top}}^{\mathrm{cld}} = \mathrm{SW}_{\mathrm{top}}^{\downarrow} - \mathrm{SW}_{\mathrm{top}}^{\downarrow}(f_{\mathrm{vu}}\alpha_{\mathrm{atm,vu}}^{\mathrm{cld}} + (1-f_{\mathrm{vu}})\alpha_{\mathrm{atm,ir}}^{\mathrm{cld}}), \tag{A72}$$

$$\mathrm{SW}_{\mathrm{sur}}^{\mathrm{cs}} = \mathrm{SW}_{\mathrm{top}}^{\downarrow}(f_{\mathrm{vu}}I_{\mathrm{atm,vu}}^{\mathrm{cs}} + (1-f_{\mathrm{vu}})I_{\mathrm{atm,ir}}^{\mathrm{cs}}), \tag{A73}$$

$$\mathrm{SW}_{\mathrm{sur}}^{\mathrm{cld}} = \mathrm{SW}_{\mathrm{top}}^{\downarrow}(f_{\mathrm{vu}}I_{\mathrm{atm,vu}}^{\mathrm{cld}} + (1-f_{\mathrm{vu}})I_{\mathrm{atm,ir}}^{\mathrm{cld}}), \tag{A74}$$

where $f_{\mathrm{vu}}$ is the fraction of solar radiation in the visible and ultraviolet spectral range, $\alpha_{\mathrm{atm}}$ is the planetary albedo and $I_{\mathrm{atm}}$ is the atmosphere integral transmission function. Planetary albedoes used to computed the net shortwave fluxes at TOA are defined as:

$$\alpha_{\mathrm{atm,vu}}^{\mathrm{cs}} = \left(\alpha_{\mathrm{sct,vu}} + \frac{(1-\alpha_{\mathrm{sct,vu}})^2\alpha_{\mathrm{sur,vu}}^{\mathrm{cs}}}{1-\alpha_{\mathrm{sct,vu}}\alpha_{\mathrm{sur,vu}}^{\mathrm{cs}}}\right)I_{\mathrm{wv,vu}}^{\mathrm{cs}}I_{\mathrm{aer,vu}}^{\mathrm{cs}}I_{\mathrm{O3,vu}}, \tag{A75}$$

$$\alpha_{\mathrm{atm,ir}}^{\mathrm{cs}} = \left(\alpha_{\mathrm{sct,ir}} + \frac{(1-\alpha_{\mathrm{sct,ir}})^2\alpha_{\mathrm{sur,ir}}^{\mathrm{cs}}}{1-\alpha_{\mathrm{sct,ir}}\alpha_{\mathrm{sur,ir}}^{\mathrm{cs}}}\right)I_{\mathrm{wv,ir}}^{\mathrm{cs}}I_{\mathrm{aer,ir}}^{\mathrm{cs}}I_{\mathrm{O3,ir}}, \tag{A76}$$

$$\alpha_{\mathrm{atm,vu}}^{\mathrm{cld}} = \left(\alpha_{\mathrm{cld,vu}} + \frac{(1-\alpha_{\mathrm{cld,vu}})^2\alpha_{\mathrm{sur,vu}}^{\mathrm{cld}}}{1-\alpha_{\mathrm{cld,vu}}\alpha_{\mathrm{sur,vu}}^{\mathrm{cld}}}\right)I_{\mathrm{wv,vu}}^{\mathrm{cld}}I_{\mathrm{aer,vu}}^{\mathrm{cld}}I_{\mathrm{O3,vu}}I_{\mathrm{cld,vu}}, \tag{A77}$$

$$\alpha_{\mathrm{atm,ir}}^{\mathrm{cld}} = \left(\alpha_{\mathrm{cld,ir}} + \frac{(1-\alpha_{\mathrm{cld,ir}})^2\alpha_{\mathrm{sur,ir}}^{\mathrm{cld}}}{1-\alpha_{\mathrm{cld,ir}}\alpha_{\mathrm{sur,ir}}^{\mathrm{cld}}}\right)I_{\mathrm{wv,ir}}^{\mathrm{cld}}I_{\mathrm{aer,ir}}^{\mathrm{cld}}I_{\mathrm{O3,ir}}I_{\mathrm{cld,ir}}. \tag{A78}$$

The atmospheric scattering albedo depends on the cosine of the solar zenith angle ($\mu$) and the optical thickness ($\tau_{\mathrm{aer}}$) and imaginary part of the refractive index ($R_{\mathrm{aer}}^{\mathrm{im}}$) of the aerosol load:

$$\alpha_{\mathrm{sct,vu}} = 1 - (1-r_{\mathrm{sct}})\exp(-\mu^{\mathrm{p1}}(0.55\tau_{\mathrm{aer}})^{\mathrm{p2}}(\alpha_1 - \alpha_2\log(1+\alpha_3 R_{\mathrm{aer}}^{\mathrm{im}}))), \tag{A79}$$

$$\alpha_{\mathrm{sct,ir}} = 1 - \exp(-\mu^{\mathrm{p1}}(0.55\tau_{\mathrm{aer}})^{\mathrm{p2}}(\alpha_1 - \alpha_2\log(1+\alpha_3 R_{\mathrm{aer}}^{\mathrm{im}}))). \tag{A80}$$

Cloud albedo is computed from cloud optical thickness as follows:

$$\alpha_{\mathrm{cld,vu}} = 1 - (1-\alpha_{\mathrm{sct,vu}})\exp(-g_{\mathrm{cld}}\frac{\tau_{\mathrm{cld}}^{p_4}}{\mu^{p_3}}), \tag{A81}$$

$$\alpha_{\mathrm{cld,ir}} = 1 - (1-\alpha_{\mathrm{sct,ir}})\exp(-g_{\mathrm{cld}}\frac{\tau_{\mathrm{cld}}^{p_4}}{\mu^{p_3}}). \tag{A82}$$



The atmospheric integral transmission functions used to derive the net shortwave fluxes at the surface are calculated as:

$$I_{\mathrm{atm,vu}}^{\mathrm{cs}} = (1 - \alpha_{\mathrm{sct,vu}})(1 - \alpha_{\mathrm{sur,vu}}^{\mathrm{cs}})I_{\mathrm{wv,vu}}^{\mathrm{cs,1}}I_{\mathrm{aer,vu}}^{\mathrm{cs,1}}I_{\mathrm{O3,vu}} + (1 - \alpha_{\mathrm{sct,vu}})\alpha_{\mathrm{sur,vu}}^{\mathrm{cs}}\alpha_{\mathrm{sct,vu}}^{0}\frac{1 - \alpha_{\mathrm{sur,vu}}^{\mathrm{cs}}}{1 - \alpha_{\mathrm{sct,vu}}^{0}\alpha_{\mathrm{sur,vu}}^{\mathrm{cs}}}I_{\mathrm{wv,vu}}^{\mathrm{cs,2}}I_{\mathrm{aer,vu}}^{\mathrm{cs,2}}I_{\mathrm{O3,vu}},$$
(A83)

$$I_{\mathrm{atm,ir}}^{\mathrm{cs}} = (1 - \alpha_{\mathrm{sct,ir}})(1 - \alpha_{\mathrm{sur,ir}}^{\mathrm{cs}})I_{\mathrm{wv,ir}}^{\mathrm{cs,1}}I_{\mathrm{aer,ir}}^{\mathrm{cs,1}}I_{\mathrm{O3,ir}} + (1 - \alpha_{\mathrm{sct,ir}})\alpha_{\mathrm{sur,ir}}^{\mathrm{cs}}\alpha_{\mathrm{sct,ir}}^{0}\frac{1 - \alpha_{\mathrm{sur,ir}}^{\mathrm{cs}}}{1 - \alpha_{\mathrm{sct,ir}}^{0}\alpha_{\mathrm{sur,ir}}^{\mathrm{cs}}}I_{\mathrm{wv,ir}}^{\mathrm{cs,2}}I_{\mathrm{aer,ir}}^{\mathrm{cs,2}}I_{\mathrm{O3,ir}},$$
(A84)

$$I_{\mathrm{atm,vu}}^{\mathrm{cld}} = (1 - \alpha_{\mathrm{cld,vu}})(1 - \alpha_{\mathrm{sur,vu}}^{\mathrm{cld}})I_{\mathrm{cld,vu}}^{1}I_{\mathrm{wv,vu}}^{\mathrm{cld,1}}I_{\mathrm{aer,vu}}^{\mathrm{cld,1}}I_{\mathrm{O3,vu}} + (1 - \alpha_{\mathrm{cld,vu}})\alpha_{\mathrm{sur,vu}}^{\mathrm{cld}}\alpha_{\mathrm{cld,vu}}\frac{1 - \alpha_{\mathrm{sur,vu}}^{\mathrm{cld}}}{1 - \alpha_{\mathrm{cld,vu}}\alpha_{\mathrm{sur,vu}}^{\mathrm{cld}}}I_{\mathrm{cld,vu}}^{2}I_{\mathrm{wv,vu}}^{\mathrm{cld,2}}I_{\mathrm{aer,vu}}^{\mathrm{cld,2}}I_{\mathrm{O3,vu}}$$
(A85)

$$I_{\mathrm{atm,ir}}^{\mathrm{cld}} = (1 - \alpha_{\mathrm{cld,ir}})(1 - \alpha_{\mathrm{sur,ir}}^{\mathrm{cld}})I_{\mathrm{cld,ir}}^{1}I_{\mathrm{wv,ir}}^{\mathrm{cld,1}}I_{\mathrm{aer,ir}}^{\mathrm{cld,1}}I_{\mathrm{O3,ir}} + (1 - \alpha_{\mathrm{cld,ir}})\alpha_{\mathrm{sur,ir}}^{\mathrm{cld}}\alpha_{\mathrm{cld,ir}}\frac{1 - \alpha_{\mathrm{sur,ir}}^{\mathrm{cld}}}{1 - \alpha_{\mathrm{cld,ir}}\alpha_{\mathrm{sur,ir}}^{\mathrm{cld}}}I_{\mathrm{cld,ir}}^{2}I_{\mathrm{wv,ir}}^{\mathrm{cld,2}}I_{\mathrm{aer,ir}}^{\mathrm{cld,2}}I_{\mathrm{O3,ir}}.$$
(A86)

The first terms on the right hand side represent the direct effect on radiation reaching the surface, while the second terms represent the effect of multiple reflections between the surface and atmospheric scatterers and clouds.

The integral transmission functions for water vapor, aerosols, ozone and clouds are given by:

$$I_{\mathrm{wv,uv}} = a_1^{\mathrm{wv}}\exp(-b_1^{\mathrm{wv}}M_{\mathrm{wv}}) + a_2^{\mathrm{wv}}\exp(-b_2^{\mathrm{wv}}M_{\mathrm{wv}}),$$
(A87)

$$I_{\mathrm{wv,ir}} = 1,$$
(A88)

$$I_{\mathrm{aer,vu}} = I_{\mathrm{aer,ir}} = \exp(-\gamma_1^{\mathrm{aer}}M_{\mathrm{aer}}(R_{\mathrm{aer}}^{\mathrm{im}})^{\gamma_2^{\mathrm{aer}}}),$$
(A89)

$$I_{\mathrm{O3,uv}} = 0.96,$$
(A90)

$$I_{\mathrm{O3,ir}} = 1,$$
(A91)

$$I_{\mathrm{cld,uv}} = 0.9,$$
(A92)

$$I_{\mathrm{cld,ir}} = 0.9,$$
(A93)



where $M_\mathrm{w}$ and $M_\mathrm{aer}$ are the effective absorber mass of water and aerosols in the column, respectively. In general, $M_\mathrm{w}$ and $M_\mathrm{aer}$ are different for clear sky and cloudy sky and for the top of the atmosphere and the surface:

$$M_\mathrm{w}^\mathrm{cs} = \left(\frac{1}{\mu} + \frac{1}{\mu_0}\right) W, \tag{A94}$$

$$M_\mathrm{w}^{\mathrm{cs},1} = \frac{1}{\mu} W, \tag{A95}$$

$$M_\mathrm{w}^{\mathrm{cs},2} = M_\mathrm{w}^{\mathrm{cs},1} + \left(1 - e^{-0.25}\right)\frac{2}{\mu_0} W, \tag{A96}$$

$$M_\mathrm{w}^\mathrm{cld} = e^{-H_\mathrm{cld}/H_\mathrm{q}}\left(\frac{1}{\mu} + \frac{1}{\mu_0} + \left(1 - e^{D_\mathrm{cld}/H_\mathrm{q}}\right)\right) W, \tag{A97}$$

$$M_\mathrm{w}^{\mathrm{cld},1} = \left[e^{-H_\mathrm{cld}/H_\mathrm{q}}\frac{1}{\mu} + f_\mathrm{exp}^1 + f_\mathrm{exp}^2\right] W, \tag{A98}$$

$$M_\mathrm{w}^{\mathrm{cld},2} = M_\mathrm{w}^{\mathrm{cld},1} + \left(f_\mathrm{exp}^1 + 2f_\mathrm{exp}^2\right) W, \tag{A99}$$

$$M_\mathrm{aer}^\mathrm{cs} = \left(\frac{1}{\mu} + \frac{1}{\mu_0}\right) 0.55\tau_\mathrm{aer}, \tag{A100}$$

$$M_\mathrm{aer}^{\mathrm{cs},1} = \frac{1}{\mu} 0.55\tau_\mathrm{aer}, \tag{A101}$$

$$M_\mathrm{aer}^{\mathrm{cs},2} = M_\mathrm{aer}^{\mathrm{cs},1} + \left(1 - e^{-0.25}\right)\frac{2}{\mu_0} 0.55\tau_\mathrm{aer}, \tag{A102}$$

$$M_\mathrm{aer}^\mathrm{cld} = e^{-H_\mathrm{cld}/H_\mathrm{q}}\left(\frac{1}{\mu} + \frac{1}{\mu_0} + \left(1 - e^{D_\mathrm{cld}/H_\mathrm{q}}\right)\right) 0.55\tau_\mathrm{aer}, \tag{A103}$$

$$M_\mathrm{aer}^{\mathrm{cld},1} = \left[e^{-H_\mathrm{cld}/H_\mathrm{q}}\frac{1}{\mu} + f_\mathrm{exp}^1 + f_\mathrm{exp}^2\right] 0.55\tau_\mathrm{aer}, \tag{A104}$$

$$M_\mathrm{aer}^{\mathrm{cld},2} = M_\mathrm{aer}^{\mathrm{cld},1} + \left(f_\mathrm{exp}^1 + 2f_\mathrm{exp}^2\right) 0.55\tau_\mathrm{aer}. \tag{A105}$$

$W$ is the column water content in $\mathrm{gcm}^{-2}$ and $\mu_0$ is the effective cosine of the zenith angle for diffuse radiation, $H_\mathrm{q}$ is the humidity height scale, $D_\mathrm{cld}$ is the geometrical thickness of clouds, $f_\mathrm{exp}^1 = \exp(-H_\mathrm{cld}/H_\mathrm{q}) - \exp(-(H_\mathrm{cld} + D_\mathrm{cld})/H_\mathrm{q})$ and $f_\mathrm{exp}^2 = 1 - \exp(-H_\mathrm{cld}/H_\mathrm{q})/\mu_0$.

### A8 Longwave radiation

Longwave radiative transfer follows in part from CLIMBER-2 as described in Petoukhov et al. (2003). Improvements relative to CLIMBER-2 include:

– relaxing the assumption that clouds act as black bodies for longwave radiation, which is not strictly true for thin clouds in cold regions,

– specific humidity profile derived from temperature and relative humidity, giving more consistent humidity/temperature values in the upper troposphere,

– $CH_4$, $N_2O$, CFC11 and CFC12 are considered through the use of an equivalent $CO_2$ concentration,





**Table A6.** Shortwave radiation parameters.

| Parameter | value |
|-----------|-------|
| $r_{\mathrm{sct}}$ | 0.17 |
| $g_{\mathrm{cld}}$ | 0.14 |
| $p_1$ | -1.97 |
| $p_2$ | 0.82 |
| $p_3$ | 0.35 |
| $p_4$ | 0.67 |
| $\alpha_1$ | $7.73 \times 10^{-2}$ |
| $\alpha_2$ | $2.39 \times 10^{-2}$ |
| $\alpha_3$ | $1.51 \times 10^{2}$ |
| $\gamma_1^{\mathrm{aer}}$ | 2.75 |
| $\gamma_2^{\mathrm{aer}}$ | 0.636 |
| $D_{\mathrm{cld}}$ | 1000 m |
| $a_1^{\mathrm{wv}}$ | 0.174 |
| $a_2^{\mathrm{wv}}$ | $1-a_1^{\mathrm{wv}}$ |
| $b_1^{\mathrm{wv}}$ | 6.27 |
| $b_2^{\mathrm{wv}}$ | 0.0267 |

– extensive retuning in offline setup using observations and output of atmospheric GCMs, including radiative kernels.

The computation of longwave radiative transfer is based on the two stream approximation, where the downward and upward longwave radiative fluxes at any height $z$ are computed as:

$$F_{\mathrm{LW}}^{\downarrow}(z) = B(z) - B(H_{\mathrm{TOA}})D(z, H_{\mathrm{TOA}}) + \int_{z}^{H_{\mathrm{TOA}}} D(z, z') \frac{dB(z')}{dz'} dz', \tag{A106}$$

$$F_{\mathrm{LW}}^{\uparrow}(z) = B(z) + (B_{\mathrm{s}} - B(z_{\mathrm{s}}))D(z_{\mathrm{s}}, z) - \int_{z_{\mathrm{s}}}^{z} D(z, z') \frac{dB(z')}{dz'} dz', \tag{A107}$$

where $B(z) = \sigma T^4(z)$ is the longwave emission at height $z$, $B_{\mathrm{s}} = \epsilon \sigma T_{\mathrm{s}}^4$ the longwave emitted by the surface with an emissivity $\epsilon$, $\sigma$ is the Stefan-Boltzmann constant, and $D(z, z')$ the integral transmission function of the atmospheric layer confined between levels $z$ and $z'$. Longwave radiation fluxes are computed separately for clear sky and cloudy sky using the following integral transmission functions, which explicitly include the effect of water vapor, $CO_2$, $O_3$ and clouds:

$$D^{\mathrm{cs}} = D_{\mathrm{wv}} D_{CO_2} D_{O_3}, \tag{A108}$$

$$D^{\mathrm{cld}} = D_{\mathrm{wv}} D_{CO_2} D_{O_3} D_{\mathrm{cld}}. \tag{A109}$$





The integral transmission functions for water vapor, $CO_2$, $O_3$ and clouds are given by:

$$D_{\text{wv}}(z_1, z_2) = \left(1 + a_1^{\text{wv}}(\beta_0 M_{\text{wv}}(z_1, z_2))^{\beta_1^{\text{wv}}} + a_2^{\text{wv}}(\beta_0 M_{\text{wv}}(z_1, z_2))^{\beta_2^{\text{wv}}} + a_3^{\text{wv}}(\beta_0 M_{\text{wv}}(z_1, z_2))^{\beta_3^{\text{wv}}}\right)^{-1}, \tag{A110}$$

$$D_{\text{CO}_2}(z_1, z_2) = \left(1 - 0.1\left(\frac{M_{\text{CO}_2}}{1000}\right)^2\right) \frac{1 + a_0^{\text{CO}_2} a_1^{\text{CO}_2}(\beta_0 M_{\text{CO}_2}(z_1, z_2))^{\beta_1^{\text{CO}_2}}}{1 + a_0^{\text{CO}_2}(\beta_0 M_{\text{CO}_2}(z_1, z_2))^{\beta_2^{\text{CO}_2}}}, \tag{A111}$$

$$D_{\text{O}_3}(z_1, z_2) = 1 - a^{\text{O}_3}(M_{\text{O}_3}(z_1, z_2))^{\beta^{\text{O}_3}}, \tag{A112}$$

$$D_{\text{cld}}(z_1, z_2) = \exp\left(-\frac{|z - z'|}{H_{\text{cld}}^{\text{top}} - H_{\text{cld}}^{\text{base}}}\tau_{\text{cld}}\right) \quad \text{only inside cloud layers.} \tag{A113}$$

The first term on the right-hand side of eq. A111 ensures that the radiative forcing of $CO_2$ increases with increasing $CO_2$, in accordance with Hansen (2005) and Colman and McAvaney (2009). The effective absorber mass of water vapor, $CO_2$ and $O_3$ is computed from:

$$M_{\text{wv}}(z_1, z_2) = \int_{z_1}^{z_2} q(z)\rho(z)\left(\frac{p(z)}{p(0)}\right)^{k_{\text{wv}}} dz, \tag{A114}$$

$$M_{\text{CO}_2}(z_1, z_2) = CO_2 \int_{z_1}^{z_2} \rho(z)\left(\frac{p(z)}{p(0)}\right)^{k_{\text{CO}_2}} dz, \tag{A115}$$

$$M_{\text{O}_3}(z_1, z_2) = \int_{z_1}^{z_2} O_3(z)\rho(z)\left(\frac{p(z)}{p(0)}\right)^{k_{\text{O}_3}} dz. \tag{A116}$$

The effect of the well-mixed greenhouse gases $CH_4$, $N_2O$ and CFCs on longwave radiation is accounted for through a $CO_2$ equivalent following Etminan et al. (2016):

$$CO_2^{\text{eq}} = CO_2^0 \exp\left(\frac{R(CO_2) + R(CH_4) + R(N_2O) + R(CFC11) + R(CFC12)}{a_1\left(CO_2 - CO_2^0\right)^2 + b_1\left|CO_2 - CO_2^0\right| + c_1\overline{N_2O} + 5.36}\right) \tag{A117}$$

The radiative forcing $R$ of $CO_2$, $CH_4$ and $N_2O$ is computed as in Table 1 of Etminan et al. (2016) and the radiative forcing of CFC11 and CFC12 as in Table 3 of Myhre et al. (1998).

Similarly to shortwave radiation, longwave radiation is computed for each macro surface type individually. Atmospheric characteristics for each type within a given grid cell are the same but surface elevation and surface temperature are different. By default, for longwave radiation the atmosphere is divided into 15 vertical levels: 6 between surface and base of clouds, 3 between the base and top of clouds, 3 up to the tropopause and 3 in the stratosphere. Total longwave radiation fluxes in each grid cell are taken as cloud fraction weighted sum of clear sky and cloudy sky fluxes.

Model parameters are tuned in offline mode using ERA-Interim reanalysis monthly climatological 3D fields of temperature and humidity in order to minimize the root mean square error in top of the atmosphere and surface longwave radiation fluxes for clear sky. Additional constraints on model parameters are provided by the total greenhouse effect separation presented in Schmidt et al. (2010) and by the radiative kernels of Shell et al. (2008); Block and Mauritsen (2013); Smith et al. (2018); Pendergrass et al. (2018).





**Table A7.** Longwave radiation parameters.

| Parameter | value |
|---|---|
| $\beta_0$ | 1.66 |
| $a_1^{\mathrm{wv}}$ | 1.5 |
| $a_2^{\mathrm{wv}}$ | 0.1 |
| $a_3^{\mathrm{wv}}$ | 0.01 |
| $\beta_1^{\mathrm{wv}}$ | 0.42 |
| $\beta_2^{\mathrm{wv}}$ | 1.5 |
| $\beta_3^{\mathrm{wv}}$ | 3 |
| $k_{\mathrm{wv}}$ | 1 |
| $a_0^{\mathrm{CO_2}}$ | 0.247 |
| $a_1^{\mathrm{CO_2}}$ | 0.755 |
| $\beta^{\mathrm{CO_2}}$ | 0.45 |
| $k_{\mathrm{CO_2}}$ | 0.8 |
| $a_0^{\mathrm{O_3}}$ | 8.246 |
| $\beta^{\mathrm{O_3}}$ | 0.539 |
| $k_{\mathrm{O_3}}$ | 0.6 |

**Appendix B: GOLDSTEIN detailed description**

The horizontal ocean velocity in GOLDSTEIN is diagnosed from a frictional-geostrophic balance:

$$-fv + \mu u - \frac{1}{\rho_0}\frac{\partial \tau_\lambda}{\partial z} = -\frac{1}{\rho_0 R_e \cos\phi}\frac{\partial p}{\partial \lambda} \tag{B1}$$

$$fu + \mu v - \frac{1}{\rho_0}\frac{\partial \tau_\phi}{\partial z} = -\frac{1}{\rho_0 R_e}\frac{\partial p}{\partial \phi}, \tag{B2}$$


The drag coefficient $\mu$ is composed of a uniform background value $\mu_0$, which, for barotropic velocities, is enhanced by a factor three in waters shallower than $1000\,\mathrm{m}$. The definition of the wind or sea–ice stress $\tau$ is given in Appendix A5 above. Hydrostatic balance is assumed:

$$\frac{\partial p}{\partial z} = -g\rho(\theta, S, p_0(z)), \tag{B3}$$

where $p_0(z) = 0.1 \cdot z$. Seawater density, $\rho(\theta, S, p_0(z))$, is computed using the UNESCO equation of state of Millero and Poisson (1981) with the bulk secant modulus expressed in terms of potential temperature using the coefficients of Jackett and Mcdougall (1995). For computational efficiency only 7 of the originally 26 terms are kept in the bulk secant modulus polynomial. A comparison between the full equation of state of Jackett and Mcdougall (1995) and the truncated version showed negligible





differences in the simulated ocean state. The vertical velocity is then derived from the continuity equation:

$\qquad \nabla \cdot \boldsymbol{u} = 0.$ (B4)

Equations B1 through B4 are solved by separation of the velocity into a barotropic and a baroclinic component, as described in detail in Edwards et al. (1998) an Müller et al. (2006). More details on the treatment of barotropic flow around islands are given in Edwards and Shepherd (2002).

The transport equation for tracers takes the following form:

$\qquad \dfrac{\partial X}{\partial t} + \nabla \cdot (\boldsymbol{u}X) = \nabla(\boldsymbol{A}\nabla X) + \mathrm{SMS}.$ (B5)

$X$ is the tracer concentration, SMS represents source-minus-sinks and $A$ is the diffusive mixing tensor:

$$A = \begin{pmatrix} K_{\mathrm{I}} & 0 & (K_{\mathrm{I}} - \kappa)S_{\lambda} \\ 0 & K_{\mathrm{I}} & (K_{\mathrm{I}} - \kappa)S_{\phi} \\ (K_{\mathrm{I}} + \kappa)S_{\lambda} & (K_{\mathrm{I}} + \kappa)S_{\phi} & K_{\mathrm{I}}\boldsymbol{S}^2 + K_{\mathrm{D}} \end{pmatrix},$$ (B6)

with $K_{\mathrm{I}}$ and $K_{\mathrm{D}}$ the isopycnal and diapycnal diffusivities, respectively, $\kappa$ the Gent-McWilliams parameter which parametrizes sub-scale eddies and the slope of the isopycnals is:

$\qquad \boldsymbol{S} = (S_{\lambda}, S_{\phi}, 0) = \left( -\dfrac{\frac{1}{R_{\mathrm{e}}\cos\phi}\frac{\partial \rho}{\partial \lambda}}{\frac{\partial \rho}{\partial z}}, -\dfrac{\frac{1}{R_{\mathrm{e}}}\frac{\partial \rho}{\partial \phi}}{\frac{\partial \rho}{\partial z}}, 0 \right).$ (B7)

Diapycnal diffusivity is a prescribed function of depth:

$$K_{\mathrm{D}} = K_{\mathrm{D}}^{\min} + \dfrac{\arctan\left(\frac{z - z_{\mathrm{ref}}}{1000}\right) - \arctan\left(\frac{-z_{\mathrm{ref}}}{1000}\right)}{\arctan\left(\frac{5000 - z_{\mathrm{ref}}}{1000}\right) - \arctan\left(\frac{-z_{\mathrm{ref}}}{1000}\right)} \cdot \left( K_{\mathrm{D}}^{\max} - K_{\mathrm{D}}^{\min} \right).$$ (B8)

If the stratification of a water column is statically unstable, convective adjustment is applied using the scheme of Rahmstorf (1993). A simple mixed layer scheme, based on Kraus and Turner (1967) is used in the model.

$\qquad$ The equations are discretized on a staggered Arakawa C grid.

## Appendix C: SISIM detailed description

Sea ice thermodynamics is based on the zero-layer model of Semtner (1976). The surface energy fluxes are computed separately over sea ice and over open ocean. Over sea ice, the surface energy balance equation is written as:

$(1 - \alpha)\mathrm{SW}^{\downarrow} + \epsilon\mathrm{LW}^{\downarrow} - \mathrm{LW}^{\uparrow} - \mathrm{SH} - \mathrm{LE} - G = 0,$ (C1)

$\qquad$ where $\alpha$ is surface albedo, $\mathrm{SW}^{\downarrow}$ is the incoming shortwave radiation, $\epsilon$ is the surface emissivity for longwave radiation, $\mathrm{LW}^{\downarrow}$ and $\mathrm{LW}^{\uparrow}$ are the incoming and outgoing longwave radiation at the surface, $SH$ is the sensible heat flux, LE is the latent heat





**Table B1.** Ocean model parameters.

| Parameter | value |
| --- | --- |
| $K_{\mathrm{I}}$ | $1500\,\mathrm{m^2s^{-1}}$ |
| $\kappa$ | $1500\,\mathrm{m^2s^{-1}}$ |
| $K_{\mathrm{D}}^{\min}$ | $1\times10^{-5}\,\mathrm{m^2s^{-1}}$ |
| $K_{\mathrm{D}}^{\max}$ | $1.5\times10^{-4}\,\mathrm{m^2s^{-1}}$ |
| $z_{\mathrm{ref}}$ | $1000\,\mathrm{m}$ |
| $\mu_0$ | $4\,\mathrm{days^{-1}}$ |

flux and $G$ the heat flux into the snow/ice. Equation (C1) is then solved for the skin temperature, $T_\star$, using the formulations for the energy fluxes described next. Bare sea ice albedo is temperature dependent following:

$$\alpha_{\mathrm{vu}} = 0.8 - 0.075\,(T_\star - T_0 + 1), \tag{C2}$$

$$\alpha_{\mathrm{ir}} = 0.6 - 0.075\,(T_\star - T_0 + 1). \tag{C3}$$

The snow albedo scheme is the same as used in the land model and includes a dependence on snow grain size and dust and soot concentration following Dang et al. (2015). The fraction of sea ice covered by snow is computed from snow thickness as:

$$f_{\mathrm{snow}} = \frac{h_{\mathrm{snow}}}{h_{\mathrm{snow}} + 0.02}. \tag{C4}$$

The surface emitted longwave radiation is given by the Stefan-Boltzmann law:

$$\mathrm{LW}^\uparrow = \epsilon\sigma T_\star^4. \tag{C5}$$

The sensible heat flux is computed from the temperature gradient between the skin and near–surface air, using the bulk aerodynamic formula:

$$\mathrm{SH} = \frac{\rho_{\mathrm{a}} c_{\mathrm{p}}}{r_{\mathrm{aer}}}(T_\star - T_{\mathrm{2m}}), \tag{C6}$$

where $\rho_{\mathrm{a}}$ is air density, $c_{\mathrm{p}}$ is the specific heat of air, $r_{\mathrm{aer}}$ is the aerodynamic resistance and $T_{\mathrm{2m}}$ is the near–surface air
temperature. Similarly, the latent heat flux over sea ice is expressed in terms of the specific humidity gradient between the surface and near–surface air:

$$\mathrm{LE} = L\frac{\rho_{\mathrm{a}}}{r_{\mathrm{aer}}}\left(q_{\mathrm{sat}}(T_\star) - q_{\mathrm{2m}}\right). \tag{C7}$$

$L$ is the latent heat of sublimation, $q_{\mathrm{sat}}$ is the specific humidity at saturation and $q_{\mathrm{2m}}$ is the specific humidity of near–surface air. The aerodynamic resistance is computed from wind speed, surface exchange coefficient and bulk Richardson number following
Willeit and Ganopolski (2016).





The conductive heat flux into the snow/ice ($G$) is computed as:

$$G = \lambda_{\text{eff}} \left( T_\star - T_{\text{f}} \right), \tag{C8}$$

where $T_f$ is the salinity–dependent freezing temperature at the base of the ice (Millero and Poisson, 1981):

$$T_f = -0.0575 \cdot S_{\text{o}} + 0.0017 \cdot S_{\text{o}}^{1.5} - 0.0002 \cdot S_{\text{o}}^2 + T_0 \tag{C9}$$

and the effective snow/ice heat conductivity is:

$$\lambda_{\text{eff}} = k_\lambda \frac{\lambda_{\text{snow}} \cdot}{h_{\text{snow}} + (h_{\text{ice}}) \lambda_{\text{snow}}/\lambda_{\text{ice}}}. \tag{C10}$$

The $k_\lambda$ factor accounts for sub-grid ice thickness distribution following eq. 2 in (Fichefet and Maqueda, 1997). The prognostic terms in $T_\star$ in the formulation of the surface energy fluxes are then linearized using Taylor series expansion assuming that the temperature at the new time step, $T_{\star,n+1} = T_{\star,n} + \Delta T$ with $\Delta T_\star \ll T_\star$:

$$T_{\star,n+1}^4 = T_{\star,n}^4 + 4T_{\star,n}^3(T_{\star,n+1} - T_{\star,n}), \tag{C11}$$

$$q_{\text{sat}}(T_{\star,n+1}) = q_{\text{sat}}(T_\star) + \frac{dq_{\text{sat}}}{dT_\star}\bigg|_{T_\star = T_{\star,n}} (T_{\star,n+1} - T_{\star,n}). \tag{C12}$$

Equation (C1) can then be solved explicitly for the skin temperature at the new time step, $T_{\star,n+1}$. If the skin temperature is above freezing the surface energy fluxes are diagnosed first with the skin temperature greater then $0\,^\circ$C and then with skin
temperature set to $0\,^\circ$C. The difference between the sum of the energy fluxes is then used to melt snow and/or ice.

Whether bottom ice accretion or ablation occurs depends on the sign of the net energy flux at the ice/water interface, which is determined by the balance between the conductive heat flux through the snow/ice layer ($G$) and the turbulent heat flux between the ice and the seawater below (McPhee, 1992; Weaver et al., 2001):

$$H = C_{\text{h}} u_\star \rho_{\text{w}} c_{\text{w}} \cdot (T_{\text{f}} - T_{\text{o}}). \tag{C13}$$

$C_{\text{h}}$ is a constant exchange coefficient, $u_\star$ the (constant) friction velocity, $\rho_{\text{w}}$ water density and $c_{\text{w}}$ the specific heat capacity of water.

Over ocean water, the heat flux into the ocean is derived as the residual of the radiation and surface energy fluxes, which are computed similarly to the surface energy fluxes over sea ice, but using sea surface temperature instead of skin temperature. The albedo of ocean water for diffuse radiation is 0.06, while the clear–sky albedo includes a dependence on the cosine of the
solar zenith angle ($\mu$):

$$\alpha = \min \left( 0.2, \frac{0.03}{\mu} \right). \tag{C14}$$

The longwave emissivity of water is set to 0.98. Sea ice in leads forms whenever the top layer ocean temperature drops below the freezing point of seawater.





Subgrid scale thermodynamic processes of sea ice growth and melt are assumed to affect the sea ice concentration ($f_{\text{ice}}$)

within a grid cell following Marsland et al. (2003) and Fichefet and Maqueda (1997):

$$\frac{\partial f_{\text{ice}}}{\partial t} = \Delta f_{\text{ice}}|_{\text{frz}} + \Delta f_{\text{ice}}|_{\text{melt}}. \tag{C15}$$

When freezing occurs over open water the sea ice concentration increases at a rate given by:

$$\Delta f_{\text{ice}}|_{\text{frz}} = \sqrt{1 - f_{\text{ice}}^2} \cdot (1 - f_{\text{ice}}) \frac{\Delta h_{\text{ice}}|_{\text{frz}}}{h_0}, \tag{C16}$$

where $\Delta h_{\text{ice}}|_{\text{frz}}$ is the thickness of new sea ice formed and $h_0$ is an arbitrary demarcation thickness. Melting of sea ice leads

to a decrease in sea ice concentration:

$$\Delta f_{\text{ice}}|_{\text{melt}} = f_{\text{ice}} \frac{\Delta h_{\text{ice}}|_{\text{melt}}}{2 h_{\text{ice}}}, \tag{C17}$$

where $\Delta h_{\text{ice}}|_{\text{melt}}$ is the change in sea ice thickness due to melting. This formulation is based on the assumption that sea ice thickness within a grid cell has a uniform distribution between 0 and $2h_{\text{ice}}$.

Sea ice drift velocities are computed from the momentum balance equation (e.g. Hibler, 1979):

$$m \frac{\partial \boldsymbol{u}_{\text{i}}}{\partial t} = \nabla \cdot \sigma + \boldsymbol{\tau}_{\text{a}} + \boldsymbol{\tau}_{\text{o}} - \hat{\boldsymbol{k}} \times m f \boldsymbol{u}_{\text{i}} - m g \nabla H_{\text{o}}, \tag{C18}$$

where $\boldsymbol{u}_{\text{i}} = (u_{\text{i}}, v_{\text{i}})$ is the sea ice velocity vector, $m$ is the mass of ice and snow per unit area, $f$ is the Coriolis parameter, $H_{\text{o}}$ is the sea surface height and $\tau_{\text{a}}$ and $\tau_{\text{o}}$ are the wind and ocean stresses, respectively. The mechanical properties of the ice are represented by the internal stress tensor $\sigma$, and the elastic-viscous-plastic rheology of Hunke and Dukowicz (1997); Bouillon et al. (2009) is used as constitutive law relating the internal stress to the strain rate. The numerical solution follows

the implementation in the GFDL model SIS2 (Adcroft et al., 2019; Delworth et al., 2006). The sea surface height is diagnosed to a first approximation from the ocean water density above a reference depth of $1500\,\text{m}$. The zonal and meridional components of the wind stress on sea ice are computed as:

$$\tau_{\text{a}}^{\text{u}} = C_{\text{d}} \rho_{\text{a}} u_{\text{s}} U_{\text{s}}, \tag{C19}$$

$$\tau_{\text{a}}^{\text{v}} = C_{\text{d}} \rho_{\text{a}} v_{\text{s}} U_{\text{s}}, \tag{C20}$$

with the symbols defined in Appendix A2 above. The components of the stress exerted by the ocean on sea ice are given by:

$$\tau_{\text{o}}^{\text{u}} = C_{\text{d}}^{\text{w}} \rho_{\text{w}} \sqrt{(u_{\text{i}} - u_{\text{o}})^2 + (v_{\text{i}} - v_{\text{o}})^2} \cdot (u_{\text{i}} - u_{\text{o}}), \tag{C21}$$

$$\tau_{\text{o}}^{\text{v}} = C_{\text{d}}^{\text{w}} \rho_{\text{w}} \sqrt{(u_{\text{i}} - u_{\text{o}})^2 + (v_{\text{i}} - v_{\text{o}})^2} \cdot (v_{\text{i}} - v_{\text{o}}), \tag{C22}$$

where $C_{\text{d}}^{\text{w}}$ is the drag coefficient between the sea ice and water and $(u_{\text{o}}, v_{\text{o}})$ the top–layer ocean velocity. The derived sea ice velocities are then used to advect the grid-cell mean ice and snow thicknesses and the sea ice concentration using a flux-

corrected transport scheme (Zalesak, 1979). No explicit diffusion is applied.



**Table C1.** Sea ice model parameters.

| Parameter | value |
|---|---|
| *Thermodynamics* | |
| $\epsilon$ | 0.99 |
| $\lambda_{\mathrm{snow}}$ | $0.3\,\mathrm{Wm^{-1}K^{-1}}$ |
| $\lambda_{\mathrm{ice}}$ | $2.2\,\mathrm{Wm^{-1}K^{-1}}$ |
| $C_{\mathrm{h}}$ | 0.0058 |
| $u_{\star}$ | $0.01\,\mathrm{ms^{-1}}$ |
| $h_0$ | $0.5\,\mathrm{m}$ |
| *Dynamics* | |
| $C_{\mathrm{d}}^{\mathrm{w}}$ | $3.24{\times}10^{-3}$ |

*Author contributions.* A.G. and M.W. conceived the model. A.G. developed the atmosphere model SESAM, with contributions from M.W. N.E. provided the GOLDSTEIN model code and supported the implementation of the ocean model. M.W. developed the sea ice model SISIM. M.W. coupled the different model components and tuned and tested the model. A.R. contributed to the model design and provided tools for model I/O, mapping/interpolation and ensemble generation. M.W. performed the model simulations, prepared the figures and wrote
the paper, with contributions from all authors.

*Competing interests.* The authors declare that they have no conflict of interest.

*Acknowledgements.* M.W. acknowledges support by the German Science Foundation (DFG) grant GA 1202/2-1 and by the BMBF-funded project PalMod. A. R. was funded by the Ramón y Cajal Programme of the Spanish Ministry for Science, Innovation and Universities (grant no. RYC-2016-20587). We acknowledge the World Climate Research Programme, which, through its Working Group on Coupled
Modelling, coordinated and promoted CMIP5 and CMIP6. We thank the climate modeling groups for producing and making available their model output, the Earth System Grid Federation (ESGF) for archiving the data and providing access, and the multiple funding agencies who support CMIP5, CMIP6 and ESGF. Xavier Fettweiss is acknowledged for providing the grain-size data from simulations of the regional climate model MAR. Data from the RAPID AMOC monitoring project is funded by the Natural Environment Research Council and are freely available from www.rapid.ac.uk/rapidmoc.



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
