# Peer review of "The Earth system model CLIMBER-X v1.0. Part 1: climate model description and validation"

_Geoscientific Model Development, 2022_

## Author Comment (AC1)

**Response to reviewer #1**

We would like to thank the reviewer for the constructive comments on our paper. In blue below is our response to the reviewer comments and suggestions (in black).

The paper presents a comprehensive description of the EMIC CLIMBER-X. This description is well balanced with a main text presenting the important principles and choices at the basis of model development as well as the model performance while the equations and parameters are given in the appendices. The model is designed as the successor of a very successful model, CLIMBER-2, which has been used extensively over the past 20 years. CLIMBER-X will very likely be as successful and this model description will thus be very useful for the scientists running the model and analyzing its results.

The paper is very clear and well written. It includes all the key elements needed for the description of a new model. I thus have mainly some minor suggestions of clarification. The only point that would deserve more discussion is the model tuning or calibration. Tuning is mentioned in the appendices and once in the main text for specific components as well as in the author's contribution but without much details or a global view of the way tuning was achieved. I think it would be nice to have a specific section, for example as a section 2.5, to explain the tuning strategy. I guess some parameters have defaults, fixed values that are not supposed to be modified. Some parameters in the various modules were tuned specifically 'offline' for those modules (e.g., lines 281, 820, 852) while some others may have been tuned so that the full model fits with observations. It would be interesting to describe which are the main parameters that were tuned, what were the calibration targets for this general tuning (for example global mean temperature, total amount of precipitation, sea ice extent ,temperature trend over the past century, etc.) and the method followed for the tuning.

We would like to thank the reviewer for the positive comments on our paper.
Tuning is certainly a fundamental part of any model development and we will follow the suggestion by the reviewer (and reviewer #3) and add a subsection where we give more details about the model tuning strategy.

Minor points

Lines 7-8. I would be more specific in the abstract on the performance of the model, giving some examples where the agreement with observations is good and mention the limitations, such as the poor performance in the tropics.

We will add a sentence about model limitations to the abstract.

Line 101. I would suggest to state at this stage that the two dimensions are latitude and longitude. If I follow well, all the vertical variations are then prognostic only. If it is the case, that would be good to mention it explicitly here. It is also probably worth insisting on this point for comparisons with observations including vertical variations (like figures 3, 6 and 8).

In the revised manuscript, we will explicitly state that the 2 dimensions are latitude and longitude and that the vertical variations are *diagnostic* only. We will insist on this point also when discussing the figures on the 3D atmospheric structure.

Sulphate aerosols are mentionned first line 159 but the spatial distribution is discussed line 166. This may give the feeling that this distribution is only valid for longwave.

We will modify this. Where sulfates are mentioned for the first time in the paper, we will specify that the spatial distribution of sulfate aerosols has to be prescribed in the model.

Line 171. It would be useful to justify in one or two sentences why an approach even simpler that the one in CLIMBER- 2 has been retained.

We did not find any advantages in a separate treatment of stratus and cumulus clouds in the framework of our modelling approach. We will add this sentence to the revised manuscript.

Line 200. What is exactly meant by 'lack of a Gulf Stream extension propagating to high latitudes'? Does it have an impact on model biases, on the location of deep-water formation or a link with the AMOC simulated by the model?

We will rephrase this to: "The frictional-geostrophic dynamics strongly damp momentum and are characterised by weak gyre circulation and associated momentum-driven effects including generally weak ACC and Gulf Stream as a result."

Line 250. Are the fluxes in the open ocean computed in the same way as in the ice-free fraction of the grid cells?

Yes, the open ocean fluxes are computed the same way as in the ice-free fraction of the partly ice-covered grid cells. We will make this clearer in the revised manuscript.

Line 254. Are the latent and sensible heat fluxes computed in the same way over the ocean? More generally, the computation of the surface balance is well explained here for SISIM in section 2.3 but not explicitly for the oceanic part in section 2.2.

The latent and sensible heat fluxes are computed the same way over the ocean, but since the air-sea fluxes are computed in the sea ice model (even in the absence of sea ice), they are described as part of the sea ice model. In this respect, SISIM is not only sea ice model but also atmosphere-ocean interface. We agree that this might be confusing to the reader and will clarify this point in the revised paper.

Line 276. This would be useful to add here a few sentences describing the main characteristics of the model without the need to read Willeit and Ganopolski (2016).

We will add a few sentences here describing the main features of PALADYN, to give the reader a short overview of the model without having to read the separate PALADYN description paper.

Line 378-380. Is the deep water formed in Antarctica above the continental shelf or because of open ocean convection?

Deep water around Antarctica is formed on the continental shelf and not through open ocean convection. We will add this information to the revised manuscript.

Caption of Figure 28. The reference to Paul et al. 2021 is not the same as on the figure (GLOMAP).

We will change the corresponding legend entry in the figure from GLOMAP to Paul 2021.

---

## Author Comment (AC2)

**Response to reviewer #2**

We would like to thank the reviewer for the comments on our paper. In blue below is our response to the reviewer comments and suggestions (in black).

Paper provides detail description of new version of CLIMBER model and evaluation of model performance for different historical periods, as well as model response to the changes in atmospheric $CO_2$ concertation. Present day climate, climate change over resent historical period (1850-2100) and response to both doubling and transient increase of the atmospheric $CO_2$ concentration simulated by CLIMBER agree well with available observations and results obtained with more sophisticated climate modes. At the same time, there are noticeable differences in climate state for Last Glacial Maximum between CLIMBER and CMIP5 models.

This does not necessarily mean that CLIMBER is wrong, but in my view, additional validation would be useful. As said in the paper "The atmospheric component of CLIMBER X is based on a statistical-dynamical approach, which employs a number of significant simplifications and assumptions." The fact that "these simplifications and a set of parameterizations explicitly derived from present–day climate limit the models' applicability to climate states fundamentally different from the present one". Authors mention Snowball Earth case as such climate state. Possibly Last Glacial Maximum climate is also different enough from preset one.

As noted by the reviewer, there are indeed some differences in the simulated LGM climate between CLIMBER-X and CMIP models. Part of the differences could be a result of CLIMBER-X having dynamic vegetation, while many CMIP models don't. We will test the effect of the vegetation feedback by running an additional LGM simulation with prescribed present-day vegetation. We would also argue that when it comes to e.g. global cooling and AMOC state, CLIMBER-X results are possibly in better agreement with reconstructions than (some) CMIP models.
The LGM is definitely not outside of the applicability range of the model, as CLIMBER-X has been designed specifically also to simulate past glacial cycles. That is also the reason why we explicitly included a comparison of the simulated LGM state with state-of-the-art CMIP models in the paper. However, we will elaborate a bit more on the differences between CLIMBER-X and CMIP models in the revised paper version.

As indicated in the paper "CLIMBER-X includes code to diagnose the strength of the different climate feedbacks". I would suggest calculating climate feedbacks for LGM climate and comparing them with estimates available in literature.

We will perform a further feedback analysis for low CO2, to get a better picture of the state dependence of the climate feedbacks in the model.

Specific comments.

There seems to be discrepancy between Table 1 and Table 2.

Total evaporation in Table 2 are larger than observation, while latent heat in table 1 is smaller

This could be a result of the use of different observation-based estimates for the energy fluxes and for the hydrological cycle and thus ultimately a consequence of the uncertainty in these estimates. We will add a short discussion on this point in the revised manuscript.

On page 14 (near line 355).

Following sentence: "During the winter months and at high latitudes, CLIMBER-X also captures the near-surface temperature inversions." It is better to say, "During the winter months, CLIMBER-X also captures the near-surface temperature inversions at high latitudes"

We will change the mentioned sentence accordingly.

---

## Author Comment (AC3)

**Response to reviewer #3**

We would like to thank the reviewer for the comments on our paper. In blue below is our response to the reviewer comments and suggestions (in black).

This paper serves as an introduction to the CLIMBER-X Earth system model. I found it to be extremely well written, and therefore pleasant to read. It describes in detail all of the key components of the model and how they interact, and then follows this with an evaluation of the model performance over the period 1850-present, which most readers will easily be able to relate to due to the authors' efforts to compare their simulation with both observations and the CMIP simulations and assessments.

CLIMBER-X itself performs well in almost every aspect shown, and the authors also make clear the model's deficiencies and limitations, which is crucial for potential users with applications in mind. As a paper intended to present a new model to the community, I think it serves that purpose well.

We would like to thank the reviewer for the positive appraisal of our paper.

Minor comment:

I think it would be of interest to the reader to know what degree of calibration has been done, and how. There are a couple of comments in the text, but I think the 'dark art' of how we tune our numerical models should be made more visible (see e.g. Mauritsen et al., 2012 for a good example). In this case, where it is clear that CLIMBER-X is well suited to methods of performing robust exploration of parameter space (see e.g. Williamson et al., 2013), it would be interesting to know the extent to which this has been done, or remains to be done. I suggest including a section (or possibly a subsection in 2), that describes the calibration (tuning) that has been done to the model. For example, were components, or schemes within the individual models tuned in isolation, or was the entirety of CLIMBER-X calibrated as one? To what observations or metrics was the model tuned to?

As mentioned already in the response to Reviewer #1, who raised a similar point, we will follow the suggestion and add a subsection where we give more details about the model tuning strategy.

Specific comments/corrections:

Line 77/78: Replace "16 CPUs" with "2 x 8 core CPUs". Also, if your computation is on a single node then the detail "Infiniband FDR14 Lenovo/IBM" is redundant and can be removed.

We will change this as suggested.

Line 96: I suggest "We made extensive use of…" instead of "We extensively made use of…"

We will change this as suggested.

Line 162: Seem to jump from Appendix 5 (line 153) to Appendix 7. I suggest carefully checking and renumbering/reordering as required.

We will shift the description of cloud parameterisations up, before the paragraphs dedicated to radiation. This will fix the order in which the different Appendix section are referenced to in the main text.

Line 233: question about rigid lid pressure?

The rigid-lid formulation does imply a surface pressure, the so-called rigid-lid pressure, which is directly related to the sea surface height. However, instead of solving the non-trivial equation for surface pressure, in the model we use an approximate approach in which surface pressure is simply diagnosed from integrating density above a reference depth. We will clarify this in the revised manuscript.

Line 269: Typo: "occupyied" > "occupied"

Done.

Line 295: Typo: "distiction" > "distinction"

Done.

Line 377: Could compare with Johns et al., (2011). Also worth noting that whilst the overturning transport itself is slightly higher than observation-based estimates suggest, the associated heat transport is lower than observation-based estimates suggest. This discrepancy suggests potential issues with the vertical structure of the transport and the ocean temperature and salinity, evidence of which can be seen in Figs. 12 and 14.

In the revised version of the manuscript we will add a comparison of the Atlantic meridional heat transport with estimates by Johns et al, 2011, together with a critical discussion of the relation of AMOC strength, heat transport and ocean temperature biases.

Line 433: Typo: "troposhere" > "troposphere"

Done.

Line 449: Suggest citing e.g. Held and Soden (2006) here as a reference for the relationship between the hydrological cycle and Clausius-Clapeyron. Do you see the same relationship between the amplification of the surface salinity pattern and the increasing hydrological cycle as Zika et al. (2018)?

We will follow the suggestion and add a reference to Held and Soden 2006. We will also analyze the pattern amplification of sea surface salinity in the model and compare it to Zika et al. 2018.

Line 466: Confusing sentence – reword.

We will rephrase the sentence.

Line 487: Not just EMICs, but also some low resolution AOGCMs (e.g. Hawkins et al. 2011). On this topic, have you calculated the $F_{ov}$ for CLIMBER-X? In contrast to observation-based estimates, many low resolution models show positive values of $F_{ov}$.

We will add a reference to hysteresis experiments performed with AOGCMs.
$F_{ov}$, computed as the difference of the AMOC freshwater transports across the southern and northern boundaries of the Atlantic (e.g. Liu 2017), is negative (around -0.05 Sv) at present in CLIMBER-X, in agreement with observations. However, the AMOC seems to be in a monostable state in the model.

Line 491: Typo: "equilibrim" > "equilibrium"

Done.

Line 506: Typo: "models'" > "model's"

Corrected.

Line 673: "anyway" is not necessary – remove

Removed.

"$K$" is multiply defined:

- Line 604: "$K$" is the kinematic vertical viscosity coefficient
- Line 697: the eddy kinetic energy, "$K$".

We will change the symbol for the kinematic vertical viscosity coefficient.

Line 767: Typo "atmophere" > "atmosphere", and "computed" > "compute"

Fixed.

Lines 866, 868: "Mcdougall" > "McDougall"

Fixed.

Lines 932, 934, 956: Suggest changing "ocean water" to "seawater"

Changed.

Line 944: Suggest rewording "Melting of sea ice leads to…" to "Melting of sea ice results in…" to avoid potential confusion with sea ice leads.

Changed.

Figures 11 and 29: The white text on the contours is difficult to read.

We will change the font color in order to make it better visible.

Figures 3, 11, 13, 24, 29: add (a) and (b) for consistency with other figure style

Done.

Hawkins, E., Smith, R. S., Allison, L. C., Gregory, J. M., Woollings, T. J., Pohlmann, H., and de Cuevas, B. (2011), Bistability of the Atlantic overturning circulation in a global climate model and links to ocean freshwater transport, *Geophys. Res. Lett.*, 38, L10605, doi:10.1029/2011GL047208.

Held, I. M., & Soden, B. J. (2006). Robust Responses of the Hydrological Cycle to Global Warming, *Journal of Climate*, *19*(21), 5686-5699, https://doi.org/10.1175/JCLI3990.1

Johns, W. E., Baringer, M. O., Beal, L. M., Cunningham, S. A., Kanzow, T., Bryden, H. L., Hirschi, J. J. M., Marotzke, J., Meinen, C. S., Shaw, B., & Curry, R. (2011). Continuous, Array-Based Estimates of Atlantic Ocean Heat Transport at 26.5°N, *Journal of Climate*, *24*(10), 2429-2449, https://doi.org/10.1175/2010JCLI3997.1

Mauritsen, T., Stevens, B., Roeckner, E., Crueger, T., Esch, M., Giorgetta, M., Haak, H., Jungclaus, J., Klocke, D., Matei, D., Mikolajewicz, U., Notz, D., Pincus, R., Schmidt, H., and Tomassini, L.: Tuning the climate of a global model, J. Adv. Model. Earth Sys., 4, M00A01, doi:10.1029/2012MS000154, 2012.

Williamson, D., Goldstein, M., Allison, L. *et al.* History matching for exploring and reducing climate model parameter space using observations and a large perturbed physics ensemble. *Clim Dyn* **41,** 1703–1729 (2013). https://doi.org/10.1007/s00382-013-1896-4

Zika, J.D., Skliris, N., Blaker, A.T., Marsh, R., Nurser, A.J.G., & Josey, S.A. (2018). Improved estimates of water cycle change from ocean salinity: the key role of ocean warming, *Environmental Research Letters,* 13(7), 074036

---

## Author Response (AR1)

**Response to reviewers**

We would like to thank the reviewers for the constructive comments on our paper.

Following the suggestions by reviewers #1 and #3, in the revised paper we have introduced additional information on the model tuning process. For that we added a new section 3 where we give more details about the model tuning strategy. And in the Appendix we added more information about the offline tuning of the radiation scheme and the snow grain size parameterisation, including 7 new figures.

In blue below is our point-by-point response to the reviewers' comments and suggestions (in black italic).

The changes made to the manuscript relative to the initial submission are highlighted in the attached pdf file.

**Reviewer #1**

*The paper presents a comprehensive description of the EMIC CLIMBER-X. This description is well balanced with a main text presenting the important principles and choices at the basis of model development as well as the model performance while the equations and parameters are given in the appendices. The model is designed as the successor of a very successful model, CLIMBER-2, which has been used extensively over the past 20 years. CLIMBER-X will very likely be as successful and this model description will thus be very useful for the scientists running the model and analyzing its results.*

*The paper is very clear and well written. It includes all the key elements needed for the description of a new model. I thus have mainly some minor suggestions of clarification. The only point that would deserve more discussion is the model tuning or calibration. Tuning is mentioned in the appendices and once in the main text for specific components as well as in the author's contribution but without much details or a global view of the way tuning was achieved. I think it would be nice to have a specific section, for example as a section 2.5, to explain the tuning strategy. I guess some parameters have defaults, fixed values that are not supposed to be modified. Some parameters in the various modules were tuned specifically 'offline' for those modules (e.g., lines 281, 820, 852) while some others may have been tuned so that the full model fits with observations. It would be interesting to describe which are the main parameters that were tuned, what were the calibration targets for this general tuning (for example global mean temperature, total amount of precipitation, sea ice extent ,temperature trend over the past century, etc.) and the method followed for the tuning.*

We would like to thank the reviewer for the positive comments on our paper.
Tuning is certainly a fundamental part of any model development and we followed the suggestion by the reviewer (and reviewer #3) and added a new section 3 where we give more details about the model tuning strategy.

*Minor points*

*Lines 7-8. I would be more specific in the abstract on the performance of the model, giving some examples where the agreement with observations is good and mention the limitations, such as the poor performance in the tropics.*

We added the following general sentence about model limitations to the abstract:
*"Limitations and applicability of the model are critically discussed."*
We think that this should be sufficient to point the interested reader to the more detailed description of the model limitations in the main text.

*Line 101. I would suggest to state at this stage that the two dimensions are latitude and longitude. If I follow well, all the vertical variations are then prognostic only. If it is the case, that would be good to mention it explicitly here. It is also probably worth insisting on this point for comparisons with observations including vertical variations (like figures 3, 6 and 8).*

In the revised manuscript, we explicitly state that the 2 dimensions are latitude and longitude and that the vertical variations are *diagnostic* only. We added a few sentences to insist on this point also when discussing the figures on the 3D atmospheric structure.

*Sulphate aerosols are mentionned first line 159 but the spatial distribution is discussed line 166. This may give the feeling that this distribution is only valid for longwave.*

We modified this. Where sulfates are mentioned for the first time in the paper, we specify that the spatial distribution of sulfate aerosols has to be prescribed in the model.

*Line 171. It would be useful to justify in one or two sentences why an approach even simpler that the one in CLIMBER- 2 has been retained.*

We did not find any advantages in a separate treatment of stratus and cumulus clouds in the framework of our modelling approach. We added this sentence to the revised manuscript.

*Line 200. What is exactly meant by 'lack of a Gulf Stream extension propagating to high latitudes'? Does it have an impact on model biases, on the location of deep-water formation or a link with the AMOC simulated by the model?*

We rephrased this to: "*One notable consequence of the strong momentum damping is a generally weak Antarctic Circumpolar Current and Gulf Stream. See Edwards et al. 2005 and Muller et al. 2006 for a more detailed discussion of the limitations of the frictional-geostrophic dynamics.*"

*Line 250. Are the fluxes in the open ocean computed in the same way as in the ice-free fraction of the grid cells?*

Yes, the open ocean fluxes are computed the same way as in the ice-free fraction of the partly ice-covered grid cells.

*Line 254. Are the latent and sensible heat fluxes computed in the same way over the ocean? More generally, the computation of the surface balance is well explained here for SISIM in section 2.3 but not explicitly for the oceanic part in section 2.2.*

The latent and sensible heat fluxes are computed the same way over the ocean, but since the air-sea fluxes are computed in the sea ice model (even in the absence of sea ice), they are described as part of the sea ice model. In this respect, SISIM is not only the sea ice model but also the atmosphere-ocean interface. We agree that this might be confusing to the reader and we have therefore added a sentence to clarify this point, right at the beginning of the SISIM description in the revised paper:

*"SISIM also serves as coupler between atmosphere and ocean, hence all surface energy fluxes over both sea ice and open ocean are computed in the sea ice model."*

*Line 276. This would be useful to add here a few sentences describing the main characteristics of the model without the need to read Willeit and Ganopolski (2016).*

In the revised paper we added a few sentences describing the main features of PALADYN, to give the reader a short overview of the model without having to read the separate PALADYN description paper.

*Line 378-380. Is the deep water formed in Antarctica above the continental shelf or because of open ocean convection?*

Deep water around Antarctica is formed mainly on the continental shelf and not through open ocean convection. We added this information to the revised manuscript.

*Caption of Figure 28. The reference to Paul et al. 2021 is not the same as on the figure (GLOMAP).*

We changed the corresponding legend entry in the figure from GLOMAP to Paul et al. (2021).

**Reviewer #2**

*Paper provides detail description of new version of CLIMBER model and evaluation of model performance for different historical periods, as well as model response to the changes in atmospheric $CO_2$ concertation. Present day climate, climate change over resent historical period (1850-2100) and response to both doubling and transient increase of the atmospheric $CO_2$ concentration simulated by CLIMBER agree well with available observations and results obtained with more sophisticated climate modes. At the same time, there are noticeable differences in climate state for Last Glacial Maximum between CLIMBER and CMIP5 models.*

*This does not necessarily mean that CLIMBER is wrong, but in my view, additional validation would be useful. As said in the paper "The atmospheric component of CLIMBER X*

*is based on a statistical-dynamical approach, which employs a number of significant simplifications and assumptions." The fact that "these simplifications and a set of parameterizations explicitly derived from present–day climate limit the models' applicability to climate states fundamentally different from the present one". Authors mention Snowball Earth case as such climate state. Possibly Last Glacial Maximum climate is also different enough from preset one.*

As noted by the reviewer, there are indeed some differences in the simulated LGM climate between CLIMBER-X and CMIP models. Part of the differences could be a result of CLIMBER-X having dynamic vegetation, while many CMIP models don't. We tested the effect of the vegetation feedback by running an additional LGM simulation with prescribed present-day vegetation. The simulation indicates that changes in vegetation cover account for ~0.5°C of the LGM cooling. This information has been added to the revised manuscript. We would also argue that when it comes to e.g. global cooling and AMOC state, CLIMBER-X results are possibly in better agreement with reconstructions than (some) CMIP models. This is explicitly stated in the paper. Consistent with the relatively cold simulated LGM climate, sea-ice expansion is at the top of the range of CMIP models. In the revised paper we also included a comparison of SH ice-area changes from reconstructions (red and blue dashed lines in figure below) and added the following sentence: *"The relatively large simulated sea ice expansion in the Southern Ocean is also largely consistent with the latest proxy reconstructions by Lhardy et al., 2021."*

[Figure]

The LGM is definitely not outside of the applicability range of the model, as CLIMBER-X has been designed specifically also to simulate past glacial cycles. That is also the reason why we explicitly included a comparison of the simulated LGM state with state-of-the-art CMIP models in the paper.

*As indicated in the paper "CLIMBER-X includes code to diagnose the strength of the different climate feedbacks". I would suggest calculating climate feedbacks for LGM climate and comparing them with estimates available in literature.*

At present, the feedback diagnostics code in CLIMBER-X can operate only with changes in atmospheric CO2. It is therefore not possible to directly calculate climate feedbacks for the LGM state. Instead we performed additional feedback analyses for CO2 doubling from 140 ppm to 280 ppm and for 560 ppm to 1120 ppm to investigate the state dependence of the different feedbacks. We also repeated the feedback analyses with prescribed present-day

vegetation in order to isolate the effect of the vegetation feedback. The results are summarized in a new figure in the revised paper (new Fig. 24) and show a strong increase of albedo and vegetation feedbacks for colder climates:

[Figure]

This analysis provides additional information on the climate feedbacks in the model and therefore also sheds some light on the mechanisms operating in the LGM simulations.

*Specific comments.*

*There seems to be discrepancy between Table 1 and Table 2.*

*Total evaporation in Table 2 are larger than observation, while latent heat in table 1 is smaller*

This could be a result of the use of different observation-based estimates for the energy fluxes and for the hydrological cycle and thus ultimately a consequence of the uncertainty in these estimates. We added a short discussion on this point in the revised manuscript.

*On page 14 (near line 355).*

*Following sentence: "During the winter months and at high latitudes, CLIMBER-X also captures the near-surface temperature inversions." It is better to say, "During the winter months, CLIMBER-X also captures the near-surface temperature inversions at high latitudes"*

We changed the mentioned sentence accordingly.

**Reviewer #3**

*This paper serves as an introduction to the CLIMBER-X Earth system model. I found it to be extremely well written, and therefore pleasant to read. It describes in detail all of the key components of the model and how they interact, and then follows this with an evaluation of the model performance over the period 1850-present, which most readers will easily be able to relate to due to the authors' efforts to compare their simulation with both observations and the CMIP simulations and assessments.*

*CLIMBER-X itself performs well in almost every aspect shown, and the authors also make clear the model's deficiencies and limitations, which is crucial for potential users with applications in mind. As a paper intended to present a new model to the community, I think it serves that purpose well.*

We would like to thank the reviewer for the positive appraisal of our paper.

*Minor comment:*

*I think it would be of interest to the reader to know what degree of calibration has been done, and how. There are a couple of comments in the text, but I think the 'dark art' of how we tune our numerical models should be made more visible (see e.g. Mauritsen et al., 2012 for a good example). In this case, where it is clear that CLIMBER-X is well suited to methods of performing robust exploration of parameter space (see e.g. Williamson et al., 2013), it would be interesting to know the extent to which this has been done, or remains to be done. I suggest including a section (or possibly a subsection in 2), that describes the calibration (tuning) that has been done to the model. For example, were components, or schemes within the individual models tuned in isolation, or was the entirety of CLIMBER-X calibrated as one? To what observations or metrics was the model tuned to?*

As mentioned already in the response to Reviewer #1, who raised a similar point, we followed the suggestion and added a new section where we give more details about the model tuning strategy.

*Specific comments/corrections:*

*Line 77/78: Replace "16 CPUs" with "2 x 8 core CPUs". Also, if your computation is on a single node then the detail "Infiniband FDR14 Lenovo/IBM" is redundant and can be removed.*

We changed this as suggested.

*Line 96: I suggest "We made extensive use of…" instead of "We extensively made use of…"*

We changed this as suggested.

*Line 162: Seem to jump from Appendix 5 (line 153) to Appendix 7. I suggest carefully checking and renumbering/reordering as required.*

We shifted the description of cloud parameterisations up, so it now appears before the paragraphs dedicated to radiation. This fixes the order in which the different Appendix section are referenced to in the main text.

*Line 233: question about rigid lid pressure?*

We added a sentence to clarify this in the revised manuscript:
*"The rigid-lid formulation does imply a surface pressure, the so-called rigid-lid pressure, which is directly related to the sea surface height. However, instead of solving the non-trivial equation for surface pressure, in the model we use an approximate approach in which surface pressure is simply diagnosed from integrating density above a reference depth."*

*Line 269: Typo: "occupyied" > "occupied"*

Done.

*Line 295: Typo: "distiction" > "distinction"*

Done.

*Line 377: Could compare with Johns et al., (2011). Also worth noting that whilst the overturning transport itself is slightly higher than observation-based estimates suggest, the associated heat transport is lower than observation-based estimates suggest. This discrepancy suggests potential issues with the vertical structure of the transport and the ocean temperature and salinity, evidence of which can be seen in Figs. 12 and 14.*

In the revised version of the manuscript we added the following sentence:
*"The maximum meridional heat transport by the Atlantic ocean is 1.12 PW, slightly lower than observation–based estimates of 1.25 PW (e.g. Johns et al., 2011). The discrepancy between stronger than observed AMOC and the weaker than observed Atlantic meridional heat transport can be explained by biases in the simulated vertical structure of the transport (Fig. 12) and in the Atlantic ocean temperature field (Fig.14)."*

*Line 433: Typo: "troposhere" > "troposphere"*

Done.

*Line 449: Suggest citing e.g. Held and Soden (2006) here as a reference for the relationship between the hydrological cycle and Clausius-Clapeyron. Do you see the same relationship between the amplification of the surface salinity pattern and the increasing hydrological cycle as Zika et al. (2018)?*

We followed the suggestion and added a reference to Held and Soden 2006. The salinity pattern amplification in the 1%/year $CO_2$ increase simulation in CLIMBER-X is ~5% per °C global warming (see panel c in figure below), in good agreement with estimates from Zika et al., 2018 for the historical period. The salinity pattern amplification is therefore larger than the hydrological sensitivity of the model, in accordance with Zika et al., 2018. It is beyond the scope of this study however to analyze whether the difference between the salinity pattern amplification and hydrological sensitivity can largely be explained by ocean warming, as suggested by Zika et al., 2018. We added this discussion to the revised manuscript and

extended the hydrological sensitivity figure with a panel showing the salinity pattern amplification:

[Figure]

*Line 466: Confusing sentence – reword.*

We rephrased the sentence.

*Line 487: Not just EMICs, but also some low resolution AOGCMs (e.g. Hawkins et al. 2011). On this topic, have you calculated the $F_{ov}$ for CLIMBER-X? In contrast to observation-based estimates, many low resolution models show positive values of $F_{ov}$.*

We added a reference to hysteresis experiments performed with AOGCMs.
$F_{ov}$, computed as the difference of the AMOC freshwater transports across the southern and northern boundaries of the Atlantic (e.g. Liu 2017), is negative (around -0.05 Sv) at present in CLIMBER-X, in agreement with observations. However, the AMOC seems to be in a monostable state in the model.

*Line 491: Typo: "equilibrim" > "equilibrium"*

Done.

*Line 506: Typo: "models'" > "model's"*

Corrected.

*Line 673: "anyway" is not necessary – remove*

Removed.

*"K" is multiply defined:*

- *Line 604: "K" is the kinematic vertical viscosity coefficient*
- *Line 697: the eddy kinetic energy, "K".*

We changed the symbol for the kinematic vertical viscosity coefficient to Kv.

*Line 767: Typo "atmophere" > "atmosphere", and "computed" > "compute"*

Fixed.

*Lines 866, 868: "Mcdougall" > "McDougall"*

Fixed.

*Lines 932, 934, 956: Suggest changing "ocean water" to "seawater"*

Changed.

*Line 944: Suggest rewording "Melting of sea ice leads to…" to "Melting of sea ice results in…" to avoid potential confusion with sea ice leads.*

Changed.

*Figures 11 and 29: The white text on the contours is difficult to read.*

We removed the numbers from the figures, as they are difficult to read and don't add much information compared to the color scheme.

*Figures 3, 11, 13, 24, 29: add (a) and (b) for consistency with other figure style*

Done.

*Hawkins, E., Smith, R. S., Allison, L. C., Gregory, J. M., Woollings, T. J., Pohlmann, H., and de Cuevas, B. (2011), Bistability of the Atlantic overturning circulation in a global climate model and links to ocean freshwater transport, Geophys. Res. Lett., 38, L10605, doi:10.1029/2011GL047208.*

*Held, I. M., & Soden, B. J. (2006). Robust Responses of the Hydrological Cycle to Global Warming, Journal of Climate, 19(21), 5686-5699, https://doi.org/10.1175/JCLI3990.1*

*Johns, W. E., Baringer, M. O., Beal, L. M., Cunningham, S. A., Kanzow, T., Bryden, H. L., Hirschi, J. J. M., Marotzke, J., Meinen, C. S., Shaw, B., & Curry, R. (2011). Continuous, Array-Based Estimates of Atlantic Ocean Heat Transport at 26.5°N, Journal of Climate, 24(10), 2429-2449, https://doi.org/10.1175/2010JCLI3997.1*

*Mauritsen, T., Stevens, B., Roeckner, E., Crueger, T., Esch, M., Giorgetta, M., Haak, H., Jungclaus, J., Klocke, D., Matei, D., Mikolajewicz, U., Notz, D., Pincus, R., Schmidt, H., and Tomassini, L.: Tuning the climate of a global model, J. Adv. Model. Earth Sys., 4, M00A01, doi:10.1029/2012MS000154, 2012.*

*Williamson, D., Goldstein, M., Allison, L. et al. History matching for exploring and reducing climate model parameter space using observations and a large perturbed physics ensemble. Clim Dyn **41,** 1703–1729 (2013). https://doi.org/10.1007/s00382-013-1896-4*

*Zika, J.D., Skliris, N., Blaker, A.T., Marsh, R., Nurser, A.J.G., & Josey, S.A. (2018). Improved estimates of water cycle change from ocean salinity: the key role of ocean warming, Environmental Research Letters, 13(7), 074036*